# Dopaminergic action prediction errors serve as a value-free teaching signal

Francesca Greenstreet[1,5], Hernando Martinez Vergara[1,2,5], Yvonne Johansson[1,5], Sthitapranjya Pati[1], Laura Schwarz[1], Stephen C. Lenzi[1], Jesse P. Geerts[1,3], Matthew Wisdom[1], Alina Gubanova[1], Lars B. Rollik[1], Jasvin Kaur[1], Theodore Moskovitz[4], Joseph Cohen[1], Emmett Thompson[1], Troy W. Margrie[1], Claudia Clopath[1,3] & Marcus Stephenson-Jones[1✉]

Choice behaviour of animals is characterized by two main tendencies: taking actions that led to rewards and repeating past actions[1,2]. Theory suggests that these strategies may be reinforced by different types of dopaminergic teaching signals: reward prediction error to reinforce value-based associations and movement-based action prediction errors to reinforce value-free repetitive associations[3–6]. Here we use an auditory discrimination task in mice to show that movement-related dopamine activity in the tail of the striatum encodes the hypothesized action prediction error signal. Causal manipulations reveal that this prediction error serves as a value-free teaching signal that supports learning by reinforcing repeated associations. Computational modelling and experiments demonstrate that action prediction errors alone cannot support reward-guided learning, but when paired with the reward prediction error circuitry they serve to consolidate stable sound–action associations in a value-free manner. Together we show that there are two types of dopaminergic prediction errors that work in tandem to support learning, each reinforcing different types of association in different striatal areas.

When animals and humans make choices, they exhibit two key tendencies: pursuing rewarding actions and repeating past actions[1,2]. Dopamine neurons that encode reward prediction error (RPE)[3] provide a critical teaching signal for value-based learning, reinforcing actions that lead to reward. Recently it has been proposed that repetitive 'habitual' choices may instead be updated by a movement-based teaching signal[4–6].

The hypothesized teaching signal encodes an action prediction error (APE), the difference between the action that is taken and the extent to which the action was predicted. Although experimental evidence is lacking, theoretical models suggest that this value-free learning system could operate alongside canonical value-based learning in the basal ganglia[5,6]. These models predict two types of dopaminergic teaching signals: RPE, which reinforces reward-driven actions, and APE, which reinforces repeated actions[4,6]. Notably, dopaminergic neurons located in the ventral tegmental area (VTA) and medial parts of the substantia nigra pars compacta (SNc) encode RPE[7,8], whereas those in the lateral SNc and substantia nigra pars lateralis (SNl), which project to the tail of striatum (TS), prominently respond to movement[9–12].

Here we investigated whether movement-related dopaminergic activity encodes an APE, providing a value-free teaching signal to reinforce repeated state–action associations. To address this, we examine the role of dopamine in the TS as mice learn an auditory discrimination, cloud-of-tones (COT)[13] task. We show that dopaminergic input to this region is important for learning, and that dopaminergic activity during the task is movement-related. We go on to show that this movement-related dopaminergic activity encodes an APE. Causal manipulations show that this value-free teaching signal can reinforce state–action associations, essentially biasing mice to repeat the actions that they have taken in the past. We propose a model where this movement-based teaching signal works in conjunction with canonical RPE signalling to boost and stabilize learning about consistent state–action associations.

## TS dopamine facilitates learning

In the COT task, mice initiate trials by nose-poking a central port, triggering auditory stimuli composed of a short train of overlapping pure tones (5–40 kHz). They selected a left or right reward port on the basis of whether the stimulus contained primarily low (5–10 kHz) or high (20–40 kHz) frequencies (Fig. 1a and Extended Data Fig. 1a). In line with previous reports[14], bilateral inactivation of the TS with muscimol (a GABA$_A$ (γ-aminobutyric acid type A) receptor agonist), but not the dorsomedial striatum (DMS), impaired task performance (Fig. 1b,c) in expert mice. Unilateral optogenetic inactivation of either type of striatal projection neurons (SPNs) in the TS also had an opposing and significant effect on choices of mice (Fig. 1d–f and Extended Data Fig. 1b,c). These results demonstrate that the TS is needed to execute the learned behaviour and that both populations of SPNs exert opposing contributions to the auditory-guided choices.

To test whether the TS was also needed to learn the task, we ablated the TS prior to training using a viral-mediated caspase-based strategy

[1]Sainsbury Wellcome Centre for Neural Circuits and Behaviour, University College London, London, UK. [2]Institut d'Investigacions Biomèdiques August Pi i Sunyer (IDIBAPS), Barcelona, Spain. [3]Bioengineering Department, Imperial College London, London, UK. [4]Gatsby Computational Neuroscience Unit, University College London, London, UK. [5]These authors contributed equally: Francesca Greenstreet, Hernando Martinez Vergara, Yvonne Johansson. ✉e-mail: m.stephenson-jones@ucl.ac.uk

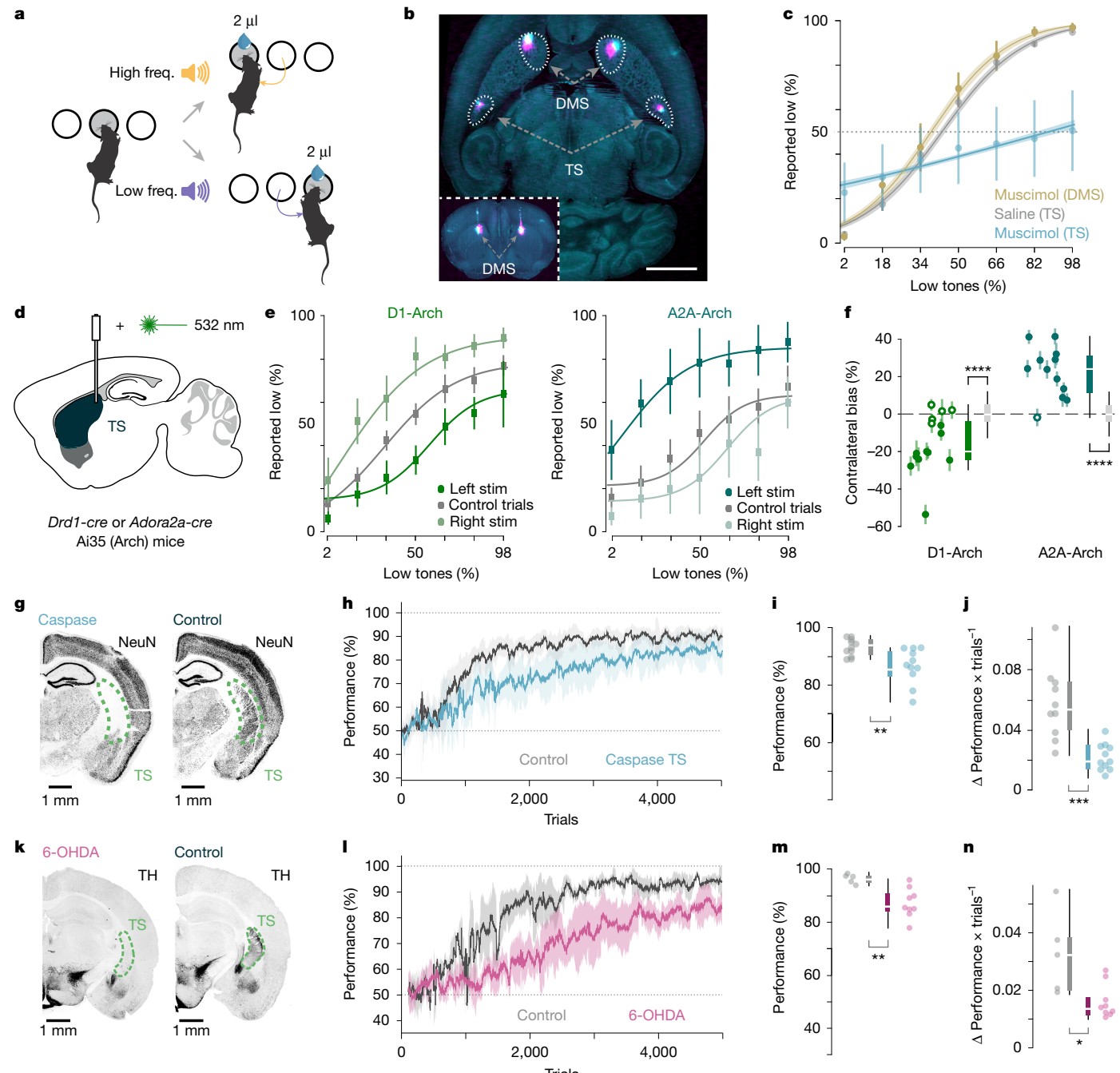

**Fig. 1 | TS is needed to facilitate learning and for execution of the auditory discrimination task. a**, Schematic of the task. Frequencies (freq.) represent auditory stimuli, volumes represent reward. **b**, Muscimol injection locations, indicated by the co-injection of fluorescent cholera toxin B. Scale bar, 2 mm. **c**, Psychometric task performance of mice, saline in TS (2 mice; 4 sessions), DMS (3 mice; 11 sessions) and TS (5 mice; 15 sessions). Lines represent logistic fits of the means. **d**, Schematic for inhibition of D1 SPNs or D2 SPNs. **e**, Psychometric task performance for opto-stimulated trials for the D1 (*Drd1-cre*) archaeorhodopsin (Arch) (8 mice; 15 sessions) and A2A (*Adora2a-cre*) Arch (8 mice; 12 sessions) mice. Lines represent logistic fits of the means. Stim, stimulation. **f**, Quantification of the bias for each session shown in **e**. Scatter dots represent the mean for each session, and error bars illustrate the variation of shuffled data (Methods). Colour-filled dots indicate $P < 0.05$. D1-Arch: $P = 2.27 \times 10^{-5}$, Cohen's $d = -1.40$;

A2A-Arch: $P = 2 \times 10^{-7}$, Cohen's $d = 2.33$; Kruskal–Wallis test. **g**, Example lesion and control mice histology. **h**, Learning rate of the lesioned TS ($n = 11$ mice) and control ($n = 10$ mice) groups (lines represents group means). **i,j**, Maximum performance (**i**; $P = 0.0023$, Kruskal–Wallis test; Cohen's $d = -1.70$) and maximum learning rate (**j**; $P = 0.0006$, Kruskal–Wallis test; Cohen's $d = -2.04$) between the groups (same mice as in **h**). **k**, Example TH staining in the TS for control and lesion mice. **l**, Learning rate of the TS dopamine-ablated ($n = 9$ mice) and control ($n = 5$ mice) groups. **m,n**, Differences on the maximum performance (**m**; $P = 0.006$, Kruskal–Wallis test; Cohen's $d = -2.39$) and maximum learning rate (**n**; $P = 0.014$, Kruskal–Wallis test; Cohen's $d = -1.71$) between the groups (same mice as **l**). Error bars indicate s.d. in **c,e,h,l**. In box plots, boxes represent quartiles 2 and 3, the centre line shows the median and whiskers extend to the furthest data point within 1.5 times the inter-quartile range from the box.

(Fig. 1g and Extended Data Fig. 2a,b). Lesions of the TS caused a deficit in learning (Fig. 1h–j and Extended Data Fig. 2c–f), reducing both the learning rate and the maximum performance reached on the task

(Fig. 1i,j). To confirm that the deficits in learning are not due to an inability of the mice to act upon the learned sound–action association we infused an *N*-methyl-ᴅ-aspartate (NMDA) receptor antagonist,

D-2-amino-5-phosphonovalerate (D-AP5) in the TS (Extended Data Fig. 3a,b). D-AP5 infusions did not affect performance in expert mice (Extended Data Fig. 3c) but significantly impaired learning when infused during training (Extended Data Fig. 3d–g). In addition, ablation of the TS-projecting dopamine neurons also recapitulated the general TS lesions effects. TS dopamine-ablated mice had deficits in learning (Fig. 1k–n and Extended Data Fig. 3h–p), without influencing the time taken to move from the centre port to the choice ports (Extended Data Fig. 3k), or the time taken between trials (Extended Data Fig. 3l). Together, these results confirm that both the TS and its dopaminergic innervation are required to facilitate learning and execute the auditory discrimination task.

## TS dopamine release is correlated with movement

To understand the role of TS dopamine in the task, we measured its dynamics with dLight1.1, a genetically encoded fluorescent dopamine sensor[15]. TS dopamine responses correlated in time with contralateral movements from the centre port, in sharp contrast to the large reward responses in ventral striatum (VS) (Fig. 2a–e and Extended Data Fig. 4).

To separate dopamine responses associated with overlapping behavioural events, we applied a linear regression model to photometry data obtained early in training. The model included three event types: cue (centre port entry), choice (centre port exit) and outcome (side port entry). VS responses were best explained by the outcome kernel, capturing the large responses to rewards and dips for unrewarded trials (Fig. 2e–g and Extended Data Fig. 4c–f). By contrast, TS showed minimal outcome-related dopamine activity (Fig. 2e–g and Extended Data Fig. 4c–f). In line with other studies[16–18], we observed dopaminergic reward responses when we recorded in the posterior region of the dorsolateral striatum (pDLS), which is located just rostral to the TS and is not prominently innervated by the primary auditory cortex[19] (Extended Data Fig. 4a,b,n,o). The largest TS dopaminergic response was contralateral movement-locked activity (Fig. 2e–g), which was also seen when mice made contralateral movements to return from the side ports to the centre port (Extended Data Fig. 4j–m). VS movement-related activity was smaller, with no significant difference between contralateral and ipsilateral actions (Extended Data Fig. 4g–i). These results show that VS dopamine activity significantly encodes reward outcome, consistent with RPE, whereas the TS dopamine activity encodes movement information.

To confirm that TS dopamine activity was unrelated to sound, we omitted the cue on some trials (Extended Data Fig. 5a–g) and found no significant difference in response. To assess task dependence, we recorded TS dopamine activity as mice explored an open arena. As in-task recordings, TS dopamine increased during contralateral movements (Extended Data Fig. 5h–j), and its signal scaled with movement amplitude (Extended Data Fig. 5k–n). Turn angle significantly correlated with TS dopamine (Extended Data Fig. 5n), a correlation that was absent in VS. These results confirm that TS dopamine encodes information about movement.

To determine whether there was any additional sensory response to the sound stimuli, we trained mice on a version of the task in which sounds were played when the mice left the side ports and returned to the centre port to initiate the next trial (Extended Data Fig. 5o). There was no significant response to the sound in this version of the task but the response to contralateral orienting movement from the centre port remained (Extended Data Fig. 5p–t). In addition, there was no significant response when the sound cues were played as the mice freely explored an open arena (Extended Data Fig. 5u–y), confirming that the recorded TS dopamine signal correlated with movement and not sound.

## Evidence for APE

Our results indicate that TS dopamine is crucial for learning the auditory frequency discrimination task and encodes information about movement rather than reward or sound. Although dopamine RPEs are known to drive cortico-striatal plasticity and learning[3,20–22], the role of movement-related dopamine remains unclear. Habit-formation models predict that movement-related dopamine could encode a value-free APE, representing the discrepancy between an executed action and its predicted occurrence in a given state[4,6].

If the movement-related dopaminergic activity in the TS encodes an APE, then dopamine activity in the TS should decrease over time as mice learn to predict the action that they will take in response to the sound. To test this prediction, we recorded TS and VS dopamine signals over the course of learning and compared these signals to values of RPE and APE from a dual value-based/value-free reinforcement learning model (Extended Data Fig. 6). Characteristic of an RPE signal, cue-related dopamine activity in the VS grew over time (Fig. 3a–d) after an initial fall in the response that might be due to changes in stimulus salience or alternative interpretation of VS dopamine signal[23–25]. In line with encoding an APE signal, the movement-related dopamine activity in the TS decreased as mice learned the task (Fig. 3e–h). This decrease resembled the exponential decay of the value-free system of the model (Fig. 3g).

To confirm that this decay in the signal was not due to changes in the way that mice were moving and performing the task, we tracked changes in speed and kinematics over the course of learning. Fitting a linear regression model that considered the log trial number, speed and turn angle in each mouse revealed that the change in TS dopamine signal that is predicted by log trial number persisted even once the proportion of the signal that could be predicted by movement alone was accounted for (Extended Data Fig. 6c–l). The majority of the explained variance in a full model with movement and log trial number was captured by changes in the trial number rather than changes in speed or turn angle (Extended Data Fig. 6m).

If the movement-related dopaminergic activity in the TS encodes an APE, then changes in the dopamine activity in the TS should also reflect the recent history of sound–action pairings that have been experienced. Therefore, the size of the dopamine signal should be smaller when the same action is taken in response to the same sound on subsequent trials (Fig. 3i). In agreement with this, the TS dopamine response was on average significantly smaller when mice repeated the same action in response to the same stimulus in the past trial (Fig. 3j,k). By contrast, the size of the cue response in the VS was larger when the correct sound–action pairing was repeated on subsequent trials (Fig. 3l,m), consistent with an RPE signal as the sound–action pairing would have a larger predicted value on the subsequent trial (Fig. 3l). Together these results show that the movement-related dopamine activity in the TS does not reflect a pure movement signal that is invariably linked to action characteristics, rather the size of the signal changes on a trial-by-trial basis depending on how predictable the sound–action association is, as expected from an APE signal.

The learning-associated changes in movement-related TS dopamine suggests that responses are modulated by how predictably an action follows a given state (auditory cue). Therefore, if trained mice were to make a familiar movement in response to an unfamiliar sound, the action would not be predicted by the stimulus and consequently the resulting APE should be larger. To examine this hypothesis, we replaced the auditory cue associated with the contralateral action with a novel cue (white noise) (Fig. 3n,o). The novel stimulus decreased the accuracy of mice (Extended Data Fig. 6o–t) and significantly increased the movement-related TS dopamine signal (Fig. 3p,q), as expected from an APE signal (Fig. 3o). There was no significant TS dopamine response to the unfamiliar white noise stimulus when it was played as mice freely explored an open arena (Extended Data Fig. 5x,y). As predicted by the RPE model, the VS dopamine cue response to the novel sound was smaller (Fig. 3r,s), as there is less value predicted by the novel cue (Fig. 3o).

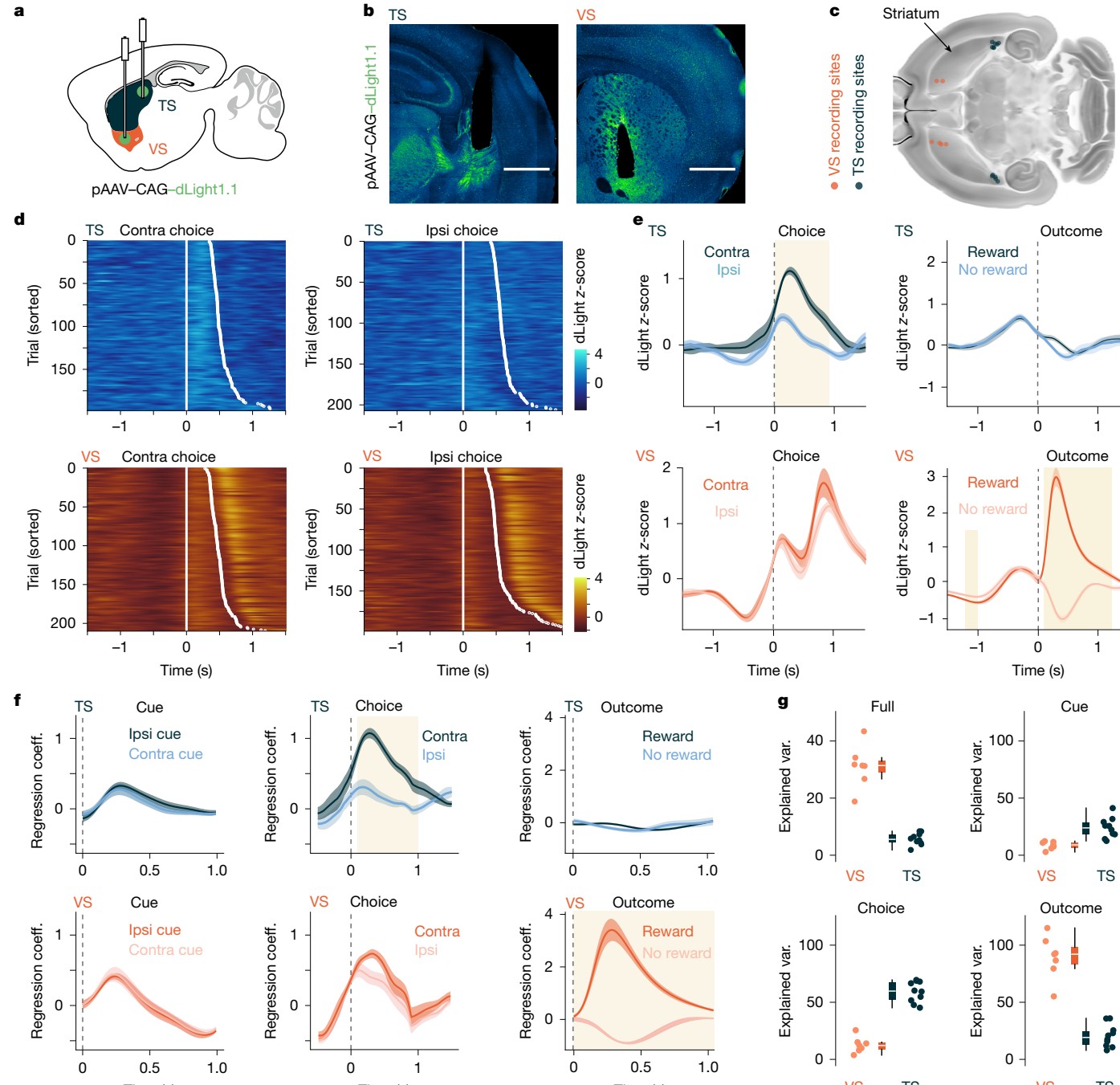

**Fig. 2 | TS dopamine release is correlated with contralateral movement.**
**a**, Schematic of the experimental approach for recording dopamine activity in the VS and TS. The anterior–posterior distance between the VS and TS are not to scale. **b**, Fluorescent images showing optical fibre locations and dLight expression in the TS and the VS. Scale bar, 1 mm. **c**, Recording fibre locations from the VS and TS matched to the reference atlas. **d**, Example of a TS and a VS recording session aligned to time of leaving the centre port to make a contralateral (contra) choice or an ipsilateral (ipsi) choice. White dot shows time of entering contralateral or ipsilateral choice port. **e**, Average photometry traces in TS (n = 10 mice) and VS (n = 7 mice) aligned to task events. Shaded time windows show significant differences between the two trace types in each subplot, calculated by performing two-sample *t*-tests on 0.1-s bins and a *P* value threshold for significance of 0.01. **f**, Average response kernels to behavioural events for recordings in the TS and VS. Shaded time windows are calculated as in **e**. Coeff., coefficient. **g**, Percentage explained variance (var.) of the whole recording session (VS median = 31.3; TS median = 5.50) and for different behavioural kernels for linear regression models fitted on VS (n = 7 mice) and TS (n = 10 mice) recordings. Error bars represent s.e.m. in **e**,**f**. In box plots, boxes represent quartiles 2 and 3, the centre line shows the median and whiskers extend to the furthest data point within 1.5 times the inter-quartile range from the box.

Another hallmark of an APE is that it should be value-free and should not be modulated by either the outcome or the predicted value of an action. In agreement with this, the dopamine signal in the TS did not significantly respond to a larger or smaller than predicted reward (Extended Data Fig. 7a–d). By contrast, the VS dopamine signal was altered in a manner consistent with RPE (Extended Data Fig. 7b,e,f). The size of the TS dopamine signal was also not modulated by changes in the predicted value of an action (Extended Data

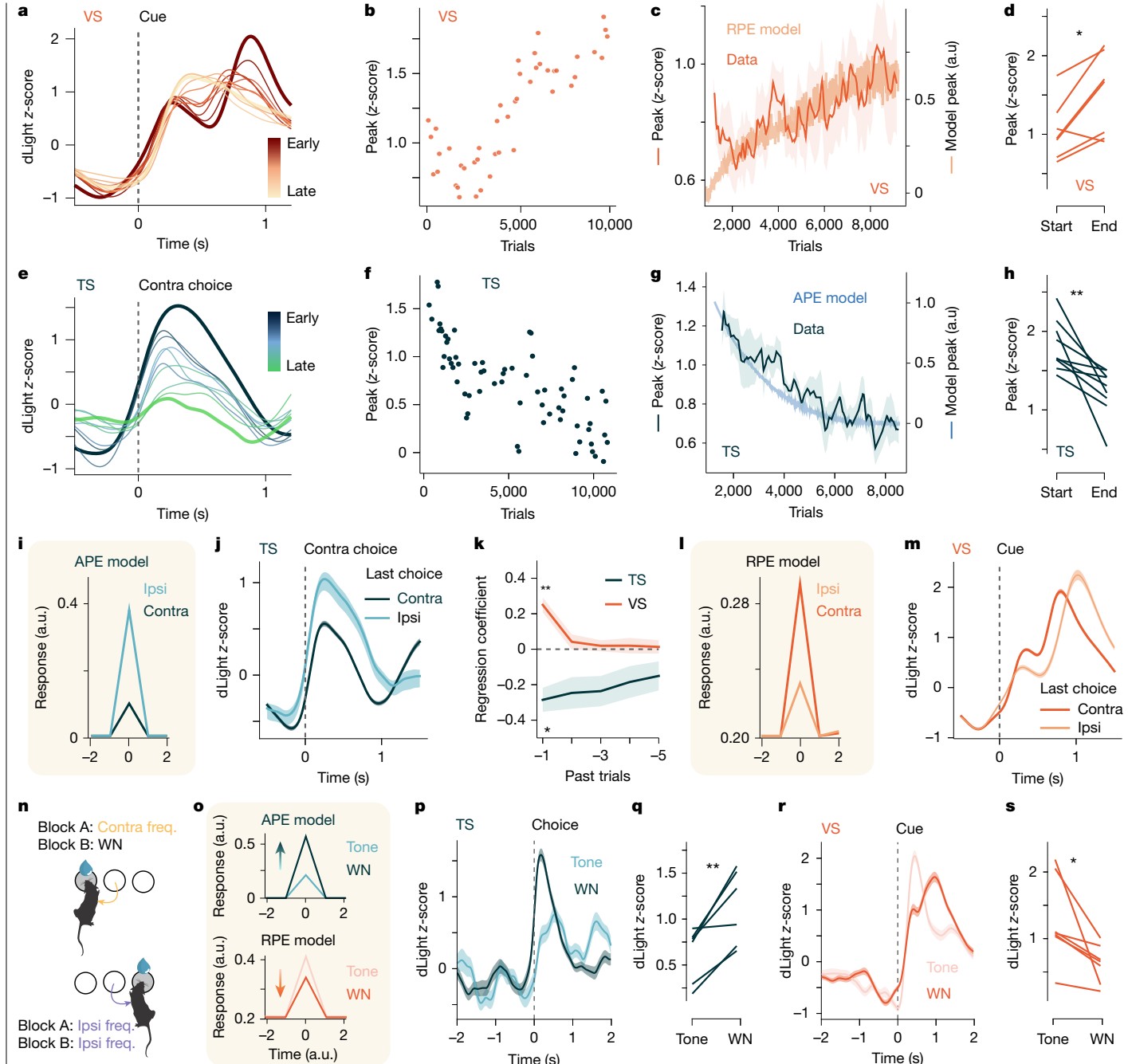

**Fig. 3 | TS dopamine release is consistent with encoding APE. a**, dLight recording in the VS. Each trace is the average of 200 trials. **b**, Example VS dopamine response size to contralateral cue binned every 40 trials. **c**, Average change in contralateral cue-aligned VS dopamine response. The solid orange represents the mean (*n* = 7 mice). The light orange trace is the mean predicted RPE response from 100 model agents. a.u., arbitrary units. **d**, Size of contralateral cue-aligned dopamine response in VS in the first and last session of training (*n* = 7 mice), *P* = 0.016 (paired two-sided *t*-test), Cohen's *d* = −1.25. **e**–**g**, As **a**–**c** but for TS recordings (*n* = 6 mice). **h**, As **d** but for the TS (*n* = 9 mice), *P* = 0.006 (paired two-sided *t*-test). Cohen's *d* = 1.19. **i**, Modelled responses for APE at the time of correct contralateral choice if the previous choice for that stimulus was ipsilateral or contralateral. **j**, As **i** but for an example average (mean) TS dopamine response.

**k**, Regression coefficients. One-sided *t*-test against zero, corrected using the Bonferroni method for multiple comparisons. VS: *n* = 7 mice, *P* = 0.005, 1.0, 1.0, 1.0, 1.0 (left to right), (Cohen's *d* = 2.23, 0.37, 0.23, 0.17, 0.13 (left to right)). TS: *n* = 6 mice, *P* = 0.04, 0.20, 0.20, 0.47, 0.63 (left to right), (Cohen's *d* = −1.72, −1.13, −1.13, −0.84, −0.75 (left to right)). **l**,**m**, As **i**,**j** but for the VS response. **n**, Task design. WN, white noise. **o**, Modelled APE and RPE signals following the state change. **p**, Example TS dopamine responses to the contralateral choice in response to the normal or the white noise cue. **q**, TS dopamine response to the contralateral action before and after the introduction of the novel state (*P* = 0.01, paired two-sided *t*-test) (*n* = 6 mice), Cohen's *d* = −1.81. **r**, As **p** but for a VS recording aligned to cue. **s**, As **q** but for VS recording aligned to cue. (*P* = 0.02, Wilcoxon signed-rank test) (*n* = 7 mice), Cohen's *d* = 1.04. Error bars represent s.e.m.

Fig. 7g–i), in contrast to the RPE-consistent VS signal (Extended Data Fig. 7h,k,l).

Finally, we compared our data to other theoretical models of dopamine activity, movement per se, novelty or salience (Methods). Unlike the models for APE and RPE, none of the models consistently matched the pattern of modulation across conditions that we observed in our recordings (Extended Data Fig. 8). Together, our results reveal that the movement-related TS dopamine signal is consistent with encoding APE.

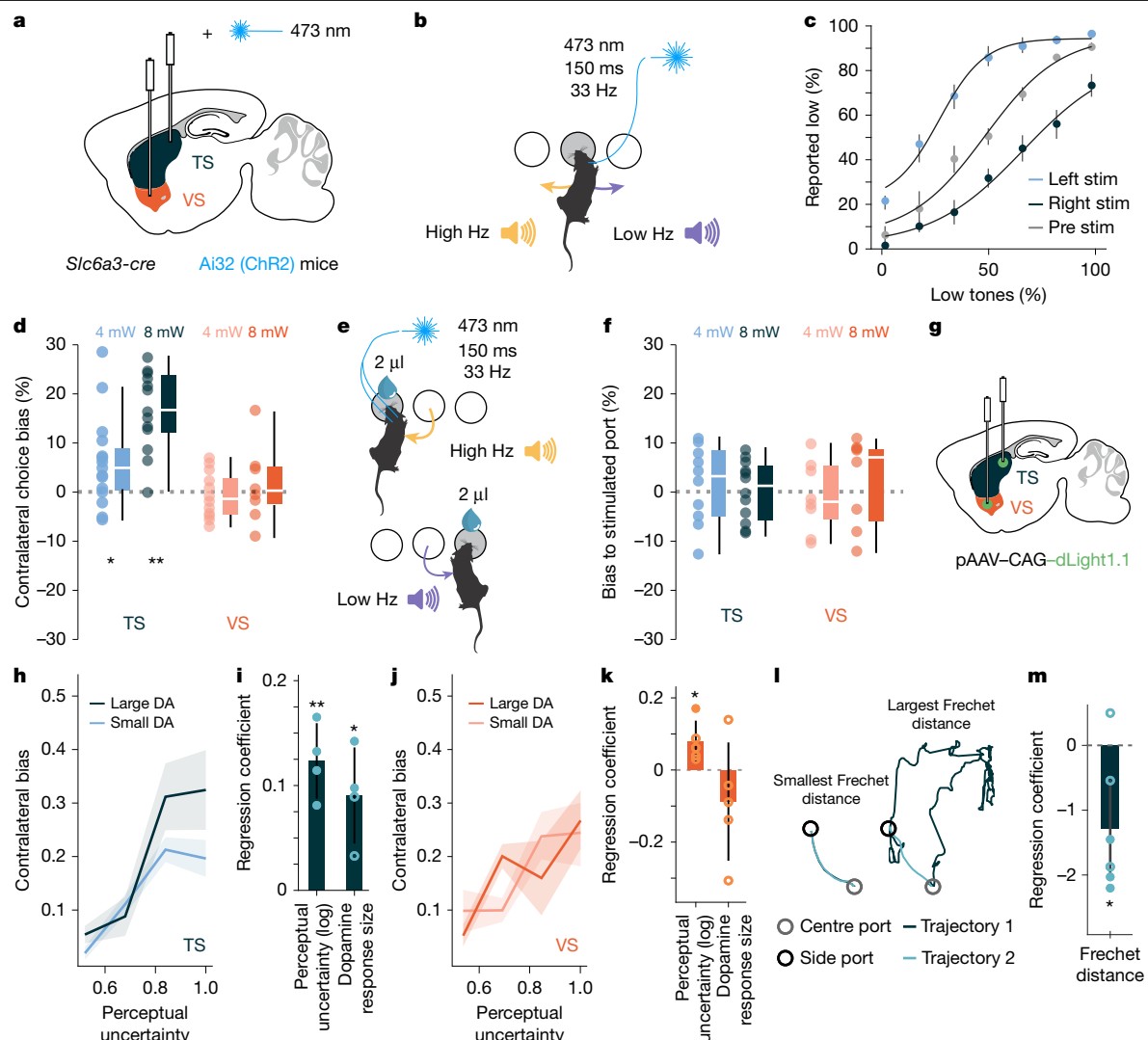

**Fig. 4 | TS dopamine release reinforces state–action associations.**
**a**, Experimental approach. *Slc6a3* encodes the dopamine transporter (DAT).
**b**, Stimulation protocol. **c**, Average performance (mean) of mice. Left hemisphere $n = 6$ mice, right hemisphere $n = 7$ mice. Pre-stimulation data are pooled across all 13 recording sessions ($n = 7$ mice, 13 hemispheres). Error bars represent s.d.
**d**, Distribution of session biases for the TS (4 mW $n = 10$ mice, $n = 15$ hemispheres, 8 mW $n = 7$ mice, $n = 13$ hemispheres) and the VS (4 mW $n = 8$ mice, $n = 11$ hemispheres, 8 mW $n = 5$ mice, $n = 8$ hemispheres) stimulations of the state–action experiment. TS: 4 mW $P = 0.018$, Cohen's $d = 1.46$; 8 mW $P < 0.001$, Cohen's $d = 2.07$. VS: 4 mW $P = 0.57$, Cohen's $d = -0.19$; 8 mW $P = 0.742$, Cohen's $d = 0.19$; Wilcoxon signed-rank test relative to zero (two-sided). **e**, Stimulation protocol. **f**, Same as **d** for the state–outcome TS-stimulation experiment (TS: 4 mW $n = 10$ mice, 8 mW $n = 6$ mice; TS: 4 mW $P = 0.625$, Cohen's $d = 0.40$; 8 mW, $P = 0.909$, Cohen's $d = 0.05$; VS: 4 mW $P = 0.84$, Cohen's $d = -0.21$; 8 mW $P = 0.461$, Cohen's $d = 0.27$; Wilcoxon signed-rank test relative to zero (two-sided)).
**g**, Experimental approach. **h**, Average change in the bias for trials preceded by small or large dopamine choice movement responses (DA) in TS ($n = 4$, mice), error bars represent 68% confidence interval. **i**, Regression coefficients from a logistic regression ($n = 4$ mice) (log uncertainty: $P = 0.006$, Cohen's $d = 3.31$; dopamine: $P = 0.03$, Cohen's $d = 2.01$; one-sample $t$-test against zero, two-sided $t$-test). Filled circles represent significant correlations. Error bars represent s.d. **j**, As **h** but for VS dopamine responses ($n = 5$ mice). Error bars represent 68% confidence interval. **k**, As **i** but for VS dopamine at time of choice. $n = 5$ mice. log uncertainty: $P = 0.03$, Cohen's $d = 1.44$; dopamine (at time of cue): $P = 0.29$, Cohen's $d = -0.54$; one-sample $t$-test against zero, two-sided $t$-test. Error bars represent s.d. **l**, Example trajectories with smallest or largest Fréchet distances. **m**, Regression coefficients. Filled circles represent significant correlations for individual mice ($n = 6$ mice). One-sample $t$-test against zero, $P = 0.03$, Cohen's $d = -1.24$. Error bars represent 95% confidence interval. In box plots, boxes represent quartiles 2 and 3, the centre line shows the median and whiskers extend to the furthest data point within 1.5 times the inter-quartile range from the box.

## TS dopamine reinforces state–action associations

To determine whether the TS dopamine signal can act as a teaching signal, we optogenetically stimulated TS dopamine release at different task epochs. To mimic the endogenous movement-related TS dopamine signal, we stimulated unilaterally at the centre choice port, in trials where there was more sensory evidence for a contralateral choice (Fig. 4a,b and Extended Data Fig. 9a,b). Within sessions, stimulation induced a significant contralateral choice bias (Fig. 4c,d and Extended Data Fig. 9c). This bias developed over the course of the session as

would be expected if it influenced learning (Extended Data Fig. 9d) and could be replicated in a model where we artificially stimulated APE (Extended Data Fig. 9e). Optogenetic stimulation did not bias action directly as there was no choice bias on individually stimulated trials (Extended Data Fig. 9f). Stimulating dopamine release in the VS at the time of choice had no significant effect (Fig. 4d), nor did stimulation of dopamine release in the TS or the VS at the time of choice outcome (Fig. 4e,f). In a free choice paradigm TS dopamine stimulation did not induce a choice bias but there was a significant bias towards the stimulated port when we stimulated dopamine release in the VS (Extended

Data Fig. 9g–j). Finally, there were no appetitive or aversive effects of the stimulation in a real-time place-preference assay when we stimulated dopamine release in the TS (Extended Data Fig. 10a,b). Together these results show that TS dopamine release can, as predicted by the computational models, reinforce state–action but not state–outcome associations.

Other theories suggest that movement-related dopamine facilitates movement initiation or modulates ongoing action[9,26,27]. However, closed-loop optogenetic stimulation of TS dopamine release in an open-field arena did not influence movement likelihood or alter movement parameters when mice did move (Extended Data Fig. 10c–i). These findings suggest TS dopamine activity reinforces state–action associations rather than influencing ongoing action.

Since TS dopamine stimulation reinforced state–action associations, we investigated whether endogenous TS dopamine release functioned similarly. A logistic regression model predicting choice repetition from the previous trial's dopamine response and current log uncertainty showed significant positive correlations for both factors (Fig. 4g–i), indicating that mice were more likely to repeat their previous choice when the TS dopamine response was larger and sensory uncertainty was higher. By contrast, dopamine reward response size in the VS did not correlate with choice bias (Fig. 4j,k). These results suggest that movement-related dopamine at choice timing serves as a value-free teaching signal, reinforcing stimulus–action associations in the TS so that mice learn to repeat the action that they have taken in the past when they hear the auditory stimulus.

If the TS dopamine response encodes an APE on the current trial, it should not only bias mice toward repeating stimulus–action associations but also influence trial kinematics, making subsequent actions more similar. To test this, we performed a linear regression between the current trial TS dopamine response at time of choice and the Fréchet distance between current and subsequent trial trajectories (which provides a measure of the similarities between the two trajectories) (Fig. 4l). The analysis revealed a significant negative correlation between TS dopamine response size and Fréchet distance, indicating that a larger TS dopamine response in a given trial led to a more similar subsequent trajectory (Fig. 4m and Extended Data Fig. 10j). These findings show that larger TS dopamine signals bias mice to repeat prior state–action associations with more similar movement trajectories.

## A dual-controller model for learning

To examine interactions between striatal regions receiving RPEs or APEs, we built a basal ganglia model that simulated learning with these prediction errors (Fig. 5a). Our dual value-based/value-free model had an actor and critic, updated by RPE, that learned to control actions and generate reward predictions. The value-free control system, updated by APE, learned to control action and generate action predictions. A model with only a value-free controller was unable to learn the task, as the model continues to repeat what it has done in the past, which is to randomly choose left or right in equal proportion (Fig. 5a). By contrast, a model with only the value-based controller was able to independently learn the task (Fig. 5b). Notably, a model with both controllers learned faster than the value-based system alone (Fig. 5b). Initially, there was no difference between the learning rate of these two models, because early learning is exclusively driven by RPE updates. However, as the dual model begins to consistently choose a particular action in response to a particular sound, APE updates started to reinforce specific stimulus–action associations, boosting performance and learning rate. This divergence between the value-based and dual-controller model recapitulates the learning deficits observed when we lesioned the TS or removed dopamine innervation to this region (Extended Data Fig. 11a). Our model shows that the value-free controller alone cannot support learning but when paired with the value-based controller, it acts to boost learning and store stable associations.

Although both the value-based and value-free controllers can control behaviour, our muscimol inactivation data (Fig. 1c) highlights that in well-trained mice, behavioural performance relies on the TS (value-free controller). Since the value-free controller cannot learn the task alone, the task must first be learned by the value-based controller and then control must transfer over time to the value-free controller. In other dual-controller models, this transfer is achieved by an external arbiter[4,6]. We examined if a similar transfer would occur if the actor's weights decayed over time when RPE was low, consistent with striatal literature[28–31]. When the value-based actor's weights decayed, there was indeed a transfer of control in our model. Initial performance was controlled by the value-based controller, and over time, the value-free controller took over (Fig. 5c,d). In the expert condition, the dual-controller model retains the predicted value of each state–outcome association in the critic and the state–action association in the value-free controller (TS) (Fig. 5c and Extended Data Fig. 11b). The differential retention of the state value information in the critic and the decay of the state–action associations in the actor, validated the selection of an actor–critic architecture for our value-based controller.

To test whether the TS also slowly takes over control of performance like the value-free controller in our model, we optogenetically inactivated either type of TS projection neuron throughout learning (Fig. 5e). Early in training optogenetic inactivation of neither type of TS-SPNs had a significant effect on behaviour (Fig. 5f). However, as training progressed the inactivation began to have a significant effect on choices and these effects grew larger as the mice became experts at the task. These behavioural effects began to consistently increase after the mice were performing above 65% (Fig. 5f,g). This is a comparable training stage to where the control and lesion groups diverged when the TS was lesioned experimentally or in the model (Extended Data Fig. 11a). Taken together, this data shows that the TS only begins to contribute to task performance and learning once mice are already preferentially choosing a specific action in response to the auditory cues.

In the COT task, frequency-specific cortico-striatal plasticity supporting appropriate sound–action associations develops in each hemisphere of the TS throughout learning[32,33]. The same appropriate sound–action associations also form in our value-free controller, showing that APEs could be used to drive the frequency-specific changes in cortical striatal plasticity as mice learn the task (Fig. 5h). Together, our behavioural results, chronic lesions, acute inactivation, dopamine recordings, optogenetic perturbations and previous synaptic plasticity results[32] are consistent with the TS forming the value-free part of a dual value-based/value-free learning system (Extended Data Fig. 11c).

## Discussion

Here we show that movement-related dopaminergic activity in the TS acts as a teaching signal to reinforce state–action associations. TS dopamine activity encodes an APE, the difference between the action taken and the predicted action in a given state. This value-free signal teaches mice to repeat past actions. Alone the value-free system (APE→TS) is not able to support reward-guided learning but in conjunction with the canonical RPE system it learns to mimic and store the value-guided state–action associations. Together we show that there are two types of dopaminergic prediction errors that work in tandem to support learning, each reinforcing different types of association in different areas of the striatum.

The identification of dopamine transients prior to movement initiation led to the idea that movement-related dopamine release may act to trigger the initiation of movement[9,11,20,26]. However, more recent experiments that used optogenetic stimulation parameters that were calibrated to mimic physiological levels of SNc dopamine release did not trigger body movement or lead to an invigoration of ongoing action[34,35]. The calibrated stimulation was however, as we observe, able to act as a teaching signal to drive conditioned place-preference learning and reinforce the use of particular behavioural syllables. This suggests that,

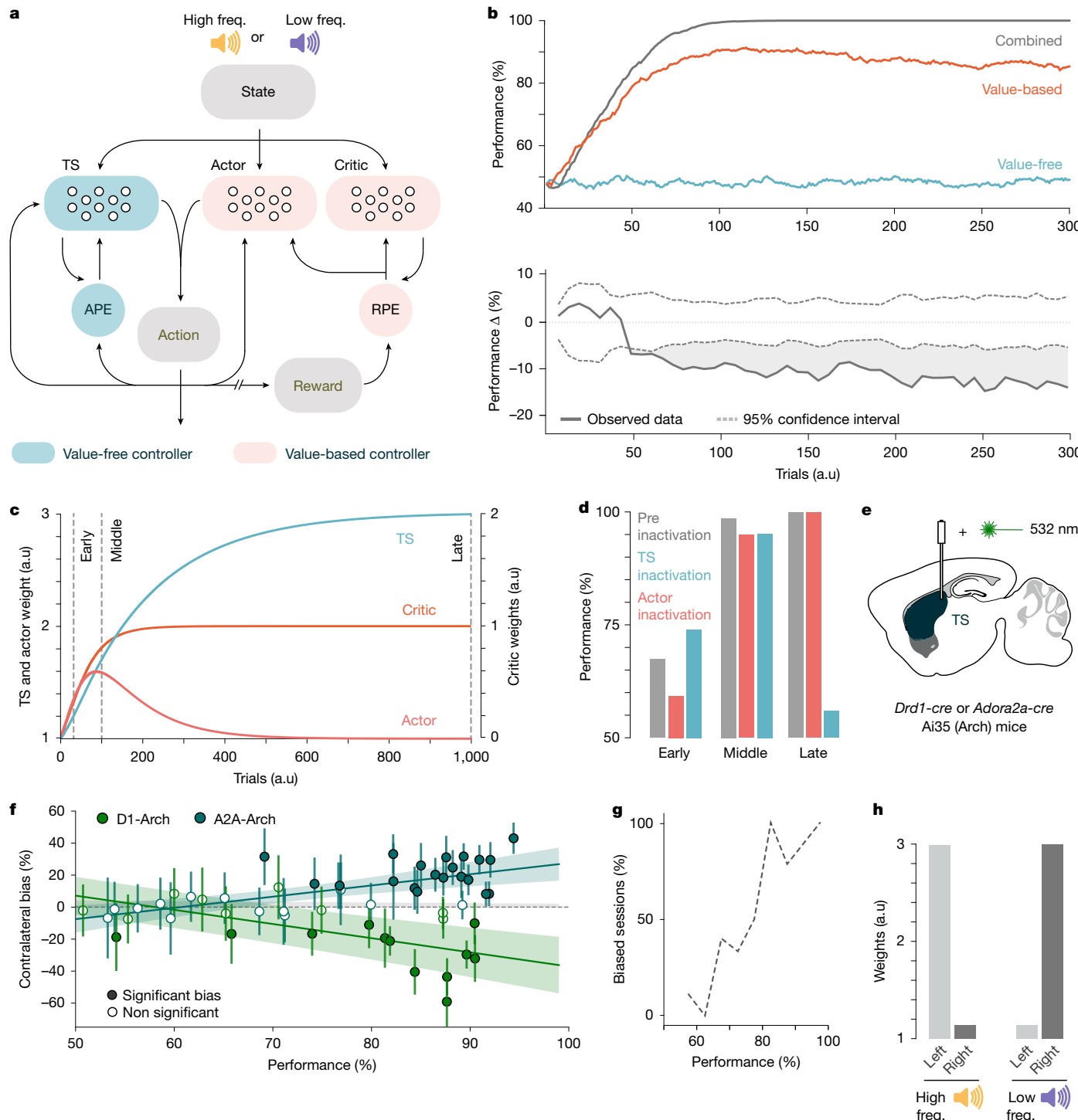

**Fig. 5 | A dual-controller model predicts the effect of experimental manipulations. a**, Schematic of the network model. **b**, Top, task performance across learning for the full dual-controller model (combined), the value-based controller, or the value-free controller. Bottom, differences in performance between the combined model and the value-based model (12 random agents selected for each). **c**, Change of the model weights for a reward association (high tone→left action), as means for 100 agents. Vertical lines indicate inactivation time points in **d**. **d**, Performance levels before and after (as the mean after ten trials) model inactivations of the TS or the actor networks. **e**, Schematic of the experimental approach for acute inhibition of D1 SPNs or D2 SPNs in the TS. **f**, Quantification of the contralateral bias on opto-stimulated trials (Methods) for each session as a function of session's performance. Error bars represent 95% confidence intervals. Lines show the mean and s.d. of linear fits for each mouse in each dataset. D1-Arch $P = 0.004$; A2A-Arch $P = 7.6 \times 10^{-4}$; Methods; $n = 8$ mice. **g**, Proportion of significantly biased sessions in **f** as a function of performance. **h**, Final weights in the TS for each sound–action association.

as we have seen in the TS and consistent with APE, movement-related dopamine activity in other striatal areas may also act as a value-free teaching signal rather than as a trigger or modulator of ongoing movement. Notably, learning rules that incorporate terms similar to APE

have been used to describe how mice learn to adjust the gain of specific kinematic features[28]. Yttri & Dudman[36] showed that the changes were dependent on the DMS, suggesting that there may be common learning rules and computations across the striatum.

This raises the possibility that movement-related dopaminergic responses throughout the striatum[9–12,20,26,37] may reflect APE. In line with this SNc movement-related dopamine activity also decreases with practice in a lever-pressing task[9], as it should if the activity reflects an APE[5]. Furthermore, dopamine transients in the dorsolateral striatum (DLS) occur when sub-second action modules are initiated even in the absence of rewards or task structure[35]. The size of these transients is proportional to how predictable the upcoming action is with respect to the action that was just taken. This pattern of activity is what would be expected if these dopamine transients reflected an APE that compared the action that is taken (or soon to be taken) to the action that was predicted given the preceding behaviour. Together, the current data on movement-related dopamine activity are consistent with APE, suggesting that all movement-related dopamine activity throughout the striatum may reflect APEs that serve as value-free teaching signals.

The caudate tail (CDt)—the primate homologue of the TS—is also innervated by an anatomically distinct population of dopamine neurons[38] that are not activated by differences in reward outcome[39]. Rather, as we observe here, these neurons are preferentially active prior to contralateral orienting eye movement or when stimuli that predict contralateral orienting responses are presented[39], as would be expected if they also encode an APE. The contralateral nature of the response in primates was interpreted as a unilaterally processed visual response[39]. Our results now show that, at least in our task, the TS dopamine responses in mice are related to contralateral movement initiation and not auditory cues. The response to movement rather than sensory cues is also consistent with TS-projecting dopaminergic neurons receiving input from the deeper motor-related layers of the superior colliculus rather than from the superficial sensory-related layers[40,41]. Whether APE signals are driven by the preparation or execution of action will need to be determined in future work and may help resolve the difference in explanation. While the interpretation of the signal was different, the proposed function of the CDt-projecting dopamine neurons is entirely consistent with what we show here, in that it serves to reinforce stable sensory–motor associations in the CDt in an outcome-independent manner[42,43]. This suggests that the TS is unique across species in that it exclusively receives a value-free movement-based teaching signal to update stable sensory–motor associations.

Other laboratories have also shown that dopaminergic activity in the TS is not activated by reward[44,45] but by threatening stimuli, such as an air puff or loud sounds. In addition, large TS dopaminergic transients are active when mice initiate avoidance behaviour, and these responses decrease as mice stop performing avoidance responses[44]. These observations led to the proposal that TS dopaminergic activity encodes a threat prediction error (TPE)[44,46]. In agreement with these results, we can also observe threat responses in the same region (and same mice) where we observe APE responses (Extended Data Fig. 12). However, whereas optogenetic stimulation of dopamine release in the TS was able to reinforce state–action associations, we could not detect any aversive effect of this stimulation even when we used higher power than was needed to influence associations in our task. Although our TS dopamine stimulation was not able to induce avoidance behaviour, it does appear clear that TS dopamine release is critical for maintaining innate aversive associations[40,44]. How TPE and APE work in the TS to support their different functions at different timescales will need to be determined in the future. One option is that threat- and movement-related activity are encoded by separate dopaminergic subpopulations. In support of this, it has recently been reported that threat and acceleration-related dopamine responses are encoded in separate genetic subpopulations that both project to the TS, expressing *Slc17a6* (also known as *Vglut2*) and *Anxa1*, respectively[37,47–49]. These populations may respectively encode TPE and APE.

APE was first conceived as a teaching signal that could update habitual value-free state–action associations[4,6]. These models offer a parsimonious account for hallmarks of habitual behaviour, such as slower adaptation to contingency degradation or reversal and an insensitivity to outcome devaluation. This is because the stable associations in these models are driven by APE so are stored in an outcome-independent manner and are therefore insensitive to changes in outcome value. Our results show that dopamine neurons that project to the TS encode this type of movement-based prediction error. Interestingly the CDt is where habitual stimulus–action associations are stored[29,39,50]. Typically for instrumental learning the posterior DMS and the DLS have been shown to be critical for goal-directed and habitual learning respectively[31,51–55]. Recently the pDMS has been shown to receive reward-related dopaminergic teaching signals[30,56]. By contrast, movement-related dopamine signals, that could reflect APEs, are prominent in the DLS[11,37]. Taken together, we propose that RPE and APEs may be used as different teaching signals to drive goal-directed (value-based) and habitual (value-free) learning in different regions of the striatum.

The value-free system updated by APEs learns to repeat the actions that have been taken in the past. This does not seem like an advantageous strategy, but this system must exist for a reason. The first thing to note is that although APE is a value-free teaching signal, this does not mean that the associations that are formed in the TS will be insensitive to value. Rather, the 'value-free' associations form through repetition of actions that were initially taken in pursuit of value. In this way, the associations that form in the TS can be thought of as storing the long-term 'value' of an association even though they were never reinforced in value-based manner per se. Sustaining these associations in an outcome-independent manner makes them more stable to short-term fluctuations. Indeed, stimulus–action associations in the CDt system are incredibly stable—once formed they are retained for >100 days in the absence of reinforcement[50]. Another advantage of the value-free system is that from a normative perspective learning from terms such as APE will bias control systems to repeat stable associations, this can help in learning to act upon regular structure in the environment and forming a compressed default policy—that is, acting the same way in response to the same regular structure irrespective of situational differences[57–59]. Overall, the value-free system could ensure that consistent associations are stored in a more stable and efficient manner that enables them to be acted upon quickly and reliably.

Another idea is that stimulus–action associations in the value-free system form a prior for Bayesian inference of action that encodes the policy that has worked in the past[6,58,59]. During inference this prior would be useful in biasing decisions when there is a high degree of uncertainty regarding the value of the action computed in the value-based controller. Finally, recent theoretical work has also predicted that if dopamine neurons encoded an APE, then it would allow the basal ganglia to implement off-policy learning algorithms[5]. Future work will be important in establishing the exact nature of the interaction between the value-based and value-free basal ganglia systems and to elucidate the contribution of APEs to habits, priors and off-policy learning.

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

# Methods

## Animals

Male and female adult mice, aged between 2 and 7 months old, from the following mouse lines were used: C57BL/6 J wild-type (Charles River), Drd1-Cre (Gensat: EY262), *Adora2a-cre* (Gensat: KG139), *Slc6a3-cre* (JAX: 006660), Ai14 (tdTomato, JAX: 007914), Ai35 (Arch-GFP, JAX: 012735) and Ai32 (channelrhodopsin-2/EYFP, JAX: 024109).

Mice were housed in HVC cages with free access to chow and water on a 12 h:12 h inverted light:dark cycle and tested during the dark phase. The ambient temperature of the rooms was kept between 20–24 °C and the humidity was maintained between 45–65%. Mice used in the COT task were water deprived. Mice had access to water during each training session, and otherwise 1 ml water per mouse was administered. Water was supplemented as needed if the weight of the mouse was below 85%. All experiments were performed in accordance with the UK Home Office regulations Animal (Scientific Procedures) Act 1986 and the Animal Welfare and Ethical Review Body (AWERB). Mice in test and control groups were littermates and randomly selected.

## Surgical procedures

**Viruses.** All viruses were made in house except for pAAV5-CAG-dLight1.1 (Addgene, 111067), which was used for the photometry recordings. dLight1.1 expression was induced by injecting 30–100 nl of pAAV5-CAG-dLight1.1 in the TS and 90 nl in the VS. Chronic lesions of the TS were achieved by injecting a 1:1 mix of AAV2/1-hSyn-Cre at $10^{14}$ viral genomes (vg) per ml and AAV2/5-EF1a-DIO-taCasp3-T2A-TEVp at $10^{14}$ vg ml$^{-1}$. The mix was diluted five times in saline buffer prior to injection. The same surgical procedure, using the virus AAV2/5-CAG-EGFP ($3 \times 10^{12}$ vg ml$^{-1}$), was used for the control group. For all chronic lesion experiments, 4 injections (30 nl each) were made in each hemisphere at 4 or 5 different depths to distribute the viruses as evenly as possible and to provide enough coverage.

**Viral injection and implant surgeries.** Mice were anaesthetized with isoflurane (0.5–2.5% in oxygen, 1 l/min), also used to maintain anaesthesia. Carpofen (5 mg kg$^{-1}$) was administered subcutaneously before the procedure. Craniotomies were made using a 1-mm dental drill (Meisinger, HP310104001001004). Coordinates are measured from the extrapolated intersection of the straight segments of the coronal sutures between the parietal and the frontal bones. This point usually lies slightly frontal to bregma and is more stereotypic than bregma itself with respect to the brain. The following coordinates were used: TS: 1.8 posterior, 3.45–3.55 lateral, 3.4–3.5 depth; VS: 1.0-1.1 anterior, 1.65 lateral, 4.4–4.5 depth; DMS: 0.5 posterior, 1.45 lateral, −2.0 depth. Viral injections were delivered using pulled glass pipettes (Drummond, 3.5 inch) in an injection system coupled to a hydraulic micromanipulator (Drummond, Nanoject III) on a stereotaxic frame (Leica, Angle Two), at approximately 10 nl min$^{-1}$. For optogenetic experiments, we used flat optical fibres of 200 μm diameter (Newdoon: FOC-C-200-1.25-0.37-7) and tapered optical fibres (Optogenix, Lambda Fiber Stubs) of 200 μm diameter with 1.5 mm active length, 4 mm implant length. For photometry experiments, flat fibres (Doric Lenses 0.57NA, 200 or 400 μm diameter, 4 to 7 mm long) were implanted vertically, 0.1 to 0.15 mm dorsal of the injection site of dLight1.1. Eight mice that were targeted for TS recordings were not used in any experiments as initial investigation revealed they had reward responses which have not been observed in the TS[44,45]. To confirm these fibres were outside the TS in six out of eight of these mice we performed serial two-photon microscopy and confirmed that in all cases the fibres were located outside of the TS. For chronic D-AP5 cannulation experiments, we used 26-gauge guide cannulas (Plastics One, C200GS-5), that were cut to 5 mm below the pedestal. We implanted the guide cannulas in the TS (coordinates: 1.82 posterior, 3.55 lateral, 3.0 depth) and then fit them with a dummy cannula with no projection (Plastics One, C200DCS). All implants were affixed using light-cured dental cement (3 m Espe Relyx U200), which was also used to attach a headbar. In mice that received only injections the wound was either sutured (6-0, Vicryl Rapide) or glued (Vetbond). For photometry experiments a subset of mice had fibres implanted in both the TS and the VS to compare how the signals in these regions developed throughout learning.

**Pharmacological manipulations.** For muscimol injections, bilateral cranial openings were performed over the DMS and the TS, and a headbar was positioned stereotactically. A landmark that aligned to bregma allowed for future injections in stereotactic positions. Closures of cranial openings were prevented by covering the exposed brain with Duragel (Cambridge Neurotech). Skull was covered with Kwiksil (World Precision Instruments), which was removed before every injection. Before each training session, mice were head-fixed while awake and 30 nl of either muscimol (Sigma-Aldrich) at 0.2 mg ml$^{-1}$ or saline were injected for experimental and control sessions, respectively, co-injected with cholera toxin B (Cambridge bioscience) of different colours to trace injection sites during histology. After a 15-min period, mice started the training session.

For chronic D-AP5 infusion experiments, we head-fixed the mice for 6 min and then habituated them in the training boxes for the first two days using the protocol described above. Starting from day 3, we administered either saline or 20 mM D-AP5 (Tocris, 0106) bilaterally in the TS using internal cannulas (Plastics One, C200IS) with 0.6 mm projection using a custom-built set up. The mice were awake and head-fixed during the infusion and received an infusion of 500 nl of 20 mM D-AP5 per hemisphere in 3 min, which was followed by a 3 min wait time before internal cannulas were retracted to be replaced by dummy cannulas again. The mice were then immediately put in the behavioural boxes to train in the auditory protocol. After the mice completed ≥3,000 trials, we administered saline in both groups in the following session. For acute D-AP5 infusion experiments, we let these mice train without infusion till they reached expert-level performance. We then infused either saline and D-AP5 in different sessions using the same method described above and measured their performance in the perceptual auditory two-alternative forced choice task.

For dopamine neuron ablations we followed the protocol in ref. 44. In brief, we first injected intraperitoneally (10 mg kg$^{-1}$), a solution made of 28.5 mg desipramine (Sigma-Aldrich, D3900-1G), 6.2 mg pargyline (Sigma-Aldrich, P8013-500MG), 10 ml water and NaOH to pH 7.4. Subsequently, mice underwent surgery, and a solution of 10 mg ml$^{-1}$ 6-hydroxydopamine (6-OHDA; Sigma-Aldrich, H116-5MG), dissolved in saline buffer was injected. Once prepared, we kept the solution on ice covered from light and injected it within 3 h. If the solution turned brown, it was discarded. For controls we only injected saline buffer. For all manipulation experiments cage mates were assigned to either control or manipulation groups prior to surgery, this was not done through an explicit randomization procedure. Power calculations were not used to define the size of the groups prior to experiments. Experimenters were not blinded to whether mice were in the control or manipulation groups.

**Transcardial perfusions.** Mice received a lethal intraperitoneal injection of pentobarbital (0.1 ml per 10 g) inducing unconsciousness and death. Once unresponsive, mice were first perfused with phosphate-buffered saline (PBS) followed by 4% paraformaldehyde (PFA). The mice were decapitated, and the brains were extracted and fixed in 4% PFA for 24 to 48 h at 4 °C. Subsequently, brains were stored in PBS until further processing.

## Behavioural procedures

**Training box.** Mouse training was carried out in a sound-isolated box containing the behavioural chamber, which consists of a closed arena of 19 cm × 15 cm with 3 ports located in a wall (1.95 cm from the floor,

each port 3 cm from the next). To better separate cues and actions and to increase variability in movement times, the three ports were spaced further apart for photometry experiments. The centre port was located in the middle of one of the long sides of the arena and the choice ports were located in the centre of the shorter sides. The ports are equipped with LEDs so they can be illuminated, and photovoltaic infrared sensors to detect poke events. Water can be delivered through each of the side ports. Ports were purchased from Sanworks or constructed in house to the same standards. Controlling chips were purchased from Sanworks. The chamber was illuminated exclusively with infrared light. Mice were monitored through standard webcams without the infrared filter. Sounds were delivered through one central speaker (DigiKey: HPD-40N16PET00-32-ND) and amplifiers (DigiKey: 668-1621-ND). Bpod (Sanworks) was used to control the state machine, and the software was run using Matlab.

**COT task.** The COT task is a self-initiated two-alternative-choice paradigm. For behavioural experiments the possibility to start a new trial is signalled by turning on the centre port LED. This LED cue was not present for mice that were used for photometry. Mice start a trial by poking in the centre port and holding their position for 100 to 300 ms. The centre port LED is turned off after that required time has elapsed. In between 0 and 50 ms after poking in the centre port, sound is triggered, lasting for 500 ms. Following poking out from the centre port, mice were trained to poke in either of the two side ports. Two microlitres of water was delivered in only one of the two ports, contingent on the stimulus.

**Training procedure.** Depending on the training stage, pokes in the wrong port abort the trial. During the first day of training, mice were habituated to the box and the poking sequence by doing a visual version of the task in which both side ports delivered water in every trial. Water amounts were decreased from 5 µl to 2 µl during the first 3 days of training. All photometry recordings started after the initial habituation day and after the reward amount was decreased, but before the performance of the mice improved beyond chance.

For the simple version of the task (not the psychometric), we included an anti-bias protocol to force the mouse to sample from the two ports. The protocol samples every ten trials for the proportion of error trials and calculates the percentages of port choices on those trials, adjusting the target port on the next ten trials to overcome any potential bias, proportionally to the errors and the bias. Therefore, this protocol engages progressively more as the mouse becomes more biased and disengages if the mouse becomes unbiased. This protocol was only active during the initial phases of learning as it uses the proportion of error trials for engagement and calibration. All the experiments using the psychometric version of the task were performed once mice were experts (>85% performance on the simple task). The simple version of the task delivered only the easiest possible trial types, with 98% of tones being from one of the two octaves. For the psychometric experiments, seven equally spaced ratios of high or low tones were used. From the beginning of training, the trial types were randomly interleaved, affected by the anti-bias protocol described above.

**Variations of COT task in photometry experiments.** Several variations of the COT task were used during photometry recordings. In the outcome value change experiment, unexpected large rewards (6 µl) and omissions were introduced randomly on both sides with a probability of 0.1 each. Mice experience multiple sessions of this protocol to get enough trials of each type.

In the predicted value change experiment, the value of one port was changed 100 trials into a session. This port delivered 6 µl rather than 2 µl, whilst the other port still delivered 2 µl. Mice did multiple sessions of this protocol and experienced both sides becoming the large reward size.

The state change experiment was performed only once per mouse. 150 trials into a training session, the sound that had previously corresponded to the contralateral choice to the recording fibre was changed for a white noise stimulus. This new white noise stimulus overlapped with the original COT frequencies. This white noise was louder than the background white noise and was clearly detectable. The mice were required to make a contralateral response to the white noise to receive a reward.

To test whether the dopamine signals were dependent on an auditory stimulus being present, the sound indicating contralateral turns was omitted in expert mice (silence trials).

In the sound-on-return task, the sound indicating a contralateral turn was played during the return from either side port instead of at the centre port. The sound lasted until the mouse returned and poked into the centre port or at most for 2.5 s. If the mouse did not return to the centre port within 5 s, the subsequent trial would be a classic COT task trial with the sound being played upon poke in into the centre port.

The response to passive sounds was always tested at the end of a classic COT or a sound on return session. Mice were left inside the training box for an additional 10 min while all three ports were covered with custom made lids. During that time both high and low frequency sounds or white noise lasting 500 ms were presented at random time intervals (mean interval 5 s) and in a random sequence.

**Sound stimuli.** Sounds consist of a stream of 30 ms pure tones presented at 100 Hz (each tone was introduced 10 ms after the previous one). One of two octaves (5 to 10 kHz, or 20–40 kHz) was selected as the target octave to indicate the side port where water was available on that trial. Each 30 ms tone was randomly drawn from 16 logarithmically spaced frequencies on each octave. The difficulty of the trial was controlled by varying the proportion of 30 ms tones from each octave. For example, in a trial catalogued as 82% of high tones, for each 30 ms tone there are 82% chances of playing a tone from the 20–40 kHz octave, and 18% (100% − 82%) chances of playing a tone from the 5–10 kHz octave. These probabilities are independently computed, so two tones from different octaves can sound simultaneously. The overall amplitude of the sound is randomly selected between 60 and 80 dB. The amplitude of the sound during the passive exposure experiments was also 60–80 dB and the sound duration was 0.5 s.

**Water delivery.** Water was delivered into the two side ports for correct choices using a solenoid valve that was carefully calibrated. As previous studies have suggested that the neurons in the TS are responsive to valve click noises[45], for the photometry experiments we placed valves outside the training boxes and muffled them using sound insulation foam. In addition, we played quiet white noise constantly inside the training box. We confirmed with a microphone that the valve clicks could not be detected with these precautions.

**Photometry and video acquisition in the COT task.** Fluorophores were stimulated using 465 nm and 405 nm LEDs (Thorlabs) of max power 0.2 mW. The 465 nm and 405 nm LED amplitudes were modulated using a sinusoid of 211 and 531 Hz respectively. The 405 nm light falls at the isosbestic point for fluorophores of the type used in this study[60]. This enabled separation of signal from movement artefacts and bleaching as done by ref. 61. Light was passed from the LED source through optical fibres with NA 0.57, through a commutator (Doric Lenses, FRJ 1 × 1 PT 0.15) to a patch cord (Doric Lenses, FC-ZF1.25 LAF). A mini cube and photodetector (Doric Lenses) were used to collect the signal. The signal was then passed to a NIDAQ (National Instruments) and recorded and analysed using custom Python scripts as described in 'Statistical analysis'. Behaviour was controlled with a Bpod which sent TTL pulses to the NIDAQ at the start of each trial.

Mice were filmed from above during training and recording sessions using a Basler acA640-750um USB 3.0 camera. Videos were acquired at 30 Hz and synchronized with the photometry acquisition using the

NIDAQ. For ease of analysis, mice were always trained in the same box and cameras were never moved.

**Open-field stimulations.** For this experiment we used a 40 cm × 40 cm square arena, and experiments were conducted with ambient light. Sessions lasted for 30 min. See 'Optogenetic manipulations' for specifics about the triggered stimulations.

**Open-field photometry recordings.** For this experiment we used a 50 cm × 20 cm × 28 cm (*L* × *W* × *H*) arena. Mice were allowed to explore the setup for 20 min, during which video footage and photometry recordings were acquired.

**Optogenetic manipulations.** For opto-inhibition of either the D1 or the D2 SPNs, in 15–25% of trials, randomly selected, a sustained pulse of green (532 nm) light was delivered after mice initiated the trial, always preceding the onset of sound delivery by at least 50 ms and lasting longer than the sound duration. Light intensity was calibrated to 12 mW at the fibre tip.

For dopamine opto-excitation, during the COT task, the first 150 trials of each session were done without stimulation to get a baseline behaviour, and subsequently, stimulated trials were introduced with these parameters: blue (473 nm) light delivered starting at the time of poking and lasting 150 ms in 5 ms pulses at 33 Hz of 4 or 8 mW intensity measured at the fibre tip. For the state–action experiment, stimulation was delivered unilaterally in the centre port for trials in which the state predicted a movement contralateral to the stimulated hemisphere. For the state–outcome experiment, stimulation was delivered bilaterally on one of the two side ports (for the whole duration of the session) every time the mice chose that port (correct and incorrect trials), coupled with water during correct trials. No anti-bias protocol was employed during these experiments. To test for effects on movement initiation, we performed an experiment similar to the one in ref. 9. We placed mice in an open field where mice received a blue (473 nm) light stimulation (5 ms pulses at 33 Hz during 500 ms of 4 or 8 mW intensity measured at the fibre tip) if they were immobile for at least one second. For each of these events, there was a 50% chance of triggering the laser. Trials in which light was not delivered were used as within-animal control.

**Threat experiments.** All mice used in the threat experiment were individually housed in the three to four weeks preceding the experiment[62]. Mice were placed in an arena (50 cm × 20 cm × 28 cm, *L* × *W* × *H*) with a white opaque floor allowing reliable tracking of the dark coated mice. At one end, the arena contained a rectangular shelter (10 cm × 20 cm) made of red Perspex. The other end of the arena constituted the threat zone (20 cm × 20 cm), in which visual stimuli were presented on a computer monitor (51 cm × 33 cm) that was centred above the arena at a height of 30 cm. IR LEDs provided diffuse illumination of the arena. Mice were allowed to explore the entire arena including the shelter for at least 7 min before any looming stimuli were triggered upon their next entry into the threat zone. Visual stimuli were generated using PsychToolBox and MATLAB. A single looming stimulus consisted of five high contrast expanding spots, which expanded linearly from 3°–50° over 0.2 s (235° s⁻¹) and remained at maximum radius for 0.25 s. The inter-spot-interval was 0.4 s. Each mouse was presented with three looming stimuli in one session. All looming stimuli trials were manually triggered for these mice. Minimum inter-trial interval was 90 s.

**Photometry and video acquisition in the threat experiment.** Data acquisition was controlled using custom scripts in MATLAB or Python and a NIDAQ (National Instruments). Fluorophores were stimulated as described above. Videos were acquired at 30 frames per second using an IR sensitive camera (Basler acA460-750 um USB 3.0) positioned 70 cm away from the arena and 70 cm above its floor. Frame acquisition was triggered using a NIDAQ generated TTL that was also recorded and used for post-hoc synchronization. Real-time stimulus presentation onsets were determined post-hoc using a photodiode (Thorlabs APD430C) and the TTL trigger acquired at 10 kHz.

## Imaging

**Whole-brain imaging.** We imaged the fixed brains using serial section[63] two-photon[64] microscopy. Our microscope was controlled by ScanImage Basic (Vidrio Technologies) using BakingTray, a custom software wrapper for setting up the imaging parameters (https://github.com/SainsburyWellcomeCentre/BakingTray, https://doi.org/10.5281/zenodo.3631609). Images were assembled using StitchIt (https://github.com/SainsburyWellcomeCentre/StitchIt, https://zenodo.org/badge/latestdoi/57851444). The 3D coordinates of the injections and fibre placements were determined by aligning the brains to the Allen Reference Atlas–Mouse Brain (available from https://atlas.brain-map.org) using brainreg[65] and visualized using custom functions and brainrender[66].

**Immunohistochemistry.** Brain slices were all stained following the same procedure: Blocking was performed in staining solution (PBS + 1% BSA + 0.5% Triton X-100) for 15 min. Primary antibodies (1:1,000 in staining solution) were incubated for 2–4 h at room temperature or overnight at 4 °C with rocking. Washes were performed for 15 min with a staining solution. Secondary antibodies (1:500 in staining solution) and DAPI were incubated for 2 h at room temperature while rocking. Slices were then washed in PBS and mounted using Mount Medium. Primary antibodies used were NeuN (abcam, ab104225), tyrosine hydroxylase (TH) (Sigma-Aldrich, AB152) and GFP (Aves labs, GFP-1020) (to reveal dLight-expressing cells). Secondary antibodies used were Alexa-488 anti-mouse (Invitrogen, AB_2534069), Alexa-567 anti-chicken (Invitrogen, AB_2535858), and Alexa-647 anti-rabbit (Invitrogen, AB_2535813).

**Quantification of chronic lesions.** Brains were sliced using a cryotome at a thickness of 30 μm. Fifteen to 20 slices covering the entire striatum at regular intervals were selected for NeuN staining. Slices were mounted in standard glass slides using standard mounting medium and subsequently imaged in the Slide Scanner (Zeiss) using a 20× objective. Individual slices were registered to the Allen Reference Atlas–Mouse Brain (https://atlas.brain-map.org) using ABBA (https://github.com/BIOP/ijp-imagetoatlas), and the NeuN channel was thresholded automatically, per slice, based on the intensity levels in the cortex. The coverage of NeuN staining in the striatum was determined for each slice, and the inverse was determined as lesioned area.

**Quantification of dopamine neuron ablation.** Brains were sectioned and mounted as described for the chronic lesions but stained for TH to specifically label dopamine cell processes. Slices were imaged using the Axio Scan (Zeiss). Manual regions of interest were drawn for the striatum, the cortex, and the background in each slice. For each slice, the mean intensity in the striatum was normalized to the cortex following background subtraction, and the relative intensity between the striatum and cortex was calculated. For the analysis of the correlation with the performance, data was normalized within each mouse (posterior striatum ratio/anterior striatum ratio). One mouse was removed from the analysis owing to lack of ablation (except in Extended Data Fig. 3j).

## Statistical analysis

**Behavioural data pre-processing for learning rate experiments.** Sessions with less than 60 trials were omitted, the first 5 trials of each session were discarded, and trials in which the mouse was not engaged (defined as having an inter-trial interval longer than 3 times the median value of that session) were not considered for analysis. Together, this amounts to less than 2% of data discarded. The remainder of the trials were ordered chronologically. Mice that did not learn the task (end performance less than 55%) were discarded from the analysis. This amounts to a total of three mice in the whole study.

**Psychometric fitting.** The LogisticRegressionCV from scikit-learn package in Python was used to fit the data from the psychometric version of the task. This was only used for visualization purposes.

**Learning rate experiments.** The first 5,000 trials for each mouse were used for analysis. To calculate individual learning parameters, per mouse, we modelled the performance of every mouse using a modified Weibull function[67,68]: performance $= 50 + a\left(1 - 2^{\left(\frac{-\text{trials}}{l}\right)^s}\right)$. The maximum performance was defined as the maximum of the median of the trials, binned using a window of size 200. Parameters were fitted using the scipy package in Python (optimize.minimize function). Statistical differences between the groups for these parameters were calculated using the non-parametric test Kruskal–Wallis from scipy.stats.kruskal. Significance of the behavioural correlation with the lesion size was performed using the scipy.stats.linregress function. To calculate the differences in performance at different times in learning, we first removed those trials in which mice were extremely biased towards one of the two ports. Bias was determined as described in 'Behavioural procedures', and extremely biased trials were defined as those having a value larger than twice the standard deviation for the whole dataset. This correction was not applied to calculate the significance of the observed differences. Two mice (one experimental and one control) were removed for the chronic lesions experiment as they did not learn the task at all, and we suspected that they were deaf. Additionally, the two last sessions of one control and one experimental mouse were removed as the performance dropped to chance. One experimental mouse was excluded from the dopamine cell ablation experiment as the lesion quantification showed no ablation (this mouse performance was comparable to controls). At each point in training, performance was defined as the performance of the past 100 trials. To assess the significance of the differences between the two groups, the data were binned using a window of 100 trials, and the differences between the means of each group were calculated. To generate a shuffled dataset, experimental labels (for example, lesion or control) were randomly assigned to each mouse, always maintaining the proportion of labels on the original dataset, and differences between groups were calculated the same way. We did this 10,000 times. The same procedure was used to analyse the data from the dual-controller model comparisons. The global significance of the dataset was assessed as the likelihood of the cohorts being different at any time, in comparison to the shuffled groups. Mixed ANOVA was used from the pingouin package.

**Opto-inhibition experiments.** Each individual session contains a few hundred trials, seven trial types, and between 15 and 25% of stimulated trials. A session of 300 trials can have as few as 6 ($300 \times 1/7 \times 0.15$) stimulated trials for a particular type. This can generate a large variability when calculating the proportion of binary choices, as each individual trial will have a large influence (17% in our example). To assess the significance of the biases caused by the optogenetic manipulations, we generated, for every session, a baseline distribution (1,000 shuffles) of port choice proportions for every unstimulated trial type (proportion of high versus low tones), using the same number of trials that were stimulated for that trial type. This generated the natural variability in the potential choices for each trial type and was used to assess the significance of the biases for individual sessions. The total bias for each session was defined as the average difference between opto-stimulated and unstimulated trials for all trial types. Only sessions with more than 150 trials in total were selected for analysis. For mice in which more than data for more than one session and stimulation type was available, we selected the one with the best performance, which is a good indicator of how well a mouse was doing on the task and we reasoned would offer the most stable control to compare the effect of the stimulation to. This resulted in only 3 sessions removed from the dataset, from a total of 30. Including all sessions or changing the session selection method did not alter the significance of the results. The statistical significance for each group was calculated using the non-parametric test Kruskal–Wallis from scipy.stats.kruskal, comparing the observed biased values of every session against a randomly-sampled counterpart, per session, using the variance described above. Only sessions in which the mice did the psychometric version of the task were included in this analysis.

**Dopamine photometry pre-processing.** The raw data were demodulated offline using custom Python scripts to produce traces that corresponded to the signal (465 nm) and background (405 nm) channels[61]. Then the data were processed according to the methods described in ref. 69. In brief, the demodulated traces were denoised using a median filter and a low-pass Butterworth filter (10 Hz cut off). The resulting signal and background channel traces were high-pass filtered at 0.001 Hz to correct for photo-bleaching. To correct for motion artefacts, the background channel data were fitted to the signal channel data using a linear regression. The proportion of signal that was explained by the background channel was then subtracted from the signal channel component, such that only the signal specific to the 465 nm excitation frequency remained. Finally, d$F/F$ was calculated by dividing this signal by the baseline fluorescence (the signal channel trace filtered using a low-pass filter with a cut off at 0.001 Hz). All traces were $z$-scored to allow better comparison across mice and sessions.

Peak dopamine responses were calculated using the Python package PeakUtils or by taking the maximum value of the trace if no peak was found in a given window. For cue responses, the window was between the cue onset and entry into the choice port. For action (APE) responses, the window was between the time of exiting the centre port and entry into the choice port. For outcome responses (used in the outcome value manipulation experiment) the window was between the time entry into the choice port and 200 ms later. As VS dopamine clearly dips to omitted rewards, instead of calculating peaks, the mean of the dopamine response within this window was used as an estimate of the response.

**Kernel regression model of photometry signal.** Following ref. 12, we built a linear regression model to predict the photometry signal at each time point from the behavioural events around this time. In this model, the predicted dLight response is calculated as the convolution of a time series $a,b$, etc., representing different behavioural events as series of 0 s and 1 s, where 1 s represent the occurrence of the behavioural event. This means that at time $t$, the predicted dLight response $g(t)$ is given by the weighted sum of the different behavioural events shifted in time within a set window. The model can be expressed as follows:

$$g(t) = g_0 + \sum_{t'=-\tau_a^-}^{\tau_a^+} a(t-t')k^a(t') + \sum_{t'=-\tau_b^-}^{\tau_b^+} b(t-t')k^b(t') + \ldots + \text{error}$$

where $[\tau_x^-, \tau_x^+]$ gives the shifted time window in which behavioural event $x$ is allowed to influence the predicted photometry signal. $k^a$, $k^b$, etc. are the kernels for the behavioural events, or equivalently, the linear regression weights for the events at each different time shift. When plotted as a time series, these regression weights form the estimated response profile for the dLight signal due to a given behavioural event. These regression coefficients were estimated using the linear regression function of the Python package scikit-learn.

Behavioural events were selected from trials in which the mouse did not repeat events (for example, did not repeatedly poke their head in the centre port). The model was fitted for each mouse for each session as the signal in the TS and VS evolved over learning. As the choice movement initiation time is unclear and may start prior to the withdrawal of the head from the centre port is detected, the movement kernels were allowed to extend 0.5 s prior to the event. As the movement duration was longer than the cue and outcome responses, the movement kernel

was allowed to extend 1.5 s after the event, whereas the cue and outcome kernel windows were limited to 1 s after the event.

**Calculation of explained variance by behavioural variable in kernel regression model.** The calculation of the percentage variance explained by the full model was performed per session per mouse and then averaged across sessions. To calculate the percentage variance explained by each regressor, the predicted dLight signal was recalculated without that particular regressor, inspired by ref. 70. The explained variance of the new prediction compared to the true signal was then calculated. Finally, the percentage variance explained by the removed regressor was calculated by comparing the explained variance of the full model, $v_{full}$, to the explained variance when that regressor is removed from the prediction calculation $v_{partial}$, using the following equation:

$$\frac{v_{full} - v_{partial}}{v_{full}} \times 100$$

Although this method does give a decent comparative measure of the contribution of each regressor to the explained variance of the data by the prediction, it can result in percentages larger than 100 if the predicted dLight signal without that regressor performs worse than the intercept at explaining the data. This can be seen for some of the VS recordings without the outcome regressor.

In Fig. 2g the full photometry trace was used to estimate the percentage variance explained, both for the full model and the individual regressors. This includes extended periods with no behavioural events as the task is self-paced. This may lead to an underestimate of the percentage of the photometry signal that is explained by the task. To account for this the percentage variance explained was recalculated solely on the portions of the photometry trace for which there were behavioural events, a process we refer to as 'trimming'. This trimming was only done in Extended Data Fig. 4m.

Additionally, we explored including return to centre movements as behavioural events for the kernel regression model. These events were taken from the time of the mouse leaving the choice port and taking a 'direct' route back to the centre port. To assess how direct the path taken by the mouse to the centre port was, we used the tracking data (see below) and computed the cosine similarity between the optimal path from the side port to the centre and the path taken by the mouse. Return vectors with a cosine similarity ≥0.9 to the optimal vector (within the first 10 s of leaving the choice port, or when the mouse entered the centre port) were included in Extended Data Fig. 4j,k and were considered for inclusion in the regression. Return event kernels had a window from 0.2 s prior to and 1.5 s after leaving the side port. This was chosen to be conservative so as to avoid crossover with the next trial/prior choice movement and to be comparable to choice movements within the task, which had a mean duration of 0.68 ± 0.53 s (ipsi) and 0.68 ± −0.61 s (contra), for which the kernel window also extended 1.5 s after leaving the port.

For ease of visualization and alignment, only short return movements (≤1 s) were included for the averages in Extended Data Fig. 4j,k.

**Regression predicting current trial dopamine from past choices.** We performed a linear regression (using statsmodels.api.OLS) predicting the size of the dopamine response (TS dopamine at time of choice, VS dopamine at time of cue) on correct contralateral trials from previous choices for the same stimulus. We included data from throughout training for this analysis. A positive regression coefficient means there is a larger dopamine signal on the current trial when the side chosen on the current trial was chosen in response to the same stimulus in the past (how far in the past is given by the $x$ axis). A negative regression coefficient means that the dopamine response is smaller when the chosen side had been chosen in response to the same stimulus in the past. Separate regressions were performed for each 'lag' (number of trials back value).

**Video tracking during photometry recordings and quantification of movement parameters.** The position of the mouse was tracked using DeepLabCut[71] and variables such as speed, acceleration, angular velocity and angular acceleration were calculated using custom scripts in Python. As the videos for the freely moving experiment were taken at a slight angle, the coordinates were transformed into standard space using custom Python scripts. Videos taken during the COT task were taken from above so there was no need for such a transform. Speed and acceleration were calculated using the nose as a marker. For angular velocity and acceleration, a triangle was formed using the nose and the two ears and a line was drawn from nose to the line between the two ear markers. This line from the nose to the back of the head was taken as the heading direction of the mouse. Turn angles were calculated using the cumulative angular velocity. In the task, 0° was defined as the angle when the mouse leaves the centre port. To calculate the maximum turn angle in the task, a sigmoid curve was fitted to the cumulative angular velocity for each trial between the sound onset and entering the choice port. The upper plateau of the sigmoid (maximum fitted cumulative angular velocity) was then taken to be the turn angle for the trial.

To determine the coarse relationship between the turn angle and the size of the dopamine response, trials were divided into four quantiles based on the size of the dopamine response between the mouse leaving the centre port and entering the choice port for contralateral choices. Turn angles were divided into using the quantiles that were created from the size of the dopamine response. A regression slope was fitted (using the function stats.linregress from the Python package SciPy) between the average quantile turn angle and average quantile dopamine response for each mouse for each session. The fit slope was then compared to the fit slope if the $x$ and $y$ quantile labels were shuffled in. The data were shuffled 100 times and the distribution of the fit slopes from the actual data were compared to the shuffled distribution.

In the freely moving experiment, turns were defined as the moment when the angular velocity crossed a threshold of ±0.5 s.d. Turns were required to last at least 0.06 s and were only included for analysis if no other turns occurred in the preceding or subsequent 0.5 s. The head angle at the times of turn onset was then used to define 0° turn angle and turn angles were determined using the cumulative angular velocity from this time point.

**Regression of speed, turn angle and trial number.** To investigate whether changes in turn angle and speed could account for the decrease in TS dopamine response size with trial number seen in Fig. 3, for each mouse we built a linear regression model predicting dopamine response size for all trials from speed and turn angle. The resulting predicted signal was subtracted from the actual signal per trial, before regressing the residuals against log trial number. To calculate the relative contributions of speed, turn angle and trial number, we also built a regression model with all three predictors to estimate the respective variance explained.

**Dopamine opto-stimulation experiments.** To quantify the bias for each session, we calculated the choice differences between the first 150 trials (pre-stimulation), and the subsequent trials, as long as there were more than 100 trials. Sessions in which the initial bias of the mouse to either of the ports was larger than 2:1 were removed from the analysis. We averaged the biases if several sessions existed for the same mouse on the same stimulation conditions, so that we had only one observation per condition. We used the non-parametric two-sided Wilcoxon sign-rank statistic (scipy.stats.wilcoxon) to calculate the significance of the observed biases.

For the open-field stimulation, we calculated the speed of the mouse as the difference between the position of the mouse in each frame, convolved with a kernel of size 5 for smoothing purposes. The overall threshold to consider that the mouse was moving was determined

empirically, but the results presented were invariant across a large range of values tested. Instant speed was calculated as the average speed from movement initiation to 300 ms after. Average movement was calculated from stimulation to 5 s later. We used the *t*-test on two related samples (as above) to assess the significance of every parameter.

**Analysis of the effect of a large dopamine response on subsequent trial bias.** Data analysis was inspired by ref. 72. Dopamine responses considered in this analysis were reward responses for the VS and movement-aligned responses for the TS. Dopamine responses were categorized as large or small based on whether they were larger than or smaller than the 65th percentile respectively. The average change in percentage of contralateral choices was calculated for large and small dopamine responses on the preceding trial for each level of sensory uncertainty on the current trial (which correspond to the different percentage mixtures of tone clouds described in 'Behavioural procedures'). A logistic regression was performed (using statsmodels.api. Logit) to investigate the relationship between dopamine response size on the previous trial, perceptual uncertainty and the choice (repeat or switch) on the current trial.

For this analysis, we used data from the psychometric version of the task. Mice were required to meet certain behavioural criteria in order for their data to be included: the slope of the psychometric curve at the Point of Subjective Equality to measure sensitivity and bias at the edges (the 'easy' trials). Bias was calculated by taking the difference in percentage correct at the two easy trial types. Mice were required to have a slope ≥1 and a bias ≤0.09 to be included, acting as a measure how well the mice had learned the task. This ensured that all mice in the analysis had similar behaviour, with little bias and strong discrimination between stimuli.

For this analysis, following Lak et al.[72], only trials where mice made a correct choice were included. Note that, as the tail dopamine response is taken between movement onset and reward delivery, for the ambiguous trials (50% high and low) the mouse would have no information as to the outcome of their choice at this time point. Therefore, all trials of this type were included.

For the VS mice, the dopamine reward responses were used so only correct trials could be used. For the TS, correct trials were used for all stimuli other than the ambiguous stimulus. For the ambiguous stimulus, both correct and incorrect trials were used. The TS dopamine responses are taken aligned to movement, prior to reward being delivered.

**Regression predicting movement similarity based on TS dopamine size.** We performed a linear regression (using statsmodels.api.OLS) between the current trial TS dopamine response at time of choice and the Fréchet distance[73] between trajectories on the current and subsequent trial (which provides a measure of the similarities between the two trajectories that takes into account the location and ordering of the points along a curve, with a low value indicating high similarity between the two trajectories). Fréchet distances were computed recursively.

**Opto-inhibition during learning.** For the experiment in which opto-inhibition was done at different points in learning, mice only did the simplest version of the task (stimuli were either 98% of high tones or 98% of low tones). The initial 20 trials of every session were discarded, and the baseline distributions of choice proportions and the session bias were calculated as indicated above. Because mice performed close to perfection in this version of the task, we only included in the analysis the choices to the port where biases were expected to happen, in agreement with our previous results (it is not possible to measure any positive contralateral bias in a trial in which the mouse chooses the contralateral port all the time in unstimulated trials). Mice were all implanted bilaterally, and the stimulated implant was alternated every

day of training. We only included in the analysis those implant sites which resulted in at least one significantly biased session ($P < 0.05$, calculated as indicated above). This removed 4 implants from the analysis out of 17. The regression models were done for each individual implant, and the average slope for each genotype was calculated. The significance of the global results were assessed against random datasets, generated by shuffles of the performance values associated with every session. *P* values were calculated as the proportion of 'random slopes' being larger (expecting negative and positive contralateral biases for D1-Arch and A2A-Arch, respectively).

## Computational modelling

**Modelling of APE and RPE photometry signal.** The RL model used an actor–critic framework with soft-max (inverse temperature = 5.0) policy selection. The model used a semi-Markov state representation, previously introduced by[74] to capture the time-course variability of dopamine commonly seen in experimental tasks. In the semi-Markov formalism, the agent keeps track of its current state as well as a representation of 'dwell time' within that state. Actor, critic and stimulus–action strengths are then updated only when a state transition takes place. It has the added advantage that the states in the model directly correspond to the within-trial behavioural stages of the task, while allowing for representation of time. This formalism has been previously used in prior work modelling dopamine signals in tasks with variable timing[75,76] and it allows us to model the time course of dopamine in a more realistic manner than other previous studies, which have often assumed that each time point within a behavioural stage of the task is an independent state (see ref. 77 for a review).

The states consisted of Start, High tone, Low tone, Outcome, as well as an action state associated with each of the time of action for each sound and action pairing (in order to align to the time of action to match the data). Actions consisted of Left, Right, Centre and Idle to represent the movement to enter left, right and centre ports (and not taking an action). A state transition happened when the agent moved between one of the above states, for example the start state to the high tone state. The behaviour is self-paced for the mice, with sometimes multiple seconds of variability in timing between trials. The critic had dimension $1 \times n$ states. The actor had dimension $n$ states × $n$ actions. The non-value dependent model had dimensions $n$ states × $n$ actions.

The RPE was scaled according to dwell time in the state to approximate temporal discounting[74] as is necessary for the semi-Markov representation. RPE was calculated at each state transition $k$ as

$$\delta_k = r_{k+1} - \rho_k \cdot d_k + \hat{V}(s)_{k+1} - \hat{V}(s)_k$$

where $r_{k+1}$ is the reward received in the new state, $\rho_k$ is the average reward per time step and $d_k$ is the time spent in the last state before transitioning. $\rho_k$ was calculated by looking at average reward per time step in the last $n$ ($n = 500$ in our model) state transitions:

$$\rho_k = \frac{\sum_{k'=k-n+1}^{k+1} r_{k'}}{\sum_{k'=k-n}^{k+1} d_{k'}}$$

The critic value function was then updated using the RPE $\delta_k$ signal:

$$\hat{V}(s)_k \leftarrow \hat{V}(s)_k + \alpha \delta_k$$

and the actor stimulus–action value function was updated in a similar manner:

$$\hat{m}(s, a) \leftarrow \hat{m}(s, a) + \beta \delta_k$$

where $\alpha$ and $\beta$ are both learning rates that were set to 0.005.

As in ref. 74, the RPE signal was then taken to be $\sigma(\delta_k + \psi)$, where $\sigma(x) = 0$ if $x \leq 0$ and $\sigma(x) = x$ if $x > 0$. $\psi$ was set to be 0.2 to represent a baseline level of dopamine.

The non-value dependent model calculated APE, $\delta_{a_k}$, at state transitions (see Extended Data Fig. 6b to see full set of actions and states in the modelled task) as the difference between the action taken, $a_k$, and the stimulus action strengths given the stimulus $A(s)$, according to the equation

$$\delta_{a_k} = a_k - A(s)$$

APEs were rectified, resulting in only one component of the APE vector being non-zero, essentially providing a scalar update signal. This rectification also matched the data better as dips in TS dopamine were never observed at time of action. Importantly, this rectification did not change the general properties of the algorithm. Stimulus–action pairing strengths were then updated using APE:

$$A(s) \leftarrow A(s) + \varepsilon \delta_{a_k}$$

where $\varepsilon$ is the rate at which the stimulus–action association was formed, which was set to 0.01 for all simulations apart from the predicted value change experiment simulation, where it was set to 0.001 to mimic how a value-free controller should update slowly. For other simulations this was less relevant as we aimed to show the direction of change of APE and RPE signals rather than their relative rates of updating (Extended Data Fig. 8).

Every time the mouse revisited a state it had visited before, a counter $I(s)$ was increased to keep track of how many times that state had been visited:

$$I(s)_{k+1} \leftarrow I(s)_{k+1} + 1.$$

This was used to compute novelty $N$, which decayed exponentially with exposure to a state, according to

$$N(s)_{k+1} = -\exp \gamma I(s)_{k+1}.$$

Salience $L$ was computed as a weighted combination of value and novelty of the state:

$$L(s)_{k+1} = \frac{\hat{V}(s)_{k+1}}{\mu} + N(s)_{k+1}.$$

In the above equations, $\gamma$ and $\mu$ were set to 0.01 and 0.5, respectively.

In all simulations except for the simulation of the effect of previous choice on subsequent dopamine response (Fig. 3i,l), the inverse temperature term of the soft-max was set to 5.0. For Fig. 3i,l it was set to 0.5, to better capture the more exploratory nature of the behaviour of the mouse and to ensure there were enough trials throughout learning with both ipsi- and contralateral previous choices.

The movement model of dopamine comprised a vector with length equal to the number of actions available. When an action was taken, the corresponding element in the vector was set to 1. All other elements were 0.

In all candidate models of dopamine photometry signals, learning rates and constants were not fitted to match the numbers of trials taken to learn the task seen in the experimental data, so trends should be interpreted as approximations of general patterns rather than models of the exact behaviour.

### Dual-controller network model
We simulated a task with two states (the first element, $s = 1$, corresponds to low tone, the second element, $s = 2$, corresponds to high tone) and two actions (first element corresponding to right and the second corresponding to left, $a = 1$ or $a = 2$). We modelled two parallel systems, one for the value-based controller that is updated by RPE, this had two components, an actor and a critic. The second system had a value-free controller that was updated by APE. The actor in the value-based controller and the value-free controller were modelled as a weight matrix (dimension $2 \times 2$) that projected from the sensory states to the two possible actions that could be taken, the weights within these matrices are notated as $W_{actor}$ and $W_{tail}$. The critic in the value-based controller was modelled as a $2 \times 1$ matrix, $W_{critic}$, with each sensory state projecting to one of the cells in the matrix. The output for the $W_{actor}$ and the $W_{tail}$ on each trial was computed as the weight matrix × the state input—that is, $A(s)_{actor} = W_{actor}s$ and $A(s)_{tail} = W_{tail}s$.

To calculate the actual action taken by the model, we first calculate the action predicted by the sum of the outputs of the two controllers. An additional noise term was added, $A(s)_{noise}$ (drawn from a uniform distribution between 0 and 1):

$$A(s)_{total} = A(s)_{actor} + A(s)_{tail} + A(s)_{noise}$$

The action associated with the maximal value of $A(s)_{total}$ was the action $a$ that the model took (corresponding to the choice of the mouse)—that is, $a = \text{argmax}(A(s)_{total})$.

The model received a reward, $r = 1$, if the index of the action taken, $a$, corresponded to the action rewarded (here $r = 1$ if: $s = 1$ and $a = 1$ or $s = 2$ and $a = 2$).

We then computed the RPE and APE signals. For the RPE signal, we first calculated the predicted value from the critic as $V = W_{critic}s$. We then calculated the reward prediction error as

$$\delta_{RPE} = r - V$$

To calculate APE and represent it as a scalar value we first binarized the instantaneous predicted action vector: $A_{tail}^{binary}$, such that the index corresponding to the maximum $A_{tail}$ was 1 and 0 everywhere else. Then we computed the predicted action $p_a$ (initialized at 0) as a low-pass filter of this binary choice vector:

$$\tau_{tail}\frac{dp_a}{dt} = -p_a + A_{tail}^{binary},$$

with a time constant $\tau_{tail} = 100$. The APE was then calculated as $\delta_{APE} = a - p_a$ for the action taken.

Finally, we updated the $W_{actor}$ and $W_{tail}$ weights as a function of a three-factor learning rule that included the sensory state, the action taken (that is, both the value-based actor and the value-free controllers receive an efference copy of the action taken) and the APE/RPE signal on the current trial:

$$W_{tail} \leftarrow W_{tail} + \beta\delta_{APE}sa,$$

where $\beta = 0.02$ is a learning rate;

$$W_{actor} \leftarrow W_{actor} + \alpha\delta_{RPE}sa,$$

where $\alpha = 0.04$ is a learning rate.

The $W_{critic}$ weights were updated by a two-factor learning rule that included the sensory state and the RPE signal on the current trial:

$$W_{critic} \leftarrow W_{critic} + \alpha\delta_{RPE}s$$

The RPE weights of the actor also decayed to its steady-state value of 1 with a time constant of $\tau_{decay} = 100$:

$$\tau_{decay}\frac{dW_{actor}}{dt} = -W_{actor} + 1$$

The $W_{tail}$ and $W_{actor}$ weights were bounded to be positive and were initialized to 1. The $W_{critic}$ values were initialized as 0 and were also bounded to be positive.

We simulated 100 trials. A trial lasted 1,000 timesteps (2,000 for the inactivations). For plotting the performance, we low-pass filtered the reward with a time constant of 10 a.u., starting from 0.5. The inactivations were conducted at time 30, 100 and 1,000 after which either the $W_{actor}$ (for the RPE system) or the $W_{tail}$ (for the APE system) were set to 0 and weights were not plastic anymore. For the psychometric curve, we varied the difficulty level $d$ from 0 to 1 and set the first element of $s = 1 - d$ and the second element of $s = d$. For the TS dopamine stimulation simulations, we doubled the APE signal.

## Reporting summary

Further information on research design is available in the Nature Portfolio Reporting Summary linked to this article.

## Data availability

The data that support the findings of this study are available at https://rdr.ucl.ac.uk/projects/Dopaminergic_action_prediction_errors_serve_as_a_value-free_teaching_signal/240596. In addition, Allen Brain Atlas connectivity (https://connectivity.brain-map.org/) data were used to demarcate the cortico-striatal projection patterns from AUDp (experiments 100149109, 116903230, 120491896 and 146858006) and S1(SSp) (experiments 112882565, 114290938 and 126908007). Source data are provided with this paper.

## Code availability

The analysis code is available at https://doi.org/10.5281/zenodo.15119142.

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

**Acknowledgements** The authors thank the Advanced Microscopy Facility and Viral Vector Core in the SWC for assistance with histology, imaging and custom production of viral vectors, especially J. Broni-Tabi, R. Campbell, G. Estrada Girona and M. Strom; F. Claudi for advice on analysis; M. Watabe-Uchida, T. Mrsic-Flogel, S. Hofer, T. Branco and A. McAskill for their comments on the manuscript; and all members of the Stephenson-Jones laboratory for their contributions to this project, particularly F. Marbach. This work was supported by an EMBO Long-Term Fellowship (ALTF 827-2018) (H.M.V.), a Ramon y Cajal Fellowship (RYC2022-035145-I) (H.M.V.), a Swedish Research Council International Postdoc Grant (2020-06365) (Y.J.), the Sainsbury Wellcome Centre Core Grant from the Gatsby Charitable Foundation and Wellcome (219627/Z/19/Z) (M.S.-J.), the Sainsbury Wellcome Centre PhD Programme (F.G.) and a European Research Council grant (starting no. 557533) (M.S.-J.). Mouse silhouettes were obtained from https://scidraw.io/ and https://doi.org/10.5281/zenodo.3925917 (Figs. 1a, 3n, 4b,e and Extended Data Figs. 3k,l, 7a,g, 9g,i, 12b,e) and https://doi.org/10.5281/zenodo.3925901 (Extended Data Figs. 5h,u, 10c).

**Author contributions** M.S.-J., F.G. and H.M.V. conceived and designed the study. H.M.V., F.G., Y.J., S.P., M.W., L.S., S.C.L., M.S.-J., A.G., J.C. and J.K. performed the experiments. H.M.V., F.G., Y.J., S.C.L. and L.B.R. wrote the software for data analysis. F.G., H.M.V., Y.J., S.P., C.C., L.S., T.M., E.T., L.B.R. and J.P.G. analysed the data. H.M.V., F.G., Y.J., S.P. and M.S.-J. made the figures. M.S.-J., F.G., H.M.V. and Y.J. wrote the manuscript. M.S.-J., T.W.M. and C.C. provided resources. All work was supervised by M.S.-J. H.M.V., F.G. and C.C.

**Competing interests** The authors declare no competing interests.

**Additional information**
**Correspondence and requests for materials** should be addressed to Marcus Stephenson-Jones.

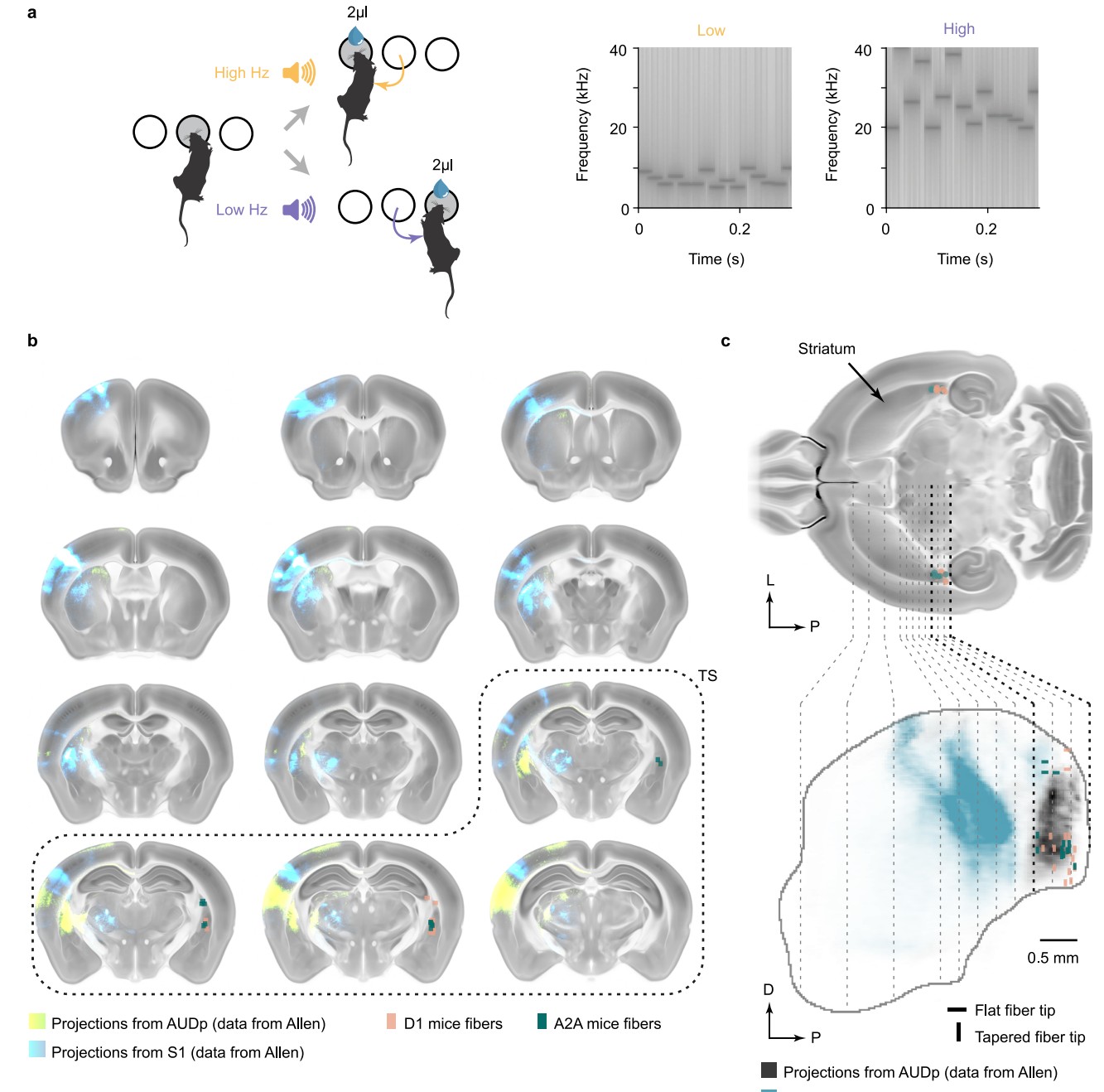

**Extended Data Fig. 1 | Task details and Optoinhibition histology. a**, Schematic of the task, left. Example spectrograms showing low (left plot) and high (right plot) "cloud of tone" stimuli. **b**, Coronal sections along the striatum indicating fiber placement positions (tip of the fiber). Note that fibers were inserted in both hemispheres and are mirrored here for illustration purposes. Primary auditory cortex (AUDp) projections (yellow) and primary somatosensory cortex (SSp) projections (blue) are shown in the other hemisphere. **c**, Horizontal (top) and side view (bottom) of the same histological data. For the horizontal section the recording location depths are collapsed onto a single horizontal atlas image for illustrative purposes. On the side view, the striatum is outlined and the AUDp projections are indicated in grayscale. All error bars represent the standard deviation.

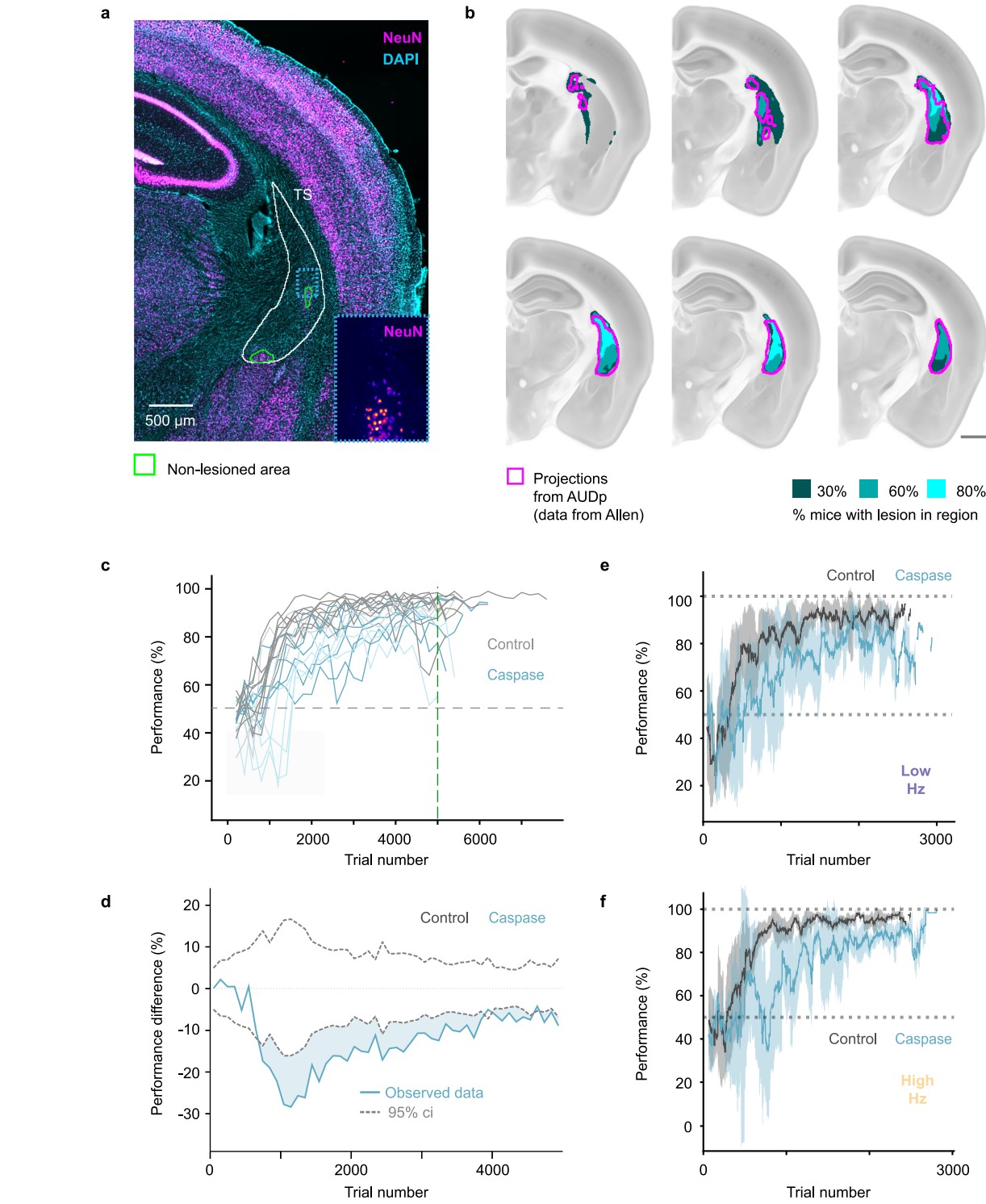

**Extended Data Fig. 2 | Histology and quantification of TS lesions.**
**a**, Representative image of a lesioned brain illustrating the image analysis used to quantify the proportion of lesioned striatum. Slices were registered to the atlas and the area with remaining neurons (as stained by NeuN) was defined. The rest of the striatum was considered as lesioned. **b**, Quantification of the lesion area for the 11 mice in the caspase dataset, across several coronal slices of the posterior striatum, that include the entire TS. **c**, Learning rate (performance vs. trial

number) of individual mice in the lesion in TS and the control groups. Light blue and light gray traces are from mice that had an initial bias. **d**, Differences of means in performance between control and lesion groups. Dotted lines indicate the 95% confidence interval for the shuffled data (see methods). **e**, Learning rate for trials with the low tone stimulus (performance vs. trial number) of lesion TS and the control groups (shaded area indicates standard deviation). **f**, Same as F but for high tone stimulus trials (shaded area indicates standard deviation).

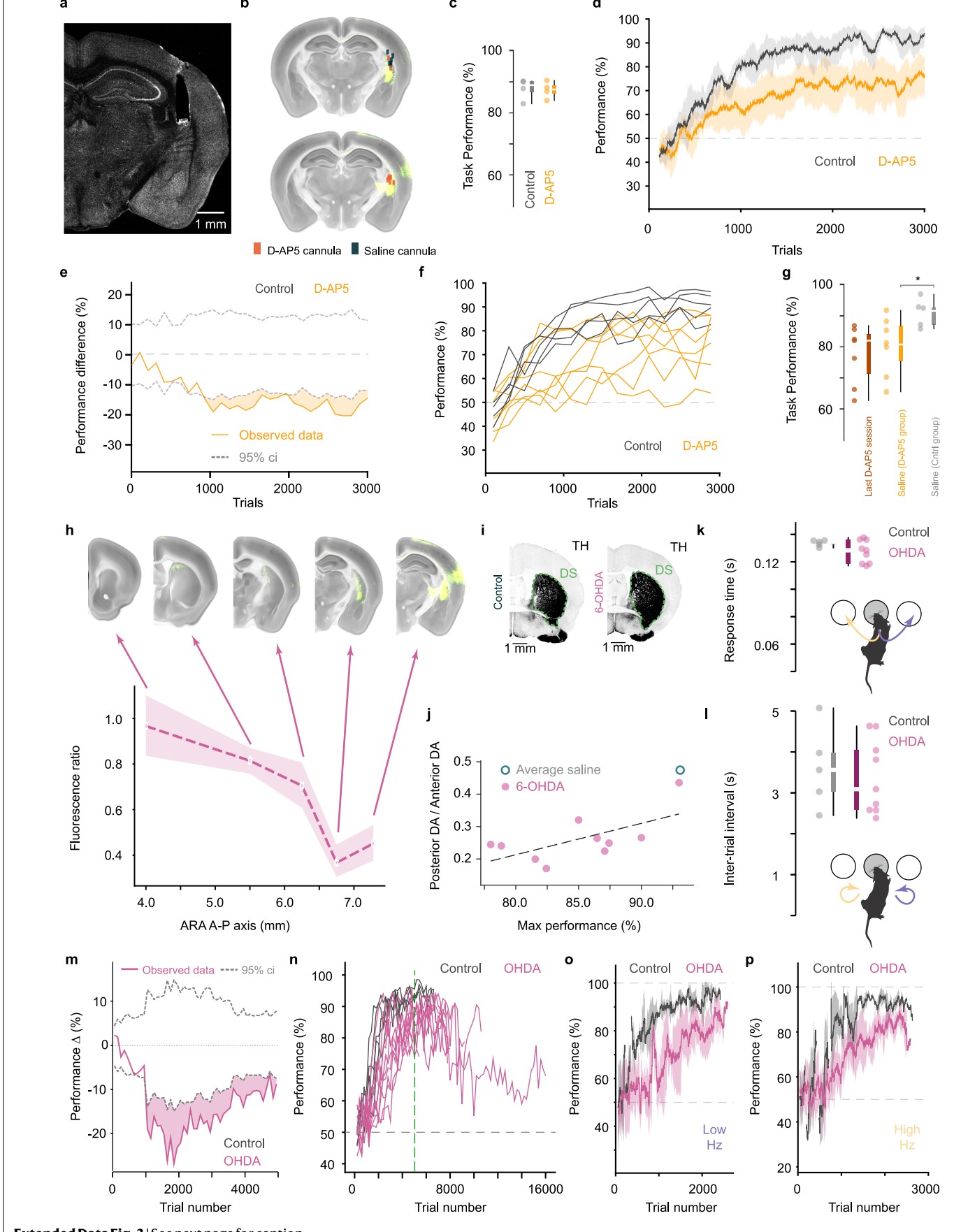

**Extended Data Fig. 3 |** See next page for caption.

**Extended Data Fig. 3 | Learning effects of D-AP5 infusions and the histology of TS-dopamine ablated mice and task response parameters. a**, Representative image showing the location of an infusion cannula implanted over the TS. **b**, Coronal sections along the striatum indicating cannula placement positions (center tip of the cannula). Note that cannulas were inserted in both hemispheres and are mirrored here for illustration purposes. Primary auditory cortex projections are also shown (yellow). **c**, Behavioral effect of acute D-AP5 infusion in expert mice (>5000 trials). n = 4 mice. **d**, Learning rate (performance vs. trial number) of D-AP5 and saline infusion groups (shaded area indicates standard deviation). D-AP5 n = 7 mice, Control n = 5 mice. **e**, Differences in performance between the groups. Dotted lines indicate the 95% confidence interval for the shuffled data (see methods). **f**, Same as D but showing the data from individual mice. **g**, Behavioral performance of mice in the last session of chronic D-AP5 infusion (brown), or when saline was infused in the first session after reaching 3000 trials, D-AP5 group (yellow) or the control group (gray). **h**, Quantification of the TH staining fluorescence ratio between the striatum and the cortex after background subtraction, at different levels in the allen reference (ARA) anterior-posterior axis. The data is shown as the fluorescence relative to controls.

Primary auditory cortex projections are shown. **i**, Example TH-stained (dopamine axons) coronal slices at the level of the dorsal striatum (DS) for control and lesion mice. **j**, Correlation between the maximum performance achieved for each mouse and the lesion size (p = 0.049, two-sided Wald test). **k**, Time elapsed between center port poke and side port pokes, as medians for each animal, for the 6-OHDA and control groups (p = 0.31, Kruskal-Wallis test). **l**, Time elapsed between trials, as medians for each animal, for the 6-OHDA and control groups (p = 0.73, Kruskal-Wallis test). **m**, Differences of means in performance between control and lesion groups (6-OHDA n = 9 mice, Control n = 5 mice. Dotted lines indicate the 95% confidence interval for the shuffled data (see methods). **n**, Learning rate (performance vs. trial number) of individual mice in the 6-OHDA and the control groups (same mice as panel m). **o**, Learning rate for trials with the low tone stimulus (performance vs. trial number) of 6-OHDA and the control groups (same mice as panel m, shaded area indicates standard deviation). **p**, Same as O but for high tone stimulus trials. All boxplots show the range from quartile (Q1 - Q3), the median and the whiskers extend to the farthest data point lying within 1.5x the inter-quartile range (IQR) from the box.

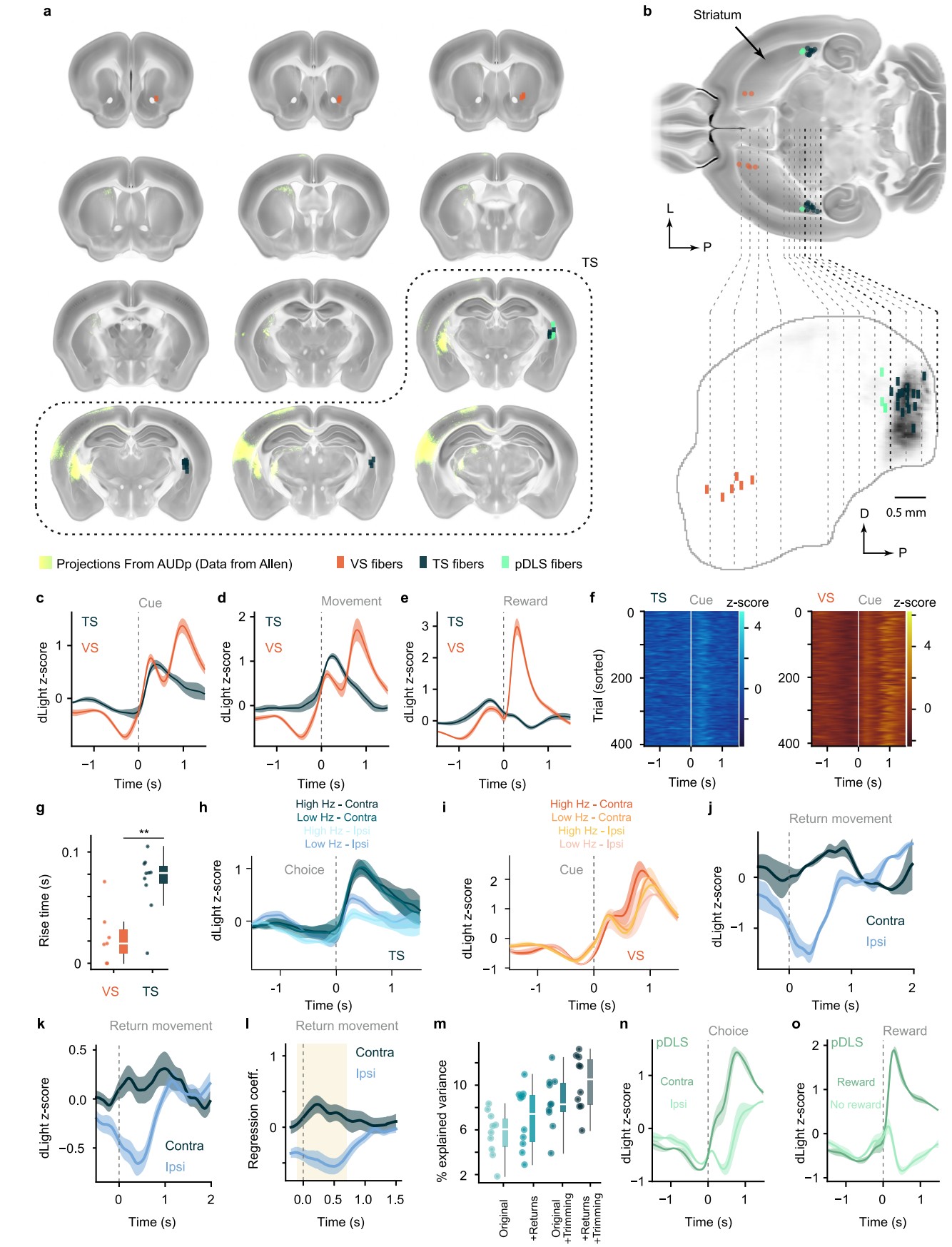

**Extended Data Fig. 4** | See next page for caption.

**Extended Data Fig. 4 | Histology and characterization of the photometry responses. a**, Coronal sections along the striatum indicating fiber placement positions (center tip of the fiber). Note that fibers were inserted in both hemispheres and are mirrored here for illustration purposes. Primary auditory cortex projections are shown in the other hemisphere. **b**, Horizontal (top) and side view (bottom) of the same histological data. For the horizontal section the recording location depths are collapsed onto a single horizontal atlas image for illustrative purposes. On the side view, the striatum is outlined and the AUDp projections are indicated in grayscale. **c**, Average photometry traces from TS (blue, n=10 mice) and VS (orange, n = 7 mice) aligned to cue early in training (first three recorded sessions). Performance in these sessions was between 52.8% to 75.8% with an average performance of 64.0%. For TS mice it was comparable to VS mice (TS: min: 51.4%, max: 80.0%, avg: 64.5%; VS: min: 55.8%, max: 70.1%, avg: 63.3%). **d**, Same as C but aligned to contralateral choice. Movement initiation for choice (leaving the center port) occurs on average (0.19 s +/- 0.05 s) after cue onset. **e**, Same as C but aligned to reward delivery. **f**, Example of a TS (blue) and VS (orange) recording session aligned to cues predicting a contralateral choice. **g**, Comparison of rise time for dopamine response from onset of the cue in the VS (n = 7) and TS (n = 10) early in training (first three recorded sessions). Rise time onset is determined by the time taken for the dopamine trace to reach more than one standard deviation above a baseline period (1.5 s prior to cue onset) (p = 0.002 two-sided independent samples t-test), Cohen's d = 1.90. **h**, TS dopamine responses for contralateral and ipsilateral choices aligned to movement onset early in training (first three

recorded sessions). Shown separately for mice for whom contra-choice corresponded to high frequencies (n = 6) or mice for whom the contra choice corresponded to the low frequencies (n = 4). **i**, Same as H but for VS recordings aligned to cue onset. Shown separately for mice for whom contra-choice corresponded to high frequencies (n = 2) or mice for whom the contra choice corresponded to the low frequencies (n = 5). (n = 7 mice). **j**, Average photometry trace across the first 3 recorded sessions early in training, for an example mouse aligned to leaving to the side ports to return to the center port. **k**, Same as J but an average of all mice (n = 10). **l**, Average regression kernels across mice (n = 10) for the return to center behavioral events. **m**, Percentage explained variance for different kernel regression models of TS dopamine. Original kernel regression model with only choice (center port exit): median = 5.1, original model including the return to center (ipsi and contra) behavioral events median = 7.1, original model only allowing photometry data for which there are behavioral events to be included in the explained variance calculation "+ trimming" median = 8.5, model with return events and trimming median = 10.3. For one session in the original regression there was no video recording, so this is why the 'original' explained variance is slightly different to that reported in the main figure, as this session was not included in this analysis. **n**, Average photometry traces from cDLS (n = 3 mice) aligned to choice. **o**, Same as N but aligned to reward. All error bars represent SEM. All boxplots show the range from quartile (Q1 - Q3), the median and the whiskers extend to the farthest data point lying within 1.5x the inter-quartile range (IQR) from the box.

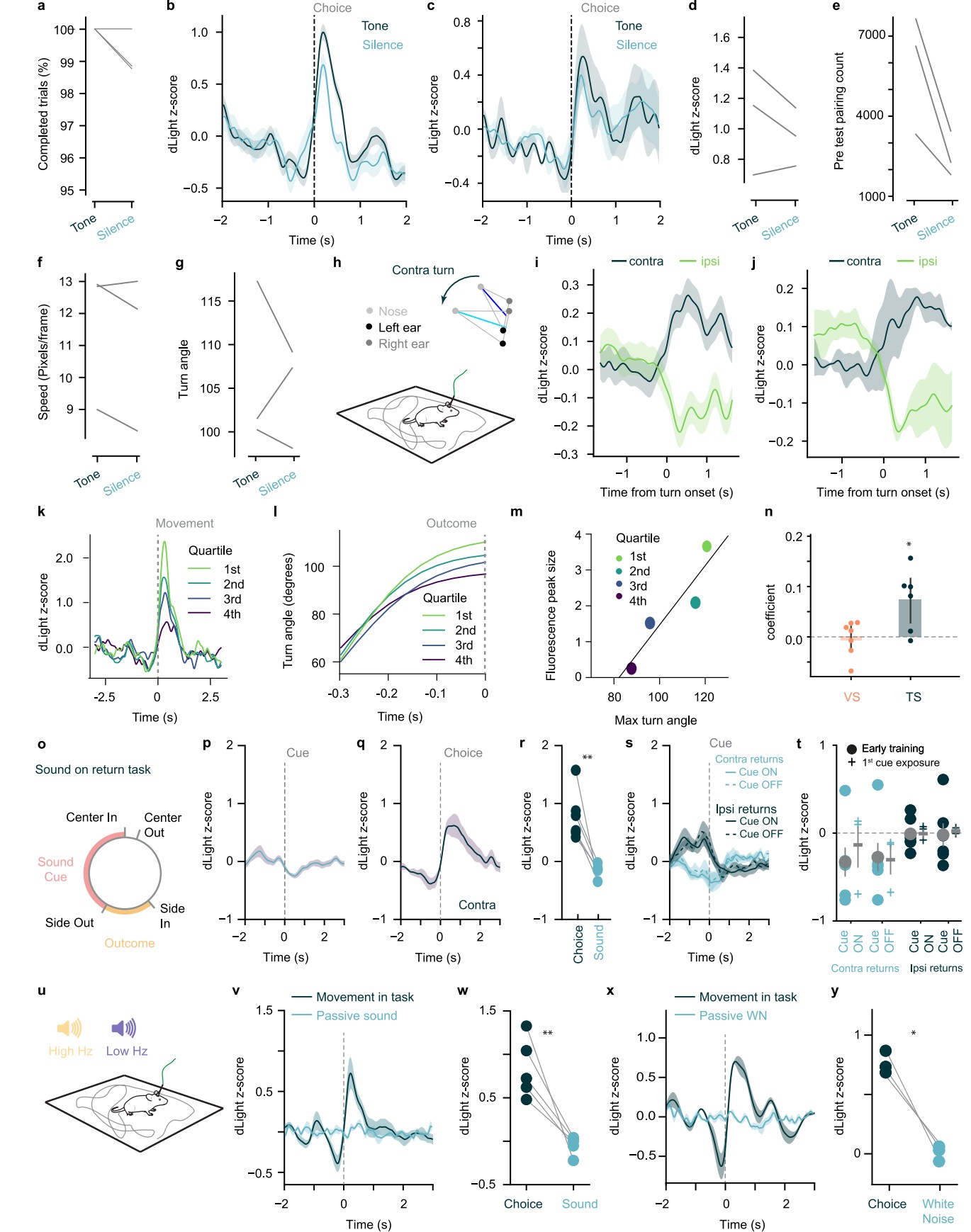

**Extended Data Fig. 5 | See next page for caption.**

**Extended Data Fig. 5 | TS dopamine is related to movement, not cue. a**, Change in percentage of completed trials (where animals made a left or right choice after leaving the center port) in trials with a normal stimulus (tone) or silence trials (p = 0.23, paired two-sided t-test, Cohen's d = 1.15). **b**, Average photometry traces of one mouse aligned to contralateral choice. **c**, As for B but an average across mice (n = 3 mice). **d**, Change in peak response to the contralateral choice if the associated cloud of tones is replaced with silence (p = 0.31, paired two-sided t-test, Cohen's d = 0.79). **e**, The number of tone-contralateral action pairings or silence-contralateral action pairings that the mice experienced prior to the recordings in B-D, (p = 0.07, paired two-sided t-test, Cohen's d = 2.10). **f**, Difference in average speed for tone and silence trials for TS mice (n = 3), (p = 0.28, paired t-test, Cohen's d = 0.84). **g**, Difference in average turn angle for tone and silence trials for TS mice (n = 3), (p = 0.75, paired two-sided t-test, Cohen's d = 0.21). **h**, Animals were allowed to move freely in a different arena to the training box whilst dLight signals were recorded from the TS. Inset: Head angle during a detected turn in the freely moving arena. Dark blue line represents the orientation of the head at the beginning of the turn. Light blue line shows head orientation 0.5 s later. **i**, Example photometry response in the TS to contralateral and ipsilateral turn onsets in the freely moving arena. **j**, As in I but averaged across animals (n = 3 mice). **k**, Example traces from a TS recording session in the frequency discrimination task separated by size of the response. **l**, Average turn angle for these quartiles plotted against quartile midpoint (example session). **m**, The plateau of the sigmoid for each trial turn angle vs the average peak size of the TS photometry signal per quartile based on the photometry signal (example session). **n**, Data from early in training (first three recorded sessions) is analyzed as shown in O and a regression slope fitted (TS: n = 18 (6 mice, 3 sessions), VS: n = 21 (7 mice, 3 sessions). The slopes of the regressions (averaged per mouse) were tested against zero (one sample two-sided t-test). The TS slopes were significantly greater than zero (p = 0.03, Cohen's d = 1.20), whereas the VS slopes were not (p = 0.87, Cohen's d = −0.06).

**o**, Schematic of the task structure when sound indicating an upcoming contralateral trial was played as mice return to the center port in 51.66 +/− 0.04 % of trials. **p**, Average dopamine response aligned to sound played while mice returned to the center during early training (n = 6 mice). **q**, Same as p but aligned to contralateral choice. **r**, Average amplitude of dopamine response aligned to the sound and choice across mice (p = 0.0047, Cohen's d = 1.97). Size of the sound response is significantly smaller than zero (p = 0.008793, one-sample one-sided t-test against zero). **s**, Same as P but ipsi- and contralateral returns are plotted separately and returns without concomitant cue are also shown. **t**, Difference in response size of returns shown in S (circles, p = 0.2945, one way ANOVA, n = 6 mice) and response sizes of mice when the cue is novel during the first training session (crosses, p = 0.2815, Kruskal Wallis test, n = 3 mice). **u**, Schematic of the arena where the high and low tone task sounds were played passively as the mice explored. Passive sounds responses were tested in mice that were at an early stage of training on the CoT task (average performance 60.6 +/− 6.4 % in n = 5 mice). **v**, Average dopamine response in the TS during contralateral movement in the 2AC task (dark blue) and during passive sound presentation in subsequent exploration (pale blue) (n = 5 mice). **w**, Average amplitude of choice aligned response in the task and of passive sound response during exploration across mice (p = 0.0092, paired t-test, Cohen's d = 2.11). There is no significant response to the sound (p = 0.45, one-sample two-sided t-test against zero). **x**, Average dopamine response in the TS during contralateral movement in the 2AC task (dark blue) and during passive presentation of white noise in subsequent exploration (pale blue) (n = 3 mice). Mice had on average experienced 93 presentations +/- 3.30 of white noise as a task cue before these recordings, this is less than half than during the CoT state-change experiment (195 + /−35 trials). **y**, Average amplitude of choice aligned response in the task and of white noise response during exploration across mice (p = 0.02, paired t-test, Cohen's d = 3.87). There is no significant response to the sound (p = 0.84, one-sample two-sided t-test against zero). All error bars represent SEM.

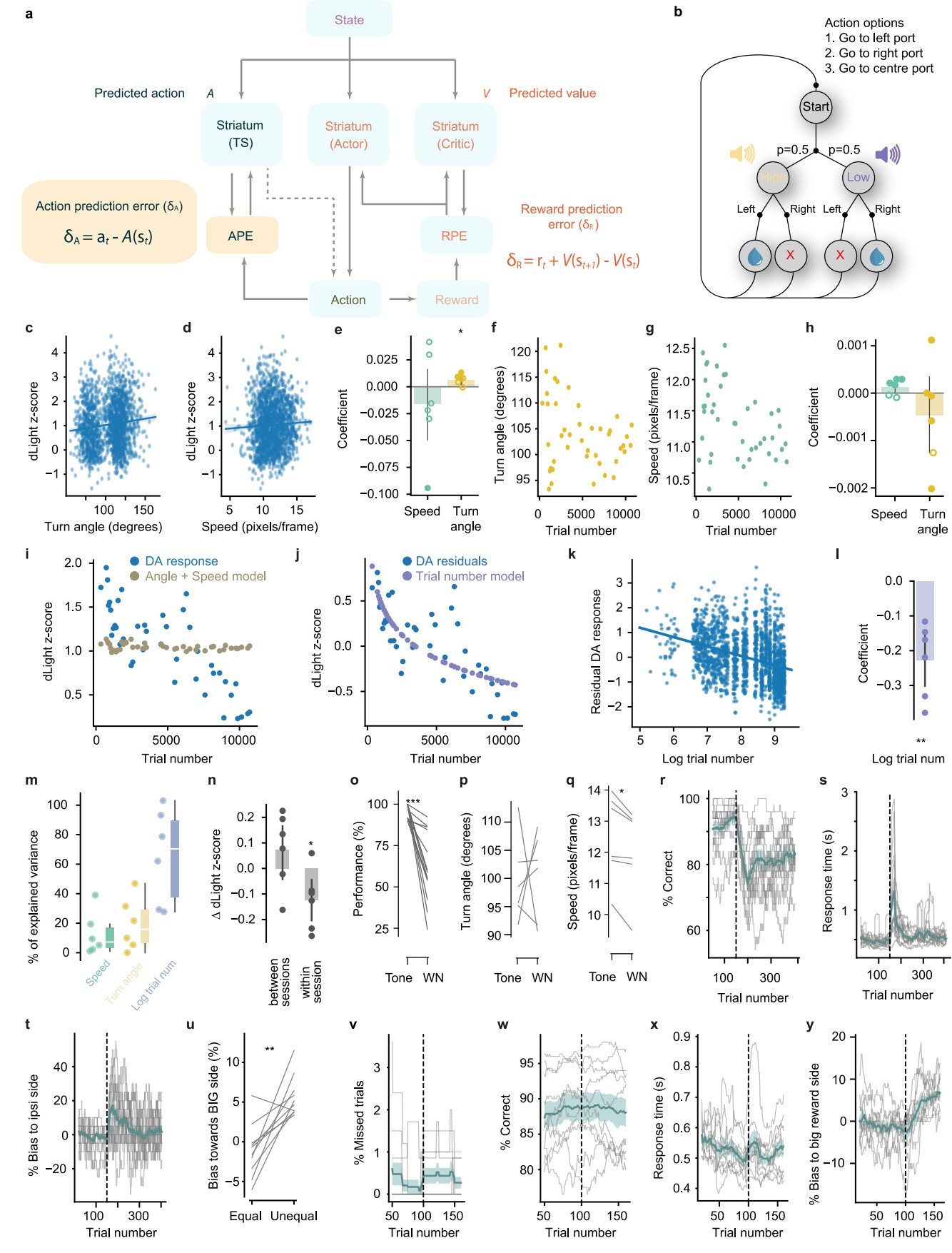

**Extended Data Fig. 6** | See next page for caption.

**Extended Data Fig. 6 | APE model and tests. a**, The model comprises an actor that learns stimulus-action values and guides action choices, a critic that learns a value function that is used to calculate RPE. The RPE signal is broadcast to the actor and critic to update their respective value functions. A value-free system learns to predict actions from those taken in the past and updates its prediction using the difference between its prediction and the action taken (APE). APE and RPE equations are written with respect to time (t), as is common, for illustrative purposes. For the model equations we use dwell time in the state (k) to approximate temporal discounting, see methods. **b**, The Markov decision process used to model the task. **c**, Correlation between the turn angle and the size of the dopamine response in the TS for all trials in all sessions of an example mouse. **d**, Correlation between the average speed of an example mouse and the TS dopamine response for all trials in all sessions. **e**, Linear regression coefficients for speed and turn angle on single trial TS (n = 6 mice) dopamine responses for the first three sessions of training. Stats: one-sample two-sided t-test against zero, speed: (p = 0.448, Cohen's d = −0.34), turn angle: (p = 0.033, Cohen's d = 1.20). Filled circles represent significant correlations for individual mice. Error bars represent 95% confidence interval. **f**, Turn angle of an example mouse over the course of training, binned per 40 trials. **g**, Average speed during a choice of an example mouse over the course of training, binned per 40 trials. **h**, Linear regression coefficients for the effect of trial number on speed or turn angle at a single trial level (n = 6 mice). Stats: one-sample two-sided t-test against zero, speed: (p = 0.154, Cohen's d = 0.68), turn angle: (p = 0.340, Cohen's d = −0.43). Filled circles represent significant correlations for individual mice. Error bars represent 95% confidence interval. **i**, TS dopamine response, binned per 40 trials of an example mouse over the course of training (blue). A linear regression model was built using average speed and turn angle to predict the TS dopamine signal. The model prediction from just the movement parameters over the course of training is shown in gold (binned per 40 trials). **j**, The movement model used in panel I was subtracted per trial from the TS dopamine responses to give the remaining signal that was not explained by speed or turn angle (residuals in blue). A new linear regression model was built using log trial number to account for the remaining TS dopamine signal (purple). Both are shown binned per 40 trials. **k**, The correlation between the individual trial residual dopamine responses and log trial number for an example mouse. **l**, Regression coefficients for the effect of log trial number on the residual dopamine response (n = 6 mice) (filled circles show significant correlations for individual mice). One-sample two-sided t-test against 0, p = 0.003, Cohen's d = −2.14. Error bars represent 95% confidence interval. **m**, Total model variance explained by each parameter in a model where speed, turn angle and trial number are used to predict the size of the TS (n = 6 mice) dopamine response throughout learning. **n**, Difference between TS dopamine response in the last 40 trials of a previous session and next 40 trials of a session (between sessions) and first 40 trials of a session and last 40 trials of the same session (within session) (n = 6, between sessions: p = 0.27 one-sample two-sided t-test against 0, two-sided t-test Cohen's d = 0.51, turn angle p = 0.05 one sample t-test against 0 two-sided t-test, Cohen's d = −1.07). Error bars represent 95% confidence interval. **o**, Performance in the 50 trials before and after the state change (n = 13 mice, p = 1.98×10-4 paired two-sided t-test, Cohen's d = 1.46). **p**, Changes in turn angle before and after the state change (n = 13 mice, p = 0.85 two paired two-sided t-test, Cohen's d =−0.08). **q**, Changes in average speed before and after the state change (n = 13 mice, p = 0.02 paired two-sided t-test, Cohen's d = 1.42). **r**, Performance before and after state change at trial 150 (black dashed line) binned per 20 trials (n = 13). Green lines show mean, error bars represent sem, grey lines represent data from individual mice. **s**, Same as R but showing the response time following the state change. **t**, Same as R but showing the bias towards ipsilateral choices following the state change. **u**, behavioral bias towards the large reward port before and after the change in value. The last 50 trials from each block are used for analysis with blocks being a minimum of 70 trials (n = 10 mice, p = 0.002 paired two-sided t-test, Cohen's d = 1.39). **v**, Percentage of trials where mice did not make a choice before and after the value change is introduced at trial 100 (black dashed line) binned per 20 trials (n = 10 mice). Green lines show mean, error bars represent sem, grey lines represent data from individual mice. **w**, Same as V but showing the change in performance. **x**, Same as V but showing the change in response time. **y**, Same as V but showing the change in choice bias over the course of the session. All boxplots show the range from quartile (Q1 - Q3), the median and the whiskers extend to the farthest data point lying within 1.5x the inter-quartile range (IQR) from the box.

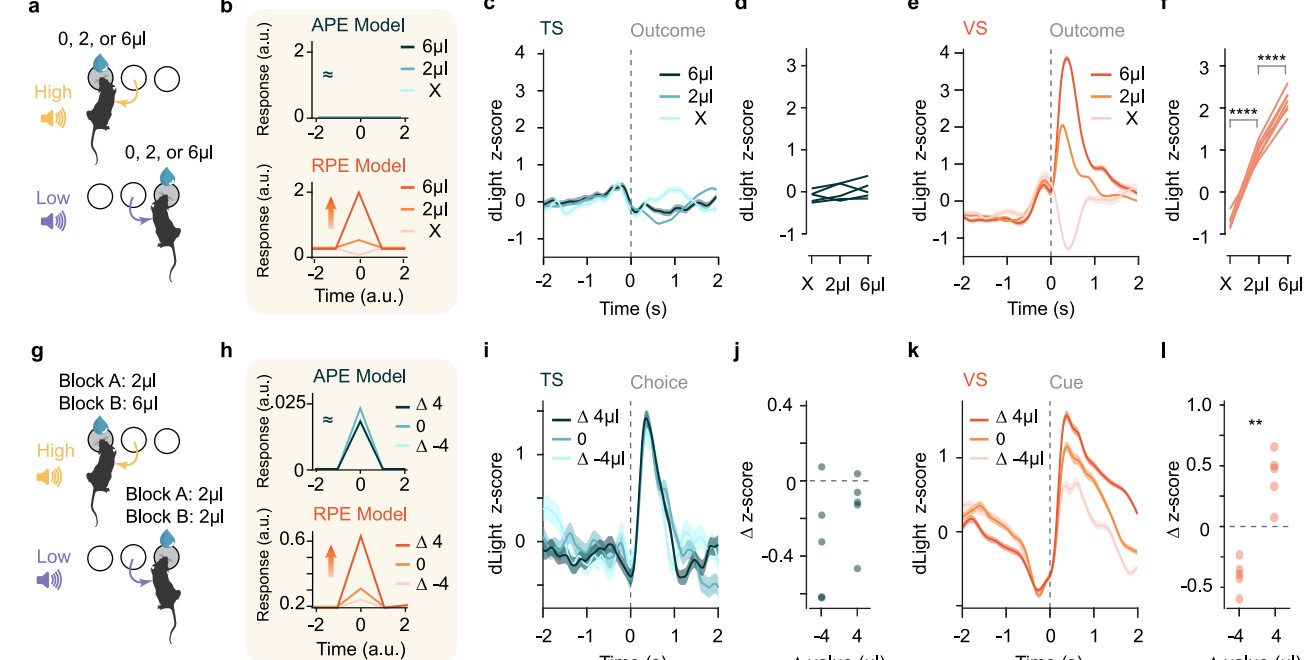

**Extended Data Fig. 7 | Response of TS and VS dopamine responses to value manipulations. a**, Schematic of the outcome manipulation task design. **b**, Modelled responses for how APE and RPE signals would respond to changes in reward outcome. **c**, Example TS response to omissions, normal sized and large rewards. **d**, Group data (n = 6 mice) for TS dopamine response size to omission, normal and large reward (p = 0.84, p = 0.54, paired two-sided t-test, adjusted using Bonferroni correction), (Cohen's d: large > normal 0.40, normal > omission 0.57). **e**, Same as c but for a VS recording. **f**, Same as d but for VS recording (n = 7 mice, p = 3.16×10-6, p = 8.87×10-6, paired two-sided t-test,

adjusted using Bonferroni correction), (Cohen's d: large > normal 7.01, normal > omission 5.89). **g**, Schematic of the predicted value manipulation task design. **h**, Model predictions for changes in predicted outcome value. **i**, Example movement-aligned TS response when the relative value of the cues changed. **j**, Summary data showing the change in movement-aligned response in the TS for relative value changes (p = 0.26, paired two-sided t-test) (n = 5 mice), Cohen's d = 0.58. **k**, Same as i but for the VS response aligned to cue. **l**, Same as j but for the VS response aligned to cue (p = 2.41×10-4, paired two-sided t-test) (n = 5 mice), Cohen's d = 5.56. All error bars represent SEM.

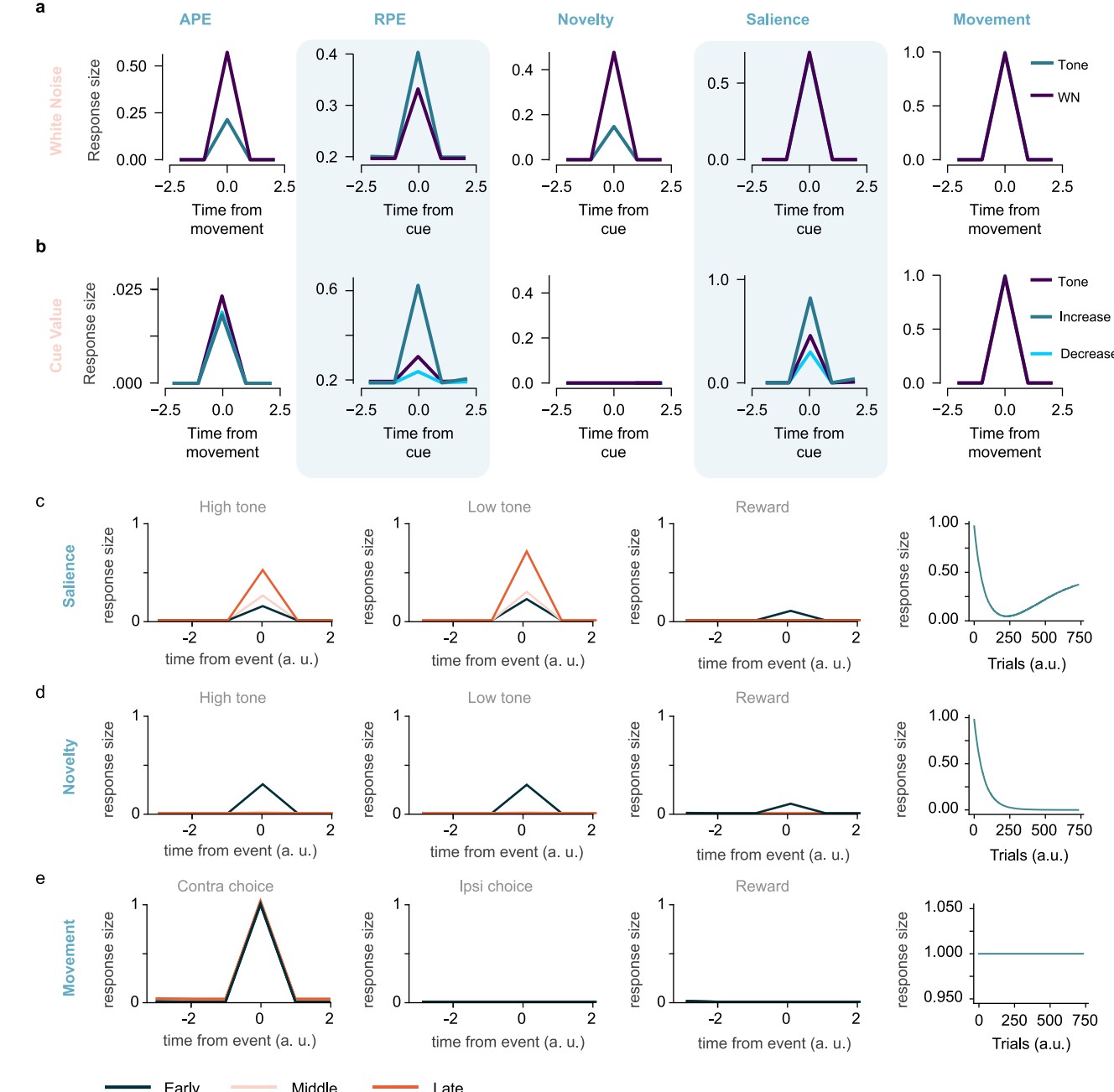

**Extended Data Fig. 8 | APE and RPE models. a**, Model predictions for how the different models of dopamine respond in the state change experiment. **b**, Model predictions for how the different models of dopamine respond in the cue value change experiment. **c**, Salience model dopamine signal aligned to time of high cue, low cue, and reward. Size of response to cues shown for 100 agents over training. **d**, Novelty model dopamine signal aligned to time of high cue, low cue, and reward. Size of response to cues shown for 100 agents over training. **e**, Movement model dopamine aligned to time of contralateral choice, ipsilateral choice, and reward. Size of response to contralateral choice shown for 100 agents over training.

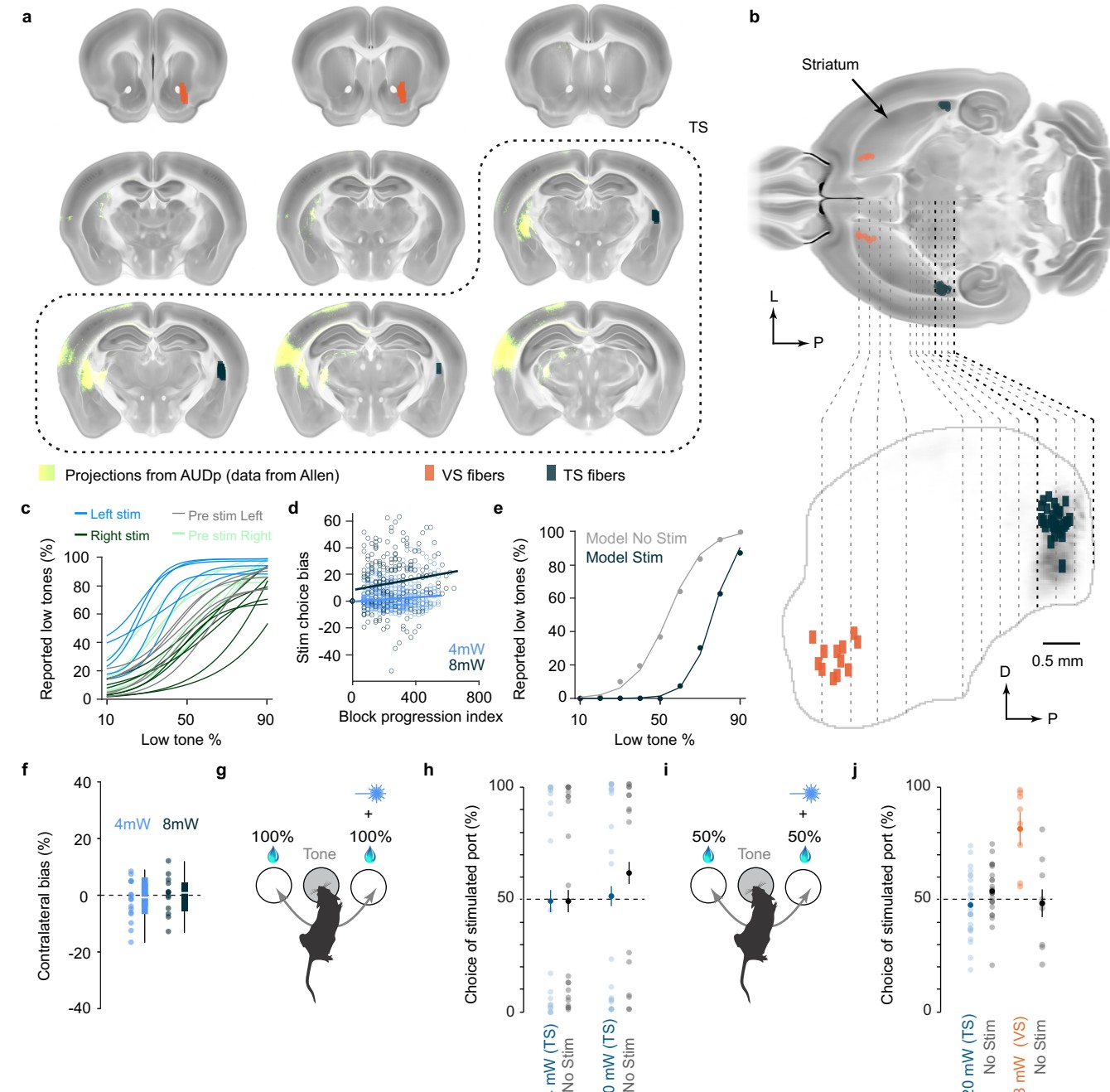

**Extended Data Fig. 9 | Histology and effects of dopamine optostimulation.**
**a**, Coronal sections along the striatum indicating fiber placement positions
(center tip of the fiber). Note that fibers were inserted in both hemispheres and
are mirrored here for illustration purposes. Primary auditory cortex projections
are shown in the other hemisphere. **b**, Horizontal (top) and side view (bottom)
of the same histological data. For the horizontal section the recording location
depths are collapsed onto a single horizontal atlas image for illustrative purposes.
On the side view, the striatum is outlined and the AUDp projections are indicated
in grayscale. **c**, Psychometric curves for all 8 mW sessions where dopamine
release was optogenetically stimulated in the TS at the time of choice (center
port). **d**, Scatterplot showing that the stimulation choice bias develops over the
course of a session (8 mW, p = 0.035; 4 mW, p = 0.006, linear regression). **e**, Model
simulation of the state-action optogenetic experiment. The APE was unilaterally
stimulated when the cue predicted a contralateral action. **f**, Quantification of
the contralateral choice bias on stimulated trials when optogenetic stimulation

in the TS was delivered on 15% of trials at the center port. session (4 mW, p = 0.67;
8 mW, p = 0.28, Kruskal-Wallis test); 4 mW: n = 9 mice, 18 hemispheres mice,
8 mW n = 7 mice, 13 hemispheres **g**, Experimental design: animals enter a
central port (gray) to initiate trials and then choose between water and water +
optogenetic stimulation of dopamine release in the TS by entering one of two
side ports. **h**, Choice bias (n = 5 mice) for 4 Hz optogenetic stimulation (p = 0.99,
t-test) or 20 Hz optogenetic stimulation (p = 0.49, t-test). Solid dots indicate
mean ± SEM and transparent dots indicate single sessions from individual
animals. **i**, Same as g but with a 50% probability of receiving a water reward at
the choice ports. **j**, Same as h but with the probability of receiving a water
reward at the choice ports reduced to 50%. (TS, p = 0.11; VS, p = 0.002 Cohen's
d = 1.80, t-test). All boxplots show the range from quartile (Q1 - Q3), the median
and the whiskers extend to the farthest data point lying within 1.5x the inter-
quartile range (IQR) from the box.

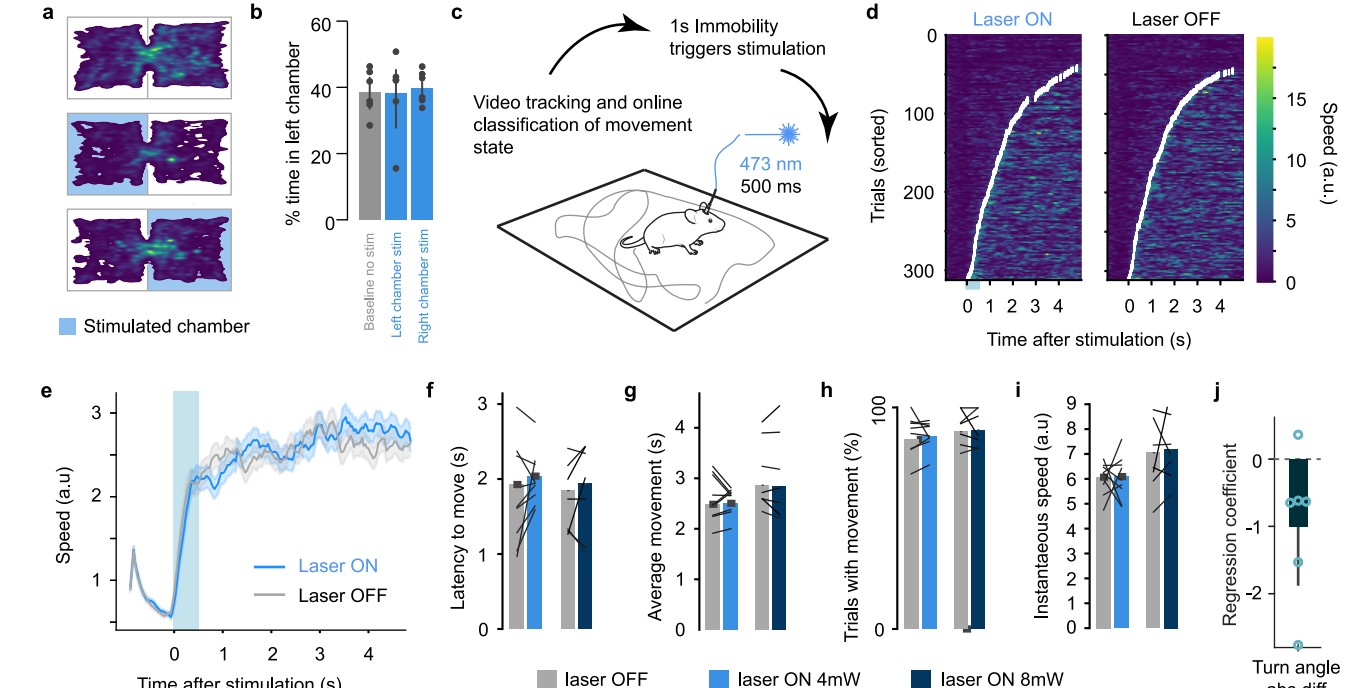

**Extended Data Fig. 10 | Motor effects of dopamine stimulation. a**, Heatmaps showing the occupancy for different sides of an arena during a real-time place-preference experiment where each side of the arena is paired with optogenetic stimulation of dopamine release in the TS. **b**, Bar graphs (mean) showing the percentage occupancy during control, stimulation in the left or in the right chamber. error bars represent SEM. **c**, Schematic showing the experimental closed-loop setup for detecting immobility and triggering dopamine release. **d**, Speed heatmap of trials sorted by movement onset for stimulated and control trials (n = 9 biologically independent animals). **e**, Speed histogram, same data as in n; error bars: SEM. **f, g, h, i**, Distribution, as means per mouse, of different movement parameters (see methods) (p:4 mW p = 0.21, 8 mW p = 0.42; q: 4 mW p = 0.67, 8 mW p = 0.78; r: 4 mW p = 0.87, 8 mW p = 0.80; s: 4 mW p = 0.93, 8 mW p = 0.50; paired two-sided t-test); same animals as in n. **j**, Regression coefficients for effect of TS dopamine response on the current trial on difference between turn angle on current and subsequent trial. Filled circles represent significant correlations for individual mice (n = 6 mice). Stats: one-sample t-test against zero, p = 0.07, Cohen's d = −0.94. Error bars represent 95% confidence interval.

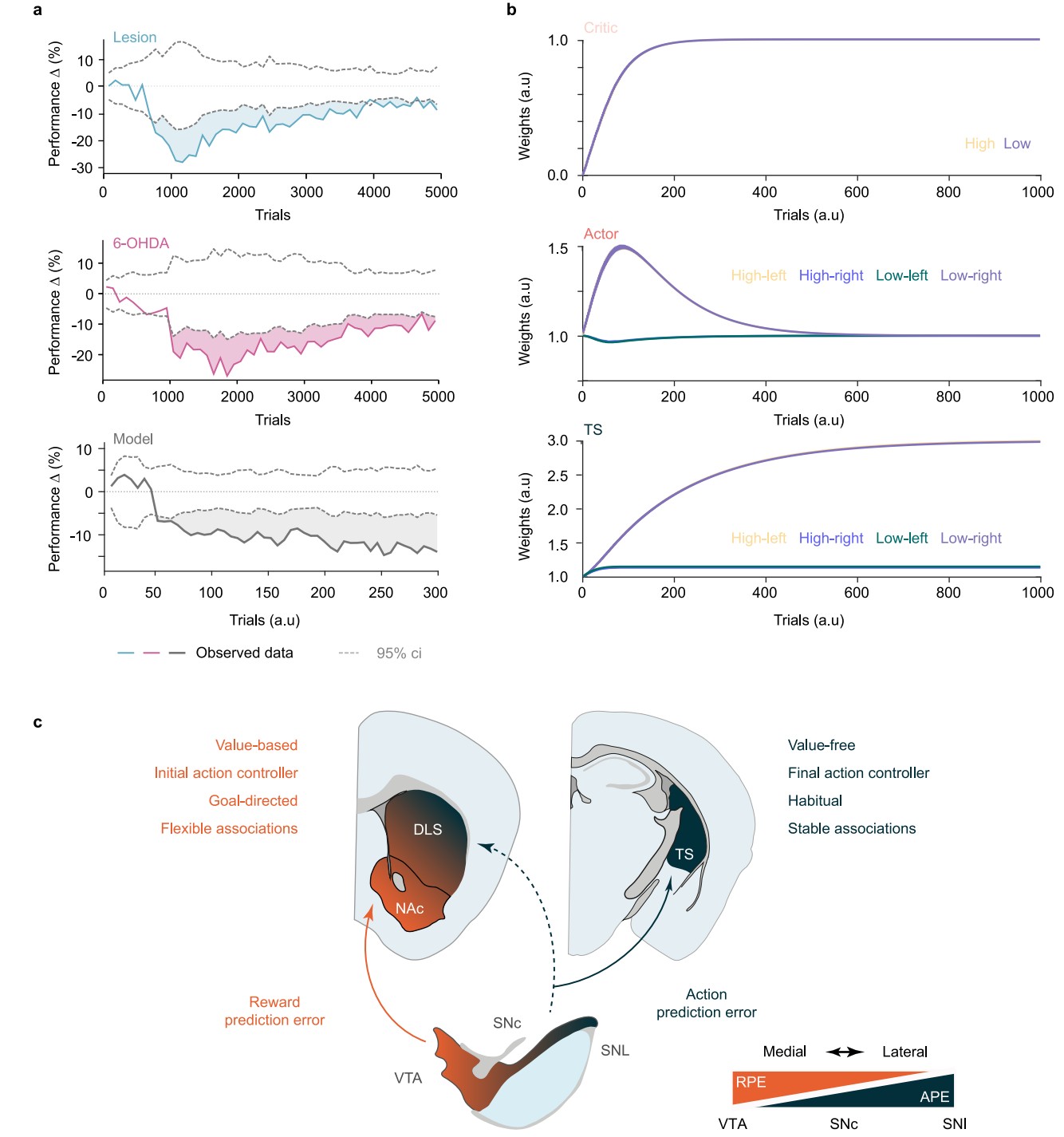

**Extended Data Fig. 11 | Network model parameters and summary. a**, Moment in learning in which differences in between groups become significant, for the behavioral data and the network model. Dotted lines indicate the 95% confidence interval for the shuffled data (see methods). **b**, Change of the model weights during learning, as means for 100 agents, for the Critic, Actor, and TS networks. **c**, Summary schematic of the anatomical representation for the dual-controller model. Dotted arrow indicates the proposed, but as yet unverified, broadcast of APEs to the dorsal striatum.

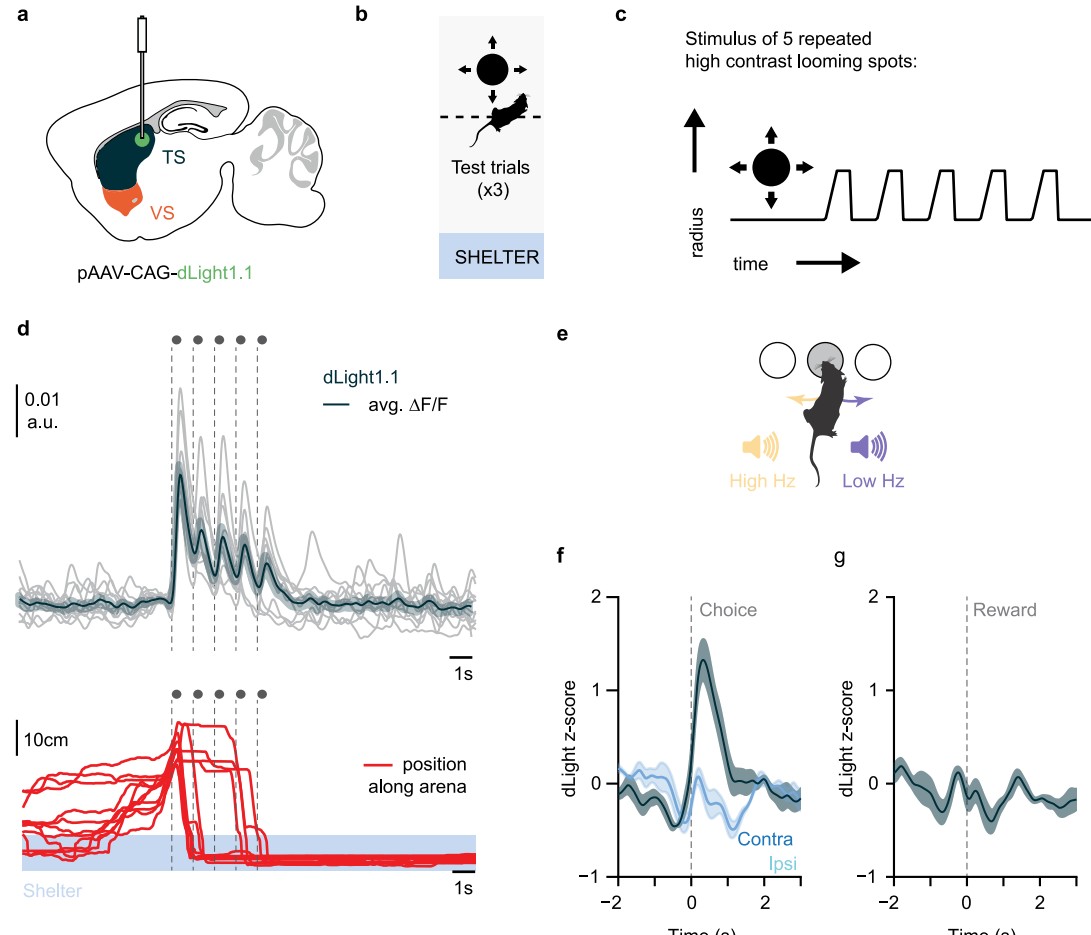

**Extended Data Fig. 12 | Response of TS dopamine to threat and movement.**
**a**, Schematic of the experimental approach for recording dopamine activity in TS. **b**, Schematic of the experimental arena containing a threat zone (indicated by a dashed line) above which a looming stimulus can be displayed and a shelter (blue shaded area). **c**, Schematic of a single trial of the looming stimulus, consisting of 5 consecutive expanding spots. Black line illustrates how the looming spot radius changes over time. **d**, The dlight1.1 photometry traces for all three trials for each mouse (gray lines) and the average (blue line) in response to the five looming stimuli (top). The position tracks for each trial of the mice along the long axis of the behavioral arena (bottom). Looming spot onsets (black, dashed lines) and their durations (black circles) are shown. (n = 4 mice). **e**, Schematic of the frequency discrimination task. **f**, Average photometry response of the same mice as shown in (D) in the frequency discrimination task aligned to choice (n = 4 mice). **g**, Same as F aligned to reward. All error bars represent SEM.

# Reporting Summary

## Statistics

For all statistical analyses, confirm that the following items are present in the figure legend, table legend, main text, or Methods section.

| n/a | Confirmed | |
|---|---|---|
| ☐ | ☒ | The exact sample size (*n*) for each experimental group/condition, given as a discrete number and unit of measurement |
| ☐ | ☒ | A statement on whether measurements were taken from distinct samples or whether the same sample was measured repeatedly |
| ☐ | ☒ | The statistical test(s) used AND whether they are one- or two-sided<br>*Only common tests should be described solely by name; describe more complex techniques in the Methods section.* |
| ☐ | ☒ | A description of all covariates tested |
| ☐ | ☒ | A description of any assumptions or corrections, such as tests of normality and adjustment for multiple comparisons |
| ☐ | ☒ | A full description of the statistical parameters including central tendency (e.g. means) or other basic estimates (e.g. regression coefficient) AND variation (e.g. standard deviation) or associated estimates of uncertainty (e.g. confidence intervals) |
| ☐ | ☒ | For null hypothesis testing, the test statistic (e.g. *F*, *t*, *r*) with confidence intervals, effect sizes, degrees of freedom and *P* value noted<br>*Give P values as exact values whenever suitable.* |
| ☒ | ☐ | For Bayesian analysis, information on the choice of priors and Markov chain Monte Carlo settings |
| ☒ | ☐ | For hierarchical and complex designs, identification of the appropriate level for tests and full reporting of outcomes |
| ☐ | ☒ | Estimates of effect sizes (e.g. Cohen's *d*, Pearson's *r*), indicating how they were calculated |

*Our web collection on statistics for biologists contains articles on many of the points above.*

## Software and code

Policy information about availability of computer code

| Data collection | Custom Matlab and python code was used to run experimental paradigms. |
|---|---|
| Data analysis | Custom Matlab and python code was used for data analysis. The analysis code is available at https://doi.org/10.5281/zenodo.15103777. For analysis the following version numbers were used<br><br>Photometry data analysis: Python 3.8, numpy==1.21.5, matplotlib==3.5.1, seaborn==0.11.2, scipy==1.7.3 tqdm==4.63.0, pandas==1.4.1, scikit-learn==1.0.2, cycler==0.11.0, scikit-image==0.19.3, nptdms==1.6.2, peakutils==1.3.3, statsmodels==0.13.5, cmocean==4.0.3, openpyxl==3.0.9.<br><br>Modeling: Python 3.7, numpy==1.19.0, pandas==1.0.5, matplotlib==3.2.2, tqdm==4.50.0, statsmodels==0.13.5<br><br>Behavioural and manipulation data analysis: matplotlib==3.5.2, numpy==1.23.1, pandas==1.4.3, scikit-learn==1.1.2, scipy==1.9.0, seaborn==0.11.2 |

For manuscripts utilizing custom algorithms or software that are central to the research but not yet described in published literature, software must be made available to editors and reviewers. We strongly encourage code deposition in a community repository (e.g. GitHub). See the Nature Portfolio guidelines for submitting code & software for further information.

## Data

Policy information about [availability of data](availability of data)

All manuscripts must include a [data availability statement](data availability statement). This statement should provide the following information, where applicable:

- Accession codes, unique identifiers, or web links for publicly available datasets
- A description of any restrictions on data availability
- For clinical datasets or third party data, please ensure that the statement adheres to our [policy](policy)

> All of the datasets needed to reproduce the findings in the paper are available at, https://rdr.ucl.ac.uk/projects/
> Dopaminergic_action_prediction_errors_serve_as_a_value-free_teaching_signal/240596.
>
> In addition Allen Brain Atlas connectivity data was used to demarcate the cortico-striatal projection patterns from AUDp (experiments, 100149109, 116903230, 120491896 and 146858006) and S1(SSp) (experiments 112882565, 114290938 and 126908007).

## Research involving human participants, their data, or biological material

Policy information about studies with [human participants or human data](human participants or human data). See also policy information about [sex, gender (identity/presentation), and sexual orientation](sex, gender (identity/presentation), and sexual orientation) and [race, ethnicity and racism](race, ethnicity and racism).

| | |
|---|---|
| Reporting on sex and gender | N/A |
| Reporting on race, ethnicity, or other socially relevant groupings | N/A |
| Population characteristics | N/A |
| Recruitment | N/A |
| Ethics oversight | N/A |

Note that full information on the approval of the study protocol must also be provided in the manuscript.

# Field-specific reporting

Please select the one below that is the best fit for your research. If you are not sure, read the appropriate sections before making your selection.

☒ Life sciences        ☐ Behavioural & social sciences        ☐ Ecological, evolutionary & environmental sciences

For a reference copy of the document with all sections, see [nature.com/documents/nr-reporting-summary-flat.pdf](nature.com/documents/nr-reporting-summary-flat.pdf)

# Life sciences study design

All studies must disclose on these points even when the disclosure is negative.

| | |
|---|---|
| Sample size | Sample size calculations were not performed but preliminary experiments were performed to gauge variation and help guide the choice of sample size. |
| Data exclusions | One experimental animal was excluded from the dopamine cell ablation experiment as the lesion quantification showed no ablation (this mouse performance was comparable to controls). Mice that were targeted for TS recordings were excluded if serial two-photon microscopy confirmed that the fibers were located outside of the TS (n=8). |
| Replication | None of the findings were explicity replicated. |
| Randomization | Cage mates were assigned to experimental and control groups. This was done without explicit randomization procedures. |
| Blinding | Experimenters were not blind to the whether mice were in the control or manipulation groups. |

# Reporting for specific materials, systems and methods

We require information from authors about some types of materials, experimental systems and methods used in many studies. Here, indicate whether each material, system or method listed is relevant to your study. If you are not sure if a list item applies to your research, read the appropriate section before selecting a response.

## Materials & experimental systems

| n/a | Involved in the study |
|---|---|
| ☐ | ☒ Antibodies |
| ☒ | ☐ Eukaryotic cell lines |
| ☒ | ☐ Palaeontology and archaeology |
| ☐ | ☒ Animals and other organisms |
| ☒ | ☐ Clinical data |
| ☒ | ☐ Dual use research of concern |
| ☒ | ☐ Plants |

## Methods

| n/a | Involved in the study |
|---|---|
| ☒ | ☐ ChIP-seq |
| ☒ | ☐ Flow cytometry |
| ☒ | ☐ MRI-based neuroimaging |

# Antibodies

| | |
|---|---|
| Antibodies used | NeuN 1:1000 (abcam, ab104225), tyrosine hydroxylase (TH) 1:1000 (Sigma-Aldrich, AB152), GFP 1:1000 (Aves labs, GFP-1020), Alexa-488 anti-mouse 1:500 (Invitrogen, AB_2534069), Alexa-567 anti-chicken 1:500 (Invitrogen, AB_2535858), and Alexa-647 anti-rabbit 1:500 (Invitrogen, AB_2535813). |
| Validation | ab104225: According to the manufacturer antibody specificity was analyzed by western blot analysis, where it specifically detects NeuN (UniProt ID: A6NFN3; Molecular weight: 34kDa). It has been cited in over 245 publications.<br>AB152: According to the manufacturer the antibody selectively labels in a western blot analysis a single band at approximately 62kDa (reduced) corresponding to Tyrosine Hydroxylase.It has been cited in over 100 publications.<br>GFP-1020: cited in 2347 publications and tested according to the manufacturer with western blot analysis. |

# Animals and other research organisms

Policy information about studies involving animals; ARRIVE guidelines recommended for reporting animal research, and Sex and Gender in Research

| | |
|---|---|
| Laboratory animals | Male and female adult mice between the ages of 2-7 months from the following mouse lines were used: C57BL/6J wild-type (Charles River), Drd1-Cre (Gensat: EY262), Adora2a-Cre (Gensat: KG139), DAT-Cre (JAX Stock No: 006660), Ai14 (tdTomato, JAX Stock No: 007914), Ai35 (Arch-GFP, JAX Stock No: 012735) and Ai32 (channelrhodopsin-2/EYFP, JAX Stock No: 024109) |
| Wild animals | N/A |
| Reporting on sex | Male and female adult mice were used in all experiments |
| Field-collected samples | N/A |
| Ethics oversight | All experiments were performed in accordance with the UK Home Office regulations Animal (Scientific Procedures) Act 1986 and the Animal Welfare and Ethical Review Body (AWERB). |

Note that full information on the approval of the study protocol must also be provided in the manuscript.

# Plants

| | |
|---|---|
| Seed stocks | *Report on the source of all seed stocks or other plant material used. If applicable, state the seed stock centre and catalogue number. If plant specimens were collected from the field, describe the collection location, date and sampling procedures.* |
| Novel plant genotypes | *Describe the methods by which all novel plant genotypes were produced. This includes those generated by transgenic approaches, gene editing, chemical/radiation-based mutagenesis and hybridization. For transgenic lines, describe the transformation method, the number of independent lines analyzed and the generation upon which experiments were performed. For gene-edited lines, describe the editor used, the endogenous sequence targeted for editing, the targeting guide RNA sequence (if applicable) and how the editor was applied.* |
| Authentication | *Describe any authentication procedures for each seed stock used or novel genotype generated. Describe any experiments used to assess the effect of a mutation and, where applicable, how potential secondary effects (e.g. second site T-DNA insertions, mosiacism, off-target gene editing) were examined.* |

