## [Peer Review file · Nature]

Dopaminergic action prediction errors serve as a value-free teaching signal

Corresponding Author: Dr Marcus Stephenson-Jones

Version 0:

Reviewer comments:

Referee #1

(Remarks to the Author)

Greenstreet and colleagues present a timely study concerning the potential role of an exciting division of the striatum and dopamine in the representation of a suggested but hitherto unrealized type of prediction error. The logic and writing is succinct and clear. Their experimental work and controls are remarkably thorough and well analyzed, and in some places, I'd recommend considering paring down the breadth or depth in favor of a more succinct story. There is this, as well as some other areas that I would recommend improving, but this is a solid manuscript on an exciting topic.

In the abstract, the authors seem to downplay the discovery of the 'hypothesized APE'. I'd encourage them to make a 'bigger deal' from the onset about establishing the neural correlates of such a mechanism.

6ohda elicits what appears to be have a stronger effect than lesioning the presumptive seat of APE, the TS. The TS lesion also appears to affect more striatal tissue than the missing DA terminals from 6ohda. I can think of several reasons for this differences if true, including that APE may be somewhat distributed like many striatal processes (e.g. the roles of the DMS/DLS) or a prolonged period in which the mice appear to not know what is going on - but I would ask for the authors to address this themselves.

I'd also ask that the y scale be the same for fig 1h and i.

In 2e, there appears to be an early suppression of DA activity for ipsi movements. Although smaller, it appears significantly different from contra. This is very interesting with respect to how the representation and use of the APE might play out. Could the authors provide any additional insights?

As a mouse finished the trial, it is in a context/state in which it should chose to return to the center port. I'm sure this doesn't happen all the time, which would make an interesting test of their results - providing an ipsi, contra, and null movement to cue to. If correct, we would assume a larger response following an ipsi port entry - specifically a contra movement back to center. Could the authors please provide these data? I recognize that this is similar to Ext Data Fig 6, but critically, it presents a specific context but removes the tone (and context may be sufficient even though the TS receives auditory input). If the authors disagree with my rationale here, I welcome a rebuttal that would help me - and potentially other readers - to more thoroughly understand how their proposed mechanism should work.

Related to the above point is a general question that persisted for me throughout the paper: why should APE exist in this select and somewhat nebulous part of the striatum? While it does receive sensory input, there are many other parts with sensory inputs, including to some extent, the DMS.

I feel that one of the missing components from the study is an assessment of trial-by-trial kinematics. While there is some discussion of learning effects and angle (Ext Data 6), it would seem - especially from their model and how the authors conceive of APE - that not only should APE reflect what *has* happened, but that trial-by-trial kinematics should change in line with APE. This is maybe a the biggest short-coming I found in an otherwise fairly solid manuscript. This also seems fairly easy enough to test.

The model can likely be improved in several ways. It is unclear why the authors selected an actor-critic model was selected compared to others, especially as there is very little anatomy offered other than the TS. The results are fine for the most part,

but more detail would greatly help the potency of this figure and the points it is attempting to make. It seems the key part is the improvement of having value based and value free - and the striking similarity of this plot to those in Fig 1. I'd encourage the authors to make it clear what the additional contribution is of the other portions of the plot / text. There is some, but I found it somewhat muddled.

Minor:

I would delete the first "repeated" in the first sentence of the introduction. The wording as is comes off a bit confused, and it is still accurate without including this qualifier.

I don't feel strongly, but invoking 'DA = RPE' in the second sentence of the introduction is not necessary and may throw off the reader - if not now, in years to come as our understanding of DA's encoding of RPE matures. Said a different way, although it is critical to mention DA's apparent role in RPE, the paper does not suffer and might benefit from holding off on mentioning DA in the second paragraph, as is currently done.

While there are only so many colors, choosing 2 hues of green for D1-Arch and A2a Arch may improve clarity, especially as blue often is used to designate Chr2 in opto experiments.

Could the authors please explain what the SEMs in 1f are? If these are the bias per session, why is it more than just a single value?

"In agreement with previous reports¹⁹⁻²², TS dopamine responses could also be elicited..." I don't see what this has to do with the current study. Is this just because TS is only more recently come to be appreciated? Or that readers would doubt your ability to target Ts after having shown it in Fig 1? Neither seem to be a good reason. Slightly better would be to take on the TPE hypothesis as the authors do in the Discussion. I'd ask the authors to include some rationale or take-home to help contextualize these findings if this is kept. If it is to contrast with TPE, I'd move this all to Discussion, potentially Ext Data (it is certainly related to the main thrust of the manuscript but isn't totally required). I'd also ask if TPE might just be posed as a select form of APE? I don't know if this is reasonable, but it could benefit everyone. Along these lines, I felt the discussion in several places could benefit from inclusion of the Yittri and Dudman 2018 perspective or other work that points to the conserved computational/functional anatomy across the striatum serving whatever need the inputs indicate.

2c - Approximate depth would be good to have, and if angled, please indicate. It is implied in 2b, but its inclusion wouldn't hurt. This is also helpful as Fig 5g appears to place TS and VS almost above/below each other rather than in front/behind.

2g - $y=0$ looks to be of a different font or font size? In my experience, 6 figures is highly-irregular for this journal. I might suggest combining substantial portions of fig 4 and 5.

Pale text - high sound in fig 5,6 and fig 6b,c - is difficult to read for my eyes.

While of potentially wide-interest, the section on habit in the Discussion could be moved to a Review piece and the conclusions removed if needed for length concerns. Similarly, while I appreciate naming the limitations, a more concise direct integration of these caveats could be done. I also don't see the need to outline 'future work' in a manuscript like this, but these are more for the Editor to determine.

While I also favor the correct spelling of behaviour as is done in Methods, consistency is key.

Referee #2

(Remarks to the Author)

This study examined the function of dopamine in the tail of the striatum (TS) using an auditory discrimination task (cloud of tones [CoT] task) in mice. The authors first show that the TS is required for learning and performance in this task. The authors then performed a series of experiments using fiber photometry and optogenetic manipulations, and conclude that dopamine in the TS signals action prediction errors (APEs), or "the difference between the action that is taken and the extent an action was predicted in a given state" to strengthen cue-action association in a value free manner.

The idea that dopamine in the TS signals APEs is potentially very interesting. The data are of high quality and the manuscript is well-written. The idea that TS dopamine is related to contralateral movement is relatively convincing (but see below). However, there are several substantive issues, which are elaborated below. Importantly, some of the results contradict previous observations. After considering these issues, the main pieces of evidence supporting APE, as opposed to "value-less" movement-related signals, come from the two following results: (1) a decrease of dopamine signals over learning, and (2) A contralateral choice bias induced by optogenetic stimulation during movement. However, the first point is not necessarily specific for APE, and the second result has errors in the statistical analysis and presentation. The explanation and analysis of the APE model is also lacking from a computational perspective. Overall, the presented data do not strongly support the main conclusion that TS dopamine signals APEs.

Major issues:

1. A previous study by Chen et al. (2022, cited) characterized dopamine signals in a similar auditory discrimination task. Their task design, which separated the timing of the cue and movement, demonstrated that dopamine responses in TS are primarily driven by sensory cues. This contrasts with the findings of the present study. Since the present study's task was not

specifically designed to differentiate between stimulus- and movement-related responses, the conclusion is not very convincing. Additionally, there are other data that appear to contradict previous studies as well. For instance, while the present study argues against sensory- and novelty-related responses, multiple previous studies have now shown that TS dopamine responds to sensory stimuli and that novelty enhances these responses.

2. One of the important pieces of evidence supporting APE is a decrease of TS dopamine responses over learning. The decrease over learning has been reported previously (Menegas et al., 2017, cited) and is a straightforward prediction of stimulus novelty/salience (Extended Data Fig. 8). Thus, such a decrease is not necessarily strong evidence for APEs. Furthermore, the study lacks proper control experiments. To demonstrate such a decrease over a long time, the authors need some control experiments to show that the observed decrease is not due to an artifact, such as a decrease in the level of sensor expressions or bleaching.

3. A previous study (Lindsey and Litwin-Kumar, 2022, cited) has already discussed a decrease in movement-related dopamine signals over learning to support the concept of APEs although they used the term, "action surprise", and action surprise is defined differently from the APEs discussed in the present study. The authors should more clearly discuss the contributions made by Lindsey and Litwin-Kumar (2022) as well as the difference in the definitions.

4. Extended Data Fig. 6a-g. To demonstrate that TS dopamine signals are not related to sensory cues, the authors omitted the sound cue on a subset of trials, and showed that TS dopamine still responded during movement with similar amplitudes. While this appears to support the authors' point that these responses are not sensory-driven, it is puzzling from the point of view of APEs. As a cue would make the left or right movements more predictable in the task context, the absence of a cue would make a left or right movement more surprising, thus the APE should become larger. This result, thus, contradicts the authors' main point that TS dopamine signals APEs.

In the same vein, it seems that the most obvious prediction of APE is that (in expert animals) it should peak on error trials, when the animal's action is relatively "more surprising" given the state. (This might be especially true for error trials in which the evidence/fraction of tones was high.) The authors should test this prediction explicitly, as most of the predictions tested (e.g. white noise) are somewhat roundabout.

5. Stimulus novelty/salience could also very easily be state-dependent and value-free. In other words, I disagree with the modeling in Extended Data Fig. 8. First, White Noise could easily be more salient than Tone (Extended Data Fig. 8a), by virtue of either being experienced less often or because of intrinsic auditory characteristics. Second, if TS only cares about the cue itself and completely ignores the associated reward, then for novelty/salience you still might not expect any change upon value manipulation (Extended Data Fig. 8b). The assertion that novelty predicts no response as a function of cue value (Extended Data Fig. 8b, center) is simply guesswork on the authors' part; there is no reason to assume a (somewhat unpredictable) tone should be treated as 0% novel and ignored, even by expert animals. Between these two points, the novelty/salience predictions would exactly match those of APE in Extended Data Fig. 8. While novelty/salience wouldn't themselves predict everything about contralateral vs. ipsilateral movements, it is quite conceivable that TS dopamine could also play a motor role, and the combination of these two roles could give rise to all the results presented here. Similarly, the significant regression coefficient on dopamine response size for TS (Fig. 5h,j) could also reflect that cues that are more salient (for whatever reason) cause greater effects on subsequent behavior without being related to the chosen action at all. Also, this analysis is discussed in the main text as if it's an ANOVA, whereas in fact (as best as I can tell) dopamine response size, perceptual uncertainty, and their interaction are all fit independently. ("Stats: two-tailed p values for the t-stats of the fitted coefficients are reported.") This seems wrong, as perceptual uncertainty can (and probably does!) directly influence DA response size, and this will "leak into" the DA response size coefficients if they are regressed separately.

6. The claim that TS dopamine reinforces state-action and not state-outcome associations rests on a grave statistical error. In Fig. 5d, the authors compare real to shuffled data using a Wilcoxon rank-sum test, which is invalid. For a permutation test, the real summary statistic should be compared to a percentile of the null distribution of that summary statistic (e.g. 97.5 for a two-tailed $p < 0.05$). Using a rank-sum test for this purpose is treating each permutation as an independent sample of data, which it is not. (You can imagine increasing the number of shuffles, say, to 100,000, to the point that any effect, no matter the size or sign, would look "significant" on this metric without collecting more data, which tells you automatically that it is invalid.) The alternative to a permutation test here is to just do a Wilcoxon signed-rank test relative to zero (2-sided), which does not appear to be significant by eye. This is the most important panel of this figure; without it, the entire figure essentially falls apart. Indeed, I would argue that the causal effect of TS dopamine on trial-by-trial behavior is the linchpin of the entire paper, and without it, the case for believing in APE becomes significantly weaker.

7. Also in Figure 5, the authors should show all the psychometric curves, for all the sessions, for both VS and TS (Fig. 5c). (They should also show VS bias in Fig. 5f.) This is because strictly speaking, bias is not the only thing to (appear to) change, as performance maxes out well below chance (Fig. 5c). The authors choose to ignore this by fitting a separate `max_performance` parameter, but it is highly relevant to the conclusion. In particular, different `max_performance` is contrary to the APE model prediction (Extended Data Fig. 9d), a fact that the authors never remark on, and which suggests a more direct role in movement facilitation for TS dopamine than the authors' APE account.

8. The authors point out that APE is (conveniently) a scalar (although see below), allowing it to extend quite nicely to a large space of possible actions. However, the only actions examined in this paper are left and right. This is common in studies of the striatum, which is highly lateralized, and much of that lateralization is also reflected in dopamine activity (Parker et al., 2016; Tsutsui-Kimura et al., 2020). However, (1) it's not immediately clear why APE would only be broadcast to one

hemisphere, as presumably it could just be ignored in the ipsilateral hemisphere which is less active and (2) it should also be possible to detect APE in a less lateralized task, just by looking e.g. at sequence entropy (Markowitz et al., 2023). The failure to do so here is unfortunate, and leaves open the possibility that the signals have more to do with a permissive signal for gross movement, as opposed to APE per se.

Related to this point, the authors say, "This raises the possibility movement-related dopaminergic responses throughout the striatum 8-11, 13, 36, may reflect APE." They then cite a handful of studies, including Markowitz et al., 2023, which record dopamine in the DS as evidence for this claim. But in the authors' own model, the DS is receiving RPE as the actor, not APE! They cannot cite these studies in support of their conclusion; if anything, these studies run counter to it.

9. From the Methods, it is not at all clear how the model works under the hood. In Miller et al., 2019 (frequently cited as inspiration by the authors), ak is a "vector over actions in which all elements are zero except for the one corresponding to" a , and A is a 2D matrix indexed by (s, a) . Consistent with this, the update equation that the authors give for W_{tail} implies that the entire matrix is updated. However, elsewhere, the authors go out of their way to say that "the APE term need not be action-specific." Which one is it? If their model requires a vector-valued dopamine signal, then their fiber photometry-based recordings can't support it. Alternatively, if for biological reasons, the authors want to make it scalar-valued by only considering the cell corresponding to (s, a) , how does this impact the learning dynamics? In either case, please resolve this ambiguity by making it explicit in the equations.

There are several other issues with this section. (1) The critic is defined by a 2×2 matrix that maps sensory states to actions. But traditionally in RL, a critic ignores actions and is directly parameterized as $V(s)$. It's unclear how important this departure from standard RL is. (2) How can δ_A equal $\text{argmax}(\delta_A)$? It seems like this ought to be $a = \text{argmax}(\delta_A)$. (3) Is the role of the differential equations just to low-pass-filter the RPE and APE? If so, make that clearer, and explain why this should occur in the model itself (as opposed to just the mapping from RPE/APE onto dopamine). (4) In one part of the Methods, the authors say that APE is proportional to "the difference between the action taken ak and the stimulus action strengths given by the stimulus $A(s)$." Later, they seem to switch variables and say it's $a - \hat{a}$. Please be consistent between $A(s)$ and \hat{a} .

10. The authors suggest four putative advantages of a value-free system, but three of them don't seem to work. (1) Stably storing associations in a manner more robust to short-term fluctuations. But there's no reason why you couldn't just adjust the learning rate in your actor, rather than having a completely independent system; indeed, plenty of work has explored these adjustments empirically (e.g. Behrens et al., 2007; Rushworth & Behrens, 2008). (2) Policy complexity. But the mathematical definition of policy complexity explored by Gershman, 2020 is equivalent to the mutual information between states and actions. In other words, a highly deterministic mapping from (Low Hz \rightarrow LEFT) and (High Hz \rightarrow RIGHT) would have high, not low, policy complexity, contrary to the authors' assertion. (3) Value-free learning as a Bayesian prior. But the authors make no suggestion as to how the strength of the value-free learner might be modulated by uncertainty.

11. Why would the actor but not the critic weights decay, as they are both updated by RPE, which is decaying to zero? Furthermore, is the actor weight decay the only reason that the combined controller can outperform the value-based controller? If so, it seems like cheating. Does this explanation imply that if the rewards were delivered probabilistically, such that RPEs continue to occur, you wouldn't get the same transfer over to the TS? If so, the authors should probably test this prediction experimentally.

Minor points:

12. Extended Data Fig. 6k-n uses quartiles, which can mask the true effect size. It would be nice to show the raw data, even if only to supplement the quartiles.

13. Turn angle does appear to change over time, it's just not significant at the population level (Ext. Data Fig. 7h). However, if "filled circles represent significant correlations for individual mice," then it appears that 4/6 animals increase speed and 5/6 animals decrease turn angle significantly over training. As such, it seems misleading to claim that "Over the course of learning there was no significant trend in how either speed or turn angle changed (Extended Data Fig. 7f-h)." Why do they plot the dopamine response in Extended Data Fig. 7i but the dopamine residuals in Extended Data Fig. 7j? This doesn't seem like an apples-to-apples comparison. I'm assuming that for the multiple regression model, they predicted only dopamine response. Is this true?

14. There are some typos/omissions, e.g., "The identification of dopamine transients prior to movement initiation led to the idea that movement-related dopamine release may trigger the initiation of 8, 10, 12, 13." Also the Methods paragraph beginning with "For chronic D-AP5 infusion experiments" is in a different font. The letter σ is used to refer to both a ReLU function and a learning rate in the same section. There are inconsistent heading formats, etc.

Referee #3

(Remarks to the Author)

A. Summary of the key results

This manuscript shows that dopamine response in the tail of the striatum correspond with an action prediction error, which provides a mechanism for the value-free learning that has been proposed on theoretical grounds. The paper provides a

series of extensive experiments using a simple binary choice task that shows a clear dissociation between reward-related responses (in the ventral striatum) and action-related responses (in the TS).

B. Originality and significance: if not novel, please include reference

The results hold transformative potential for re-defining how we think about dopamine and reinforcement learning, providing anatomically distinct functional roles to dopamine in different parts of the striatum. The long-standing approach in the field (for almost 30 years) has been that dopamine encodes reward prediction errors. More recent work has challenged this view, but mostly along the lines that dopamine may encode more general prediction errors about outcomes (e.g., about identity) or even a broader distribution of outcomes. This work provides the first evidence for a different role of dopamine in learning—through action prediction errors in the tail of the striatum. Such action prediction errors have been proposed theoretically (in cited work) as being a key mechanism for the established and maintenance of learning. This paper provides a potential neural mechanism to this theoretical work. There is a nice parallel to the original RL-dopamine findings, which discovered a correspondence between the prediction supposed by TD learning and the behaviour of (some) dopamine neurons.

C. Data & methodology: validity of approach, quality of data, quality of presentation

1. The experiments together present a fairly strong set of evidence in favour of TS dopamine encoding action prediction errors. There is very strong evidence that the responses are not reward-related and strong evidence that the response are related to action selection. I am a little less convinced that these represent action prediction errors. One of the main pieces of evidence used in favor of this claim is the experiment that shows how these response decrease over the course of learning. While it may be true that APEs should decline over time, other things change over time as well, including likelihood of different actions, and perhaps more abstract features like action initiation costs.

To show an action prediction error would require one step further—showing that the dopamine response on any given trial is a function of the action chosen on previous trials. That's what defines a prediction error—current action minus predicted actions. This is precisely what is shown with reward prediction errors and what made that story so compelling. Here, that is not shown. The closest is with the white noise experiment, where removing any state-action connection increases responding in the TS. This issue might be resolvable with the existing data (and no further experiments), but would require looking at the recent history of choices and showing that the current action response is a function of those. It might require further research that explicitly manipulates the frequency of different choices and assesses the size of the action prediction errors.

2. One major issue is the sheer number of researcher degrees of freedom available in the data analysis pipeline. In other research areas with which I am familiar, this challenge would be handled through pre-registration, but that seems not to be attempted (or maybe not possible here). As a result, every analytic decision (even when justified) seems post-hoc, and there are no robustness tests to ensure that these choices are not causing the results. Were these choices all made data blind? As a few examples just from a few pages (but really it's pervasive in the paper): there is the picking the sessions with the best performance (p. 26), trimming of the photometry trace to maximize variance accounted for (p. 27), arbitrary cut of 1 s on return events (p. 27), and splitting data into larger and small at an arbitrary cuts (p. 29). These analysis decisions have limited justification, and, though perhaps each one could be justified, a convincing analysis would need to show that the results are not driven by post-hoc pipeline decisions to extract the best possible story out of the data.

3. In Figure 1H and elsewhere, I am not sure how this is a learning rate and not wholly co-extensive with max level? How well does the sigmoid formula used represent what animals actually did? Learning is often negatively accelerated (not sigmoid), and that also seems to be true in the current data. In addition, this approach assumes a smooth, continuous learning, which does seem apparent in group curves, but may not represent individual animals (not shown here). There is a lot of work on fitting learning curves, showing that they are often not smooth for individuals and better thought of as transitions and modelled with Weibull functions (e.g., Gallistel et al, 2004, in PNAS).

4. Fig 5. If dopamine in the VS is encoding an RPE, why would it be the case that responding on the next trial was not related to VS responding on the previous trial? An RPE should increase the value of that option and make it more likely to be selected on the next trial. This, of course, is one of the challenges of distinguishing the impacts of reward (value-based) against the impacts of repetition (value-free) and may need to a targeted experiment to disentangle.

5. The RL modeling was perhaps the least convincing part of the paper. First, the base model used a semi-Markov formalism that only appears in a (very) small handful of papers and certainly does not represent the standard way that RPEs are formalized. This is even acknowledged in the paper where the schematics (e.g., Figure 7) do not correspond to the model at hand. What justifies this choice of formalism, and would the results generalize to a more standard RL treatment of the RPEs? Beyond that, there are many details missing in the APE (value-free) component of the model, which would seem to be the most important (and novel) component. For example, there is no update equation for the action component and no details on the number of actions in the model, nor what counts as a transition for the semi-markov model. Without these details, it is very difficult to assess the functioning of the model, its expected behaviour and how well it explains the neural results. In addition, although the paper cites two pieces of theoretical work on APEs as the inspiration for the current work, those models are not directly used, and a novel formalism is adduced yet not directly compared to that prior work.

D. Appropriate use of statistics and treatment of uncertainties

- In Figure 5D, why are the statistical tests compared against a shuffled dataset, rather than simply against one another?
- Throughout, there are no effect sizes provided to go along with the inferential statistics

- The analysis on p. 29 that arbitrarily splits data into small/large at the 65th percentile throws away a lot of the data. A more appropriate analysis would seem to be a logistic regression with size as a continuous predictor.

E. Conclusions: robustness, validity, reliability
See comments elsewhere.

F. Suggested improvements: experiments, data for possible revision

See first comment in section C for suggested improvement with regards to proving that these responses represent an APE. I'd also like to see more consideration of the goal-directed vs habitual distinction (and connection with DMS/DLS work). Presumably, these action prediction errors are particularly important for (in)forming value-free habitual behaviour, yet they seem to get smaller as learning goes on. How do the proposed mechanisms fit with existing ideas about the emergence and maintenance of habitual behaviour?

G. References: appropriate credit to previous work?
Yes.

H. Clarity and context: lucidity of abstract/summary, appropriateness of abstract, introduction and conclusions

The text (and figures) were exceptionally dense, and it took multiple reads to parse the paper. One easy and major improvement would be to provide a little more methodological detail about the behavioural task in the main text. Simply referring to the "cloud of tones" task did not suffice, which made both deciphering the text difficult, but even all the figures which referred to "% of tones", rather than frequency. After reading the methods in detail, this makes more sense, but made the paper proper hard to understand.

- The section on the dual controller was particularly difficult to understand. The section seems to describe a network model (with weights etc), but the specification is such that I could not really understand the structure of the model, the different components, nor how it fit together.

- Small point: middle p.15 "negatively reinforce" is misused (it actually means to increase behaviour through the absence of a punishment).

Version 1:

Reviewer comments:

Referee #1

(Remarks to the Author)

The authors responded thoroughly to all my comments and appeared to do a commendable job with the excellent issues raised by the other Referees. I only offer a few minor points that the authors can address as they see fit.

The readability of some figure text is difficult due to the color and thickness, e.g. 3a 'late' (the text could just be in black, or if color coded, changed to be slightly darker... we'll get the point), Ext Data 9, etc.

I appreciate the shading used to determine significant difference. The authors may want to place these bars at the bottom of the plot, as the shading done in the current way could be confused by an unaware reader as implying the use of orange light stim. This is not required, just a suggestion.

I encourage the authors to expand beyond model-free and make further use of the growing appreciation of policy-based learning rather than value-based. Bennett, Niv, and Lagdon have an excellent review of this and other studies like Hodge et al (bioRxiv) provides corroborative evidence for the idea and the work here.

I applaud the use of the Frechet distance and its inclusion in the main document. As it is not a common metric in much of neuroscience, I might suggest a more thorough description of it in the methods or even results section.

The authors seem to indicate Yttir and Dudman studied DLS, but their work was in DMS if I recall.

Referee #2

(Remarks to the Author)

The authors have performed additional experiments and analyses to address my previous concerns. The optogenetic stimulation experiment was repeated using a stronger stimulation (8 mW instead of 4 mW). Although it remains to be determined whether the 8 mW stimulation condition is physiological, it produced a stronger effect and thus partially replicated the previously observed effect. Ideally, the authors should have also conducted a replication using the previous condition (i.e., 4 mW), but I do not consider it necessary to request that here. The experiment with 8 mW increases confidence in the result. The revised text also provides important information. This work presents an interesting set of results.

If further verified and refined, it could provide a novel framework for understanding diverse dopamine signals.

Referee #3

(Remarks to the Author)

The authors have done an excellent job of addressing my concerns about the previous version. I only have some minor suggestions to improve a further version. In particular, from my perspective, the key addition in this revision is the trial-to-trial analysis (now Fig 3), which was really the missing piece to establish this signal as an APE. The modelling work is also much better justified and explained now, though I still think the meatier contribution is the novel neural evidence for the previously theorized APE signal. Also, though adding further explanation about the analytic choices was helpful, I already thought each one was reasonable, as post-hoc analyses often seem to be. Showing the results are robust to other analytic choices was a much better counterpoint.

Here are the minor suggestions I have for improving this version of the manuscript:

- The little paragraph above Fig 3 seems to recap only one of the points in the previous section (and the less convincing one at that). The point about trial-to-trial changes in TS responding is omitted. Same with the caption to Fig 3 and the lead sentence of the next section.
- The new text describing the logistic regression needs to be stitched into the text better. Right now, it reads more like a response to the previous version (i.e., “without splitting the data” and “now a continuous predictor”), as though the reader had just gone through the old split analysis.
- p. 18. “Classically for instrumental learning” is a confusing use of the word “classically”, which of course evokes classical conditioning (not operant). Easy switch for a different word (typically, commonly, etc...).
- Not great that the photometry datasets are being withheld for future use (results are less verifiable and extensible), but the journal policies will dictate whether that's ok (which it shouldn't be).
- The Eiselt et al, 2021 paper that is used as source for the modified Weibull does not seem to appear in the references. The Gallistel paper I noted last time should probably get some credit here (it was their innovation).
- Minor quibble with the semi-markov justification: There are other ways to handle variable timing with standard TD/RL (e.g., the microstimulus model as discussed in the newer Sutton & Barto text which also appears as the base model in the Nambodiri papers). The complete serial compound idea was always recognized as a computationally convenient fiction.

We are grateful to the reviewers for their detailed reading of our manuscript and for their constructive comments. Based on the referees' feedback, we have substantially revised our manuscript. We provide a point-by-point response to the reviewers' concerns below, and relevant changes are highlighted in yellow in the manuscript. We believe the additional experiments, discussion and analysis all help solidify and support our original conclusions.

Referee #1 (Remarks to the Author):

Greenstreet and colleagues present a timely study concerning the potential role of an exciting division of the striatum and dopamine in the representation of a suggested but hitherto unrealized type of prediction error. The logic and writing is succinct and clear. Their experimental work and controls are remarkably thorough and well analyzed, and in some places, I'd recommend considering paring down the breadth or depth in favor of a more succinct story. There is this, as well as some other areas that I would recommend improving, but this is a solid manuscript on an exciting topic.

In the abstract, the authors seem to downplay the discovery of the 'hypothesized APE'. I'd encourage them to make a 'bigger deal' from the onset about establishing the neural correlates of such a mechanism.

We thank the reviewer for their suggestion. We agree with their sentiment that we could emphasize that we are the first to establish the neural correlates of APE. We still refrain from doing so because we think our title highlights the discovery sufficiently and we want to keep the narrative that highlights the theoretical work that inspired us.

6ohda elicits what appears to have a stronger effect than lesioning the presumptive seat of APE, the TS. The TS lesion also appears to affect more striatal tissue than the missing DA terminals from 6ohda. I can think of several reasons for this differences if true, including that APE may be somewhat distributed like many striatal processes (e.g. the roles of the DMS/DLS) or a prolonged period in which the mice appear to not know what is going on - but I would ask for the authors to address this themselves.

I'd also ask that the y scale be the same for fig 1h and i.

Thank you for these comments, they are important points and have implications for how the APE system is anatomically arranged. As the reviewer notes, ablating the DA cells that project to the TS shows slightly stronger effects in learning than ablating the TS itself. This can be appreciated in a larger difference between control and experimental mice along the learning curves (see Figure 1h and I). However, this is consistent with the histology, as the 6OHDA injections affect a larger striatal region than the caspase injections, see plot below.

Side by side quantification of the Caspase and 6OHDA experiments. Given the different quantifications methods (fluorescence ratio for 6OHDA vs area affected for Caspase), we have normalised the two curves to the minimum value in order to compare these two experiments directly. Note that in both cases this minimum value is absolute lack of projections or cells (and the max value is equal to controls).

This is probably due to the fact that DA cells that project to the TS also send collaterals to other regions of the striatum. We did experiments to quantify this and found that this is indeed the case: around 20% of cells in the SNc and SNI that project to the TS also project to the DLS (panel C below). This is consistent with the fact that around 40% dopamine neurons that project to the TS express Aldh1a1 (Panel E below). This subpopulation of dopamine neurons is activated in response to movement initiation and has been shown to project widely throughout the dorsal striatum (Azcorra et al., 2023). Future work will be needed to determine the molecular subpopulation of DA neurons that encode APE but if it is this movement related Aldh1a1+ population, then as the reviewer points out, APE would be present throughout the dorsal striatum, and not exclusively confined to the TS. This would explain the differences observed in the two ablation experiments. We have clarified this point in our discussion (relation to other theories section) and added an additional arrow to our final schematic (Extended Data Figure 9c) to illustrate that we believe that APE will be broadcast to other areas of the striatum beyond the TS.

A) Injections of Cholera toxin subunit B (CTB) into the TS (Cyan, ctb-Alexa Fluor 647) and DLS (Green, ctb-Alexa Fluor 488) into a DAT-cre/Ai14 tdtomato reporter mouse line. B) Percentage of TS projecting DA cells located in the SNL and SNc. C) Percentage of TS projecting DA cells that also project to the DLS (triple labeled cells). D) Antibody labeling for Aldh1a1, in DAT-cre/Ai14 mice with TS projecting cells labeled from CTB-647 injections into the TS. E) Quantification of the percentage of TS projecting DA cells that express different molecular markers. The data for the VGlut2 and Sox6 experiments were collected in similar antibody labelling experiments shown in panel D. F) The quantification of the number of DAT/Ai14+ cells that were also double labelled from dual VGlut2 and Aldh1a1 staining.

We have also changed the axis of Figure 1h,i and 1l,m so they are plotted on the same scale.

In 2e, there appears to be an early suppression of DA activity for ipsi movements. Although smaller, it appears significantly different from contra. This is very interesting with respect to how the representation and use of the APE might play out. Could the authors provide any additional insights?

We appreciate the reviewer's observation. In response to this we tested whether the observed dip in DA activity for ipsilateral movements was statistically significant, and it was not. To clarify this in the paper, we have added shaded regions to Figure 2e to indicate periods of significance between the traces. These shaded time windows represent significant differences between the two trace types in each subplot. The significance was determined by conducting two-sample t-tests on 0.1-second bins, with a p-value threshold for significance set at 0.01, consistent with our other analyses and the methods described in Parker et al. (2016).

Average photometry traces in TS (n=10 mice) and VS (n=7 mice) aligned to task events. Shaded time windows show significant differences between the two trace types in each subplot, calculated by performing two-sample t-tests on 0.1 s bins and a p-value threshold for significance of 0.01.

As a mouse finished the trial, it is in a context/state in which it should chose to return to the center port. I'm sure this doesn't happen all the time, which would make an interesting test of their results - providing an ipsi, contra, and null movement to cue to. If correct, we would assume a larger response following an ipsi port entry - specifically a contra movement back to center. Could the authors please provide these data? I recognize that this is similar to Ext Data Fig 6, but critically, it presents a specific context but removes the tone (and context may be sufficient even though the TS receives auditory input).

If the authors disagree with my rationale here, I welcome a rebuttal that would help me - and potentially other readers - to more thoroughly understand how their proposed mechanism should work.

We agree with the reviewer that this time point provides an interesting test for our model. When we analyze this time point we see as the model predicts there is a significant difference when the mice make contralateral vs ipsilateral returns from the side ports to the center port despite the lack of tone, this data was previously included and is reported in (Ext Data Fig 4j-l). As the reviewer notes we think this is good additional evidence that the APE signal is related to movement initiation and not the tone. In these expert mice there are too few trials when the animals do not directly return to the center ports, less than 5%. Due to this we do not have enough data to align to the null movement.

While we can't align to null movements in expert mice we do have another experimental data set where we recorded at the very start of learning, in the first recording sessions mice make very few returns directly to the center port after visiting the side ports (Extended Data Figure 5s,t). In this data we adapted our task to play one of the tones 20ms after mice leave the side ports, while the other tone is still played at the center port. Recordings of the TS dopamine response in this session are now included in the manuscript (aligned to cue, Extended Data Figure 5s) to address a different point related to novelty but we mention it here as you can still see what the reviewer was asking. In the plot below two things are clear, the first is the reason we include these plots in the manuscript: there are no differences between the traces on trials where sounds are played at the side ports or not, despite this being the first session in which the mice are exposed to the tones. This again rules out that there is any response to the tones or that this is a novelty response to the cues when they are played in the task. Secondly in this data where mice make fewer direct returns to the center port, we see far less separation of the traces between when mice would make an ipsilateral and contralateral return to the center port. Indeed, there is no significant difference in the signal between ipsilateral and contralateral returns in this data.

Average dopamine response aligned to leaving the side ports during the first recordings sessions (n=6 mice). Contralateral and ipsilateral returns are plotted separately and returns without concomitant cue are also shown.

We apologize to the reviewer that we do not have a good dataset where we can align the return movements to all three conditions and hope this addresses their request.

Related to the above point is a general question that persisted for me throughout the paper: why should APE exist in this select and somewhat nebulous part of the striatum? While it does receive sensory input, there are many other parts with sensory inputs, including to some extent, the DMS.

Thank you for pointing this out. It is an important point; we don't think that APE only exists in the projections to the TS and have attempted to better clarify this in the discussion. We think

that all movement related dopamine signals may reflect APE. We have expanded our text in the “*All movement-related dopamine signals may encode action prediction errors*” section of the discussion to better highlight this.

To better highlight what we think, we have also added a dotted arrow to our anatomical summary figure (Extended Data Figure 9), showing that APE may be broadcast to the dorsal striatum as well as the TS.”

We think that the regions that receive the most movement-related dopamine signals and the least reward-related signals, i.e. the DLS and the TS are special in that they have both been shown to control habitual behaviour. Our long-term hypothesis is that habitual behaviours are updated by APE's and goal-directed behaviours are updated by RPE's. In this sense it is interesting that the pDMS also receives sensory information because this region is proposed to control goal-directed behaviour/learning (e.g. Hart et al., 2018). This would make it an excellent candidate for the value-based actor in our model. Recently the pDMS has also been shown to receive reward-related dopamine signals (e.g. Hart et al., 2024, Reinhold et al., 2023). So as with our model the sensory “state” information is passed to two striatal regions that differentially receive reward or movement-related DA teaching signals. We believe this should be discussed further but we think that a review article would be a better place to expand on this.

I feel that one of the missing components from the study is an assessment of trial-by-trial kinematics. While there is some discussion of learning effects and angle (Ext Data 6), it would seem - especially from their model and how the authors conceive of APE - that not only should APE reflect what *has* happened, but that trial-by-trial kinematics should change in line with APE. This is maybe the biggest short-coming I found in an otherwise fairly solid manuscript. This also seems fairly easy enough to test.

We thank the reviewer for this very important suggestion to analyze the effect of APE on trial-by-trial kinematics. As the reviewer notes it is likely that the size of the APE on the current trial should influence kinematics on the subsequent trial. To test this we performed a linear regression between the current trial TS dopamine response at time of choice and the Frechet distance between trajectories on the current and subsequent trial (which provides a measure of the similarities between the two trajectories, with a low value indicating high similarity between the two trajectories). We found a significant negative correlation between the size of the TS dopamine response and the Frechet distance between the current and subsequent trajectories (Coef = -3.04 Pval = 0.03), indicating that the larger the APE is on the current trial the more similar the trajectory is on the subsequent trial. This is consistent with our model, as APE would promote the animal to essentially repeat what it has done in the past. We are very grateful to the reviewer for this suggestion. We now report this as a main finding in Fig 5l,m.

We also find the same trend when we perform the same regression but use the similarity in turn angle between trials. This trend is now reported in Extended Data Fig 8t. This was added to be consistent with the other parts of the manuscript where we use turn angle but it should

be noted that it captures less information about the similarities of the trajectories in this analysis than the Frechet distance.

We thank the reviewer again for this suggestion, as it has improved our insights into the behavioral relevance of APE.

The model can likely be improved in several ways. It is unclear why the authors selected an actor-critic model was selected compared to others, especially as there is very little anatomy offered other than the TS. The results are fine for the most part, but more detail would greatly help the potency of this figure and the points it is attempting to make. It seems the key part is the improvement of having value based and value free - and the striking similarity of this plot to those in Fig 1. I'd encourage the authors to make it clear what the additional contribution is of the other portions of the plot / text. There is some, but I found it somewhat muddled.

The reviewer makes a very valid point that there are alternative ways that we could have constructed our value-based controller. Actor-critic models, while popular, are not the only way that value-based basal ganglia circuitry may be arranged. We chose to use an actor-critic model because we believe it offers the most parsimonious explanation for our data and others. This was not well motivated in the text and we have now added an explanation to the results section that helps clarify our rationale.

To expand further on our rationale we specifically wanted our model to be able to capture the shift in behavioural control from the value-based to the value-free controller, while retaining the predicted value of the state. Previous models have used an arbitration mechanism to explicitly shift which control system an agent uses (Miller et al., 2019, Bogacz et al., 2020). While this is a common approach there is a host of experimental evidence that the activity in the dorsal medial striatum, which controls initial flexible learning (e.g. Hart et al., 2018, Kim & Hikosaka, 2013), is not active nor has an influence on behaviour when tasks are stably performed. This is consistent with a decay in the weights of the inputs that drive the DMS. Indeed, this has been beautifully demonstrated in a study by Yttri & Dudman in 2016, in this study they could influence the policy a mouse took in lever pressing task by optogenetically manipulating the principal neurons in the DMS. The effects of the optogenetic manipulation persisted after the optogenetic stimulation was stopped, suggesting that the policy had been reinforced but crucially for our point the induced changes slowly decayed back to baseline in the absence of any additional reinforcement. Again, this is consistent with the weights in the DMS decaying in the absence of continual reinforcement. While there is evidence that the weights in the DMS (one candidate for the value-based actor in our task) decay, there is no evidence that the predicted value of state information decays in the same way. Indeed, when we recorded the "RPE" signal over the course of training the cue response remained prominent and the reward response suppressed even in expert mice when the TS was controlling behaviour, this is similar to what others have also observed (e.g. Schultz et al., 1997, Amo et al., 2022). Taken together this suggests that animals stably retain the predicted

value of the state, potentially in the NAc, but do not stably retain the value-based policy information in the DMS. This suggests that animals have separate systems for retaining these two types of information and this separation is exactly what occurs in actor-critic models. Because of the need in our data to explain the differential retention of these different types of information we chose to model our results with an actor-critic system, which is the most common model that we know where the state-value and state-action associations are separated.

We have also clarified our discussion/rationale for the additional plots in Figure 6. In brief, we think that 6d,e are needed to show that our combined value-based/value-free model predicts that each of the controllers will drive behavior at different stages of learning. Figures 6e-g are needed to experimentally test the model predictions that the TS does not control behavioural choice early in learning and that the TS control of behaviour slowly develops as the animals become experts at the task. Figure 6h is needed to show that our model can replicate the frequency specific changes in corticostriatal plasticity that have been observed (Xiong et al., 2015).

Minor:

I would delete the first "repeated" in the first sentence of the introduction. The wording as is comes off a bit confused, and it is still accurate without including this qualifier.

Thank you, we have changed this and it reads better.

I don't feel strongly, but invoking 'DA = RPE' in the second sentence of the introduction is not necessary and may throw off the reader - if not now, in years to come as our understanding of DA's encoding of RPE matures. Said a different way, although it is critical to mention DA's apparent role in RPE, the paper does not suffer and might benefit from holding off on mentioning DA in the second paragraph, as is currently done.

It's a good point that the understanding of "RPE" coding is rapidly changing. We are aware of the current debates about what value-coding DA neurons may encode and have cited recent alternative interpretations. Having thought about it, we still think that evoking RPE early on is the easiest way to lead the majority of readers into what we think updates the different choice strategies.

While there are only so many colors, choosing 2 hues of green for D1-Arch and A2a Arch may improve clarity, especially as blue often is used to designate ChR2 in opto experiments.

We have adopted this suggestion to reduce the chance of confusion. The changes have been made to Figures 1 and 6.

Could the authors please explain what the SEMs in 1f are? If these are the bias per session, why is it more than just a single value?

We apologize for the confusion here. In that panel, each dot represents the bias between the opto-ON and opto-OFF choices for one session. To calculate the significance of this effect (which depends on factors such as the total number of trials of each type and the total number of trials with optostimulation in each trial type), we generated, for each session, 1000 shuffles, matching, for each trial type, the number of optostimulated trials. Choices were drawn from the unstimulated trials. This generated a baseline distribution of port choice proportions and reflects the natural variability in the potential choices for each trial type. The error bars behind each dot represent the standard deviation (and not SEMs) of these distributions. We have now calculated the significance values for each session (by assessing if the distributions were different than 0, $p < 0.05$), and report this in the figure with different colors for each dot. We have now further clarified this approach both in the figure legend and in the methods.

"In agreement with previous reports¹⁹⁻²², TS dopamine responses could also be elicited..." I don't see what this has to do with the current study. Is this just because TS is only more recently come to be appreciated? Or that readers would doubt your ability to target Ts after having shown it in Fig 1? Neither seem to be a good reason. Slightly better would be to take on the TPE hypothesis as the authors do in the Discussion. I'd ask the authors to include some rationale or take-home to help contextualize these findings if this is kept. If it is to contrast with TPE, I'd move this all to Discussion, potentially Ext Data (it is certainly related to the main thrust of the manuscript but isn't totally required). I'd also ask if TPE might just be posed as a select form of APE? I don't know if this is reasonable, but it could benefit everyone. Along these lines, I felt the discussion in several places could benefit from inclusion of the Yittri and Dudman 2018 perspective or other work that points to the conserved computational/functional anatomy across the striatum serving whatever need the inputs indicate.

This data was included to demonstrate that we could replicate at least part of the work related to TPE and show that we can record both action and threat related signals in the same TS recording sites. We agree that this sentence disrupts the flow of the story, so we have moved reference of this to the discussion.

We agree that in the extreme case TPE could be a select form of APE, we now highlight this better in the discussion. We also point out that it's possible that different populations of DA neurons encode APE and TPE, this is likely because there are multiple distinct genetic populations of dopamine neurons that project to the TS (see the data in response to the first comment) so more work will be needed to relate these findings. The idea of a core computational function across the striatum, such as the elegant HDG model is important and is now included in the discussion. As is reference to the original Yittri & Dudman 2016 paper that is very relevant and important to this work in many ways, not least that the update term for APE was very similar to that used in their learning equations.

2c - Approximate depth would be good to have, and if angled, please indicate. It is implied in 2b, but its inclusion wouldn't hurt. This is also helpful as Fig 5g appears to place TS and VS almost above/below each other rather than in front/behind.

We agree with the reviewer that knowing the exact location of the fibers in x,y,z is useful knowledge. Because of this we serial sectioned every brain and aligned the fibers locations to the common allen reference atlas, Extended Data Figs 1, 3, 4 and 9. In all cases we provide plots that show the x, y and z coordinates of the fibers.

None of the fibers were implanted at an angle, this is now mentioned in the methods section.

Our brain schematic used in Fig 5g and other places is compressed in the AP axis to save space. These schematics are only intended to illustrate the experiment that is being performed. We have adjusted figure legends to highlight this.

2g - $y=0$ looks to be of a different font or font size? In my experience, 6 figures is highly-irregular for this journal. I might suggest combining substantial portions of fig 4 and 5.

The font was different, we thank the reviewer for noticing this.

We believe the guideline on the number of figures has recently changed along with some other changes to the journal, such as removing the "letter" format. If we are asked to reduce the figure number then the way of combining the figures that the reviewer suggests is a good idea.

Pale text - high sound in fig 5,6 and fig 6b,c - is difficult to read for my eyes.

These have now been adjusted.

While of potentially wide-interest, the section on habit in the Discussion could be moved to a Review piece and the conclusions removed if needed for length concerns. Similarly, while I appreciate naming the limitations, a more concise direct integration of these caveats could be done. I also don't see the need to outline 'future work' in a manuscript like this, but these are more for the Editor to determine.

The limitations and the future work sections were relics of a previous submission and have now been removed. We have retained the section on habits because this was requested by reviewer 3. We agree with the reviewer though that a review article would be a good format to expand on this and other topics.

While I also favor the correct spelling of behaviour as is done in Methods, consistency is key.

Thank you for noticing the discrepancy, we now use behavior throughout to match the American spelling in the rest of the text.

Overall, we would like to thank the reviewer for their helpful comments, we believe that addressing their points has made the paper stronger.

Referee #2 (Remarks to the Author):

This study examined the function of dopamine in the tail of the striatum (TS) using an auditory discrimination task (cloud of tones [CoT] task) in mice. The authors first show that the TS is required for learning and performance in this task. The authors then performed a series of experiments using fiber photometry and optogenetic manipulations, and conclude that dopamine in the TS signals action prediction errors (APEs), or “the difference between the action that is taken and the extent an action was predicted in a given state” to strengthen cue-action association in a value free manner.

The idea that dopamine in the TS signals APEs is potentially very interesting. The data are of high quality and the manuscript is well-written. The idea that TS dopamine is related to contralateral movement is relatively convincing (but see below). However, there are several substantive issues, which are elaborated below. Importantly, some of the results contradict previous observations. After considering these issues, the main pieces of evidence supporting APE, as opposed to “value-less” movement-related signals, come from the two following results: (1) a decrease of dopamine signals over learning, and (2) A contralateral choice bias induced by optogenetic stimulation during movement. However, the first point is not necessarily specific for APE, and the second result has errors in the statistical analysis and presentation. The explanation and analysis of the APE model is also lacking from a computational perspective. Overall, the presented data do not strongly support the main conclusion that TS dopamine signals APEs.

We thank the reviewer for their detailed reading of our manuscript. Their comments were very helpful and we believe that addressing their points has made the paper stronger. We provide detailed responses to their points below, including new analysis and experiments that we believe help support our conclusion that the TS dopamine signals reflect APEs.

Major issues:

1. A previous study by Chen et al. (2022, cited) characterized dopamine signals in a similar auditory discrimination task. Their task design, which separated the timing of the cue and movement, demonstrated that dopamine responses in TS are primarily driven by sensory cues. This contrasts with the findings of the present study. Since the present study's task was not specifically designed to differentiate between stimulus- and movement-related responses, the conclusion is not very convincing. Additionally, there are other data that

appear to contradict previous studies as well. For instance, while the present study argues against sensory- and novelty-related responses, multiple previous studies have now shown that TS dopamine responds to sensory stimuli and that novelty enhances these responses.

We appreciate the reviewer raising this important point about the proximity of cue and movement onsets. This comment helped us better articulate the evidence supporting our conclusions about the nature of the DA response. To address this concern directly, we had designed a version of the task that temporally separates these events. In this task variant we play the sound as the mice return to the center port and have the sound terminate when they poke into the center port (Extended Data Figure 5o). We chose to play the sound on the return as opposed to imposing a waiting period as they did in (Chen et al., 2022) as we don't know if the movement responses, we see are related to movement preparation or execution. With this task variant we still see a large response to contralateral movements initiated from the center port even though there is no sound at this timepoint (Extended Data Figure 5q). In addition, we don't see any significant response to the cue when it is played while mice return to the center port (Extended Data Figure 5p, r-t), even if they have not been experienced by the mice before (Extended Data Figure 5t). To further confirm there is no response to our sound stimuli we also play the same task sounds randomly as mice explore the arena with the ports blocked. Again, these sounds elicit no response in the DA signal we record in the TS (Extended Data Figure 5u-y).

Together we have seven separate pieces of evidence that show the DA response in the TS is related to movement and not the stimulus.

1. There is a significant response to contralateral movements in an open field arena (Ext Data Fig 5h-j).
2. There is no response to the high tone, low tone or WN stimuli when they are played in an open field arena (Ext Data Fig 5u-y).
3. The size of the response to orienting movements in the task is significantly different depending on if a contralateral or ipsilateral action is taken (Figure 2e, f). This difference in the response does not depend on whether the contralateral stimulus was paired with a high or low tone stimulus (Ext Data Fig 4h).
4. There is a significant response to contralateral action in the task even when the sound is omitted (Ext Data Fig 5a-g, q).
5. There is a significant movement response at times when there is no sound in the task, i.e. when the animals leave the reward ports and return to the center choice port. These responses are also significantly different depending on if a contralateral or ipsilateral action is taken (Ext Data Fig 4j-l).

6. There is no response to the sound cues in the task when the sound is temporally isolated from the movement (Ext Data Fig 5o-t), this is true even in the first recording session where the tones are experienced for the first time (Ext Data Fig 5t).

7. The size of the dopamine response is correlated with kinematic parameters such as speed and turn angle (Ext Data Fig 5k-n, Ext Data Fig 6c-g).

With regards to the observations about the DA signal from the Chen et al., 2022 paper, there are three main differences between our results and those reported in that study: they don't see any difference in their DA response whether the mouse makes an ipsi- or contralateral action; their response seems locked to the cue; they have reward responses in their recordings. We believe it is likely that we are recording a different DA signal, even though both papers are supposed to be recording DA responses in the TS. In line with this we can replicate some of their findings when we place our recording fibers more rostrally in a region termed the posterior DLS (Tsutsui-Kimura et al., 2022, bioRxiv) (Ext Data Fig 4a,b,n,o). When others have recorded in the posterior DLS they also see reward related responses like we do (Tsutsui-Kimura et al., 2022, bioRxiv). Others that record DA responses in the more caudal TS also see value free signals as we do (Tsutsui-Kimura et al., 2022, bioRxiv). Equally, these value-free DA responses have been recorded in the primate homolog of the TS (Kim et al., 2015, Cell). In line with what we see, there is a difference in the size of these responses depending on whether the action that is taken is contralateral or ipsilateral (Kim et al., 2015, Cell). Taken together, the Chen paper is the outlier, in the fact that they see consistent reward responses in the region they claim is the TS. Given the systematic mapping from the (Tsutsui-Kimura et al., 2022, bioRxiv) and our more rostral posterior DLS recordings, it is likely the Chen et al recordings came from the posterior DLS or adjacent regions and not from what we and others refer to as TS. This is probable, as the coordinates for the TS are more rostral in the Chen paper than ours. All our recordings were aligned to the Allen common coordinate framework and verified to be in the auditory recipient TS, as defined as the region that receives input from the primary auditory cortex. Similar verification was done in the Tsutsui-Kimura et al., paper but this was not done in the Chen paper so it's unfortunately not possible to systematically confirm the difference in location of our recording sites.

2. One of the important pieces of evidence supporting APE is a decrease of TS dopamine responses over learning. The decrease over learning has been reported previously (Menegas et al., 2017, cited) and is a straightforward prediction of stimulus novelty/salience (Extended Data Fig. 8). Thus, such a decrease is not necessarily strong evidence for APEs. Furthermore, the study lacks proper control experiments. To demonstrate such a decrease over a long time, the authors need some control experiments to show that the observed decrease is not due to an artifact, such as a decrease in the level of sensor expressions or bleaching.

We thank the reviewer for these important points about signal decay and experimental controls. Their comments encouraged us to perform additional analyses that strengthen our

conclusions. We provide several new pieces of evidence showing that the decay in the TS DA response reflects APE rather than salience/novelty and demonstrate that the decrease in signal is not due to bleaching.

Firstly, we performed a linear regression predicting the size of the dopamine response (TS dopamine at time of choice, VS dopamine at time of cue) on correct contralateral trials from previous choices for the same stimulus (left panel below). In this analysis, a positive regression coefficient means there is a larger DA signal on the current trial when the side chosen on the current trial was chosen in response to the same stimulus in the past (how far in the past is given by the x axis). A negative regression coefficient means that the DA response is smaller when the chosen side had been chosen in response to the same stimulus in the past.

a) Example mouse showing the average DA response traces when a contralateral choice was made in response to the appropriate stimuli. The traces were grouped depending on whether the mouse had made an ipsilateral or contralateral choice in response to the stimulus associated with the contralateral choice b) same as A but for an example recording of the DA signal in the VS. c) Regression coefficients for when the size of the DA peak on correct contralateral choices is predicted from the appropriate or inappropriate contralateral state-action pairing on the previous trials, 1-5 trials back. Regressions were performed separately for each lag. Stars represent 1 sample t test against zero for per mouse regression coefficients, corrected using the Bonferroni method for multiple comparisons. VS: $n=7$ mice, $p = (0.005, 1.0, 1.0, 1.0, 1.0)$, effect sizes = (cohen d : 2.23, 0.37, 0.23, 0.17, 0.13). TS: $n=6$, $p = (0.04, 0.20, 0.20, 0.47, 0.63)$, effect sizes = (cohen d : -1.72, -1.13, -1.13, -0.84, -0.75). Lines represent the mean across mice for each recording site, shaded regions depict SEM. d) Model predictions for how the APE response should change depending on the conditions described in panel A. e) Same as d but for the predicted changes in the RPE response.

From this analysis it is clear that the TS DA response on the current trial to a contralateral choice is smaller if the same choice had been chosen in response to the same stimulus on the previous trials (panel a,c), whereas the VS DA response to the cue is larger if the same choice is repeated in response to the same stimulus (panel b, c). This is consistent with TS dopamine encoding APE as a repeated choice would be more predicted, making TS dopamine signal smaller on the current trial (panel d). VS dopamine would be expected to be larger at the cue in the case of the repeated (correct) choices as the cue would be more predictive of reward (panel e). The fact that this difference is seen between one trial and the next is inconsistent with bleaching artifacts. It is also inconsistent with salience and novelty because the sensory history is the same between the current and the subsequent trial, the difference is just what action was taken in response to this stimulus on the previous trial.

These results are now reported in Figure 3i-m.

Secondly, to do an additional check for the effects of bleaching we have performed a correlation analysis between the mean session TS dopamine response at the time of contralateral choice and the days since last recording. If the decrease in signal is due to bleaching, the dopamine response size should be highly correlated with how frequently the recordings were performed (bleaching would be predicted to lead to larger APE responses (y-axis) when there was a longer gap between recordings (x-axis) due to recovery of the fluorophore and/or new protein translation with time. In contrast to this, there is no clear trend in this analysis showing that the number of days since the last recording session does not significantly affect the size of the TS dopamine response, making bleaching an unlikely explanation for the decrease in signal seen over time that we report.

Average TS dopamine responses at time of contralateral choice per session as a function of the number of days since the previous recording. n=6 mice, (lines are lines of best fit, using seaborn.lmplot. Each color represents an individual mouse, each point a session.) 3 of the 6 mice were recorded only every 2 days so it was not possible to fit a line.

We also examined whether the difference in DA signal between subsequent sessions is related to how frequently animals were recorded from. For this, we correlated the days since the last recording session with the change in TS DA since last session. We observed no relationship between how much DA changed between sessions and how often we recorded (see plot below). Two of the mice were always recorded every two days (hence the lack of other data points for those colors).

Average difference (current session mean response - previous session mean response) in dopamine responses at time of contralateral choice per session as a function of the number of days since the previous recording. $n=6$ mice, (lines are lines of best fit, using seaborn.lmplot. Each color represents an individual mouse, each point a session.) 3 of the 6 mice were recorded only every 2 days so it was not possible to fit a line.

In addition to the conclusions from this new analysis, we do not believe that bleaching offers a simple explanation of our previous results. Firstly, in all of the mice we record the DA response in both the TS and the VS. While the signal in the TS decreases, it increases in the VS. This is despite using the same virus and recording for the same period in the same mice. That the recordings came from the same mice is now highlighted further in the methods section. Secondly the state-change experiment (Figure 4a-e) gives another acute timepoint to see how the signal changes.

Taken together we now have four separate pieces of information that support our conclusion that the changes that we observe are not due to bleaching and additional evidence to show that the decrease in time is consistent with APE but not changes in salience/novelty.

3. A previous study (Lindsey and Litwin-Kumar, 2022, cited) has already discussed a decrease in movement-related dopamine signals over learning to support the concept of APEs although they used the term, “action surprise”, and action surprise is defined differently from the APEs discussed in the present study. The authors should more clearly discuss the contributions made by Lindsey and Litwin-Kumar (2022) as well as the difference in the definitions.

We appreciate the reviewer highlighting the importance of the Lindsey and Litwin-Kumar (2022) theory paper. We agree that it provides valuable theoretical insights, particularly in demonstrating the consequences of combining RPE and APE driven learning. To clarify the relationship between our work and theirs: in both papers APE is defined in the same way as just a subtraction of the action predicted from the action taken. The main difference is how we combine RPE and APE: in our paper these teaching signals are used to independently

update the weights in different controllers; in the Lindsey et al., paper they propose that RPE and APE are combined into a single dopamine signal. Our paper provides the first direct evidence that APE is present in the brain and that RPE and APE are present in separate populations that update different circuits, like what was proposed in the original theory papers that proposed APE should exist (Miller et al., 2019, Bogacz., 2020).

The reviewer is correct that this paper discusses experimental evidence for APE, for which they cite our preprint stating,

“Excitingly, concurrent experimental work provides direct evidence for an action surprise-like signal in dopamine neurons projecting to the tail of the striatum, and for a causal influence of these neurons on task learning [22 “Greenstreet et al., 2022”]. Our results show that such an influence is a necessary component of an architecture capable of off-policy learning.”

They also mention an additional study and ours as showing that movement-related dopamine decreases over the course of learning.

“Thus, we predict a reduction in the magnitude of action surprise signals over the course of learning. We observed this phenomenon in both simulated tasks (Fig.4A). This prediction is supported by findings that dopamine activity coinciding with lever-press movements decreases with repeated task practice [12,22].”

12. *“Da Silva et al., Nature 2018”. Movement initiation paper from Rui Costa lab.*

22. *“Greenstreet et al., 2022”. Our APE paper*

The paper from the Costa lab is an excellent paper that we now discuss more and cite heavily. It is especially important as the decrease they see in the movement-related dopamine signal over the course of learning is seen in a level-pressing task so the decrease cannot be attributed to changes in sensory salience/novelty. We already cited this paper as a result that is consistent with APE but we now highlight this more as well as a new theory paper from Nathaniel Daw’s lab that shows that a feature-specific APE model can explain the diversity of movement related responses that were first observed in Da Silva et al., 2018 paper.

4. Extended Data Fig. 6a-g. To demonstrate that TS dopamine signals are not related to sensory cues, the authors omitted the sound cue on a subset of trials, and showed that TS dopamine still responded during movement with similar amplitudes. While this appears to support the authors’ point that these responses are not sensory-driven, it is puzzling from the point of view of APEs. As a cue would make the left or right movements more predictable in the task context, the absence of a cue would make a left or right movement more surprising, thus the APE should become larger. This result, thus, contradicts the authors’ main point that TS dopamine signals APEs.

We are glad that the reviewer appreciates the use of this experiment for showing that the response is seen regardless of the presence of the tone. As they correctly note, this was indeed the primary intention of these experiments. We refrained from using this experiment as a test of the APE response to a novel state as mice had experienced having no sound when they poke into the center port many times prior to data that is reported. This is because when the mice make the wrong choice in the task there is a time out period ('punishment') where if they poke in the center port there will be no sound and no trial is initiated. Despite this, mice almost always still make a choice ($99.2 \pm 0.56\%$ of times) after poking in the center port during this time out period. Indeed these mice had experienced thousands of silence-action pairings before we ran the reported "silence" sessions (mean: 2516 ± 689). We report the number of silence-action pairings in the paper to make it clear this isn't a novel state (Extended Data Figure 5e). As the silence-action pairing is not novel and should have been reinforced by the APE system in the same way as the tones it is not surprising the APE response is smaller than the naive state and would in fact be expected within our APE framework. It may seem surprising that the silence response is not a little larger than the tone responses given that there were less, albeit still thousands, of silence-action pairings prior to the test. There could be many reasons for this, for example mice also experience silence-action pairings when they return from the side ports to center port so this would add thousands of additional pairings. This could intuitively be treated as a different state but the action and the lack of sound are the same so it might be reinforced in the same way. Additionally, although there is no sound played on the silence trials we cannot rule out that mice "hallucinate" a sound, as it has been shown they do given their prior for hearing a sound (Schmack et al., 2021), and therefore generate a prediction. Finally, we don't know how confidence modulates the APE signal and there will presumably be a change in choice confidence following the lack of a sound being played. Taken together we do not think that our "silence" results contradict our conclusions about APE. We rather feel like it is hard to generate accurate predictions for how APE should look in this experiment. This was why we refrained from using it as a test for APE but just mentioned it as a means of showing the choice-related signal is seen regardless of the presence of a tone as the reviewer appreciates.

In the same vein, it seems that the most obvious prediction of APE is that (in expert animals) it should peak on error trials, when the animal's action is relatively "more surprising" given the state. (This might be especially true for error trials in which the evidence/fraction of tones was high.) The authors should test this prediction explicitly, as most of the predictions tested (e.g. white noise) are somewhat roundabout.

In terms of error trials, the situation is complex, as the size of the APE tends to be smaller on error trials (not a significant effect but see below for Figure). This would not immediately be predicted by the model of APE, as the reviewer points out one could predict the APE would get larger on error trials. A couple of points make the interpretation of this result not straightforward. First, on error trials there are significant systematic changes in the animals' behavior. These changes are larger and more consistent than anything that we see in our

other experimental manipulations. The differences in movement are also so large that we cannot find trials that can be matched for behavioral performance within the same mice. Due to this we cannot determine if the change in the size of the response is due to changes in behavior or in prediction. Second, we do not know why mice are making errors. Errors could be, for example, caused by errors in perception, attention or choice. In each case these would lead to different expected changes in APE. Due to this we don't include this data in the paper as we do not believe the results are as well controlled or interpretable as our defined experiments.

Differences between correct and incorrect trials in TS-recorded mice. Stats: paired t-test. From left: z-scored DA response at time of contralateral choice ($p=0.07$), movement duration in seconds ($p=0.0008$), average speed ($p=0.004$), time to move ($p=0.017$).

5. Stimulus novelty/salience could also very easily be state-dependent and value-free. In other words, I disagree with the modeling in Extended Data Fig. 8. First, White Noise could easily be more salient than Tone (Extended Data Fig. 8a), by virtue of either being experienced less often or because of intrinsic auditory characteristics. Second, if TS only cares about the cue itself and completely ignores the associated reward, then for novelty/salience you still might not expect any change upon value manipulation (Extended Data Fig. 8b). The assertion that novelty predicts no response as a function of cue value (Extended Data Fig. 8b, center) is simply guesswork on the authors' part; there is no reason to assume a (somewhat unpredictable) tone should be treated as 0% novel and ignored, even by expert animals. Between these two points, the novelty/salience predictions would exactly match those of APE in Extended Data Fig. 8. While novelty/salience wouldn't themselves predict everything about contralateral vs. ipsilateral movements, it is quite conceivable that TS dopamine could also play a motor role, and the combination of these two roles could give rise to all the results presented here.

The reviewer makes a good point that if the novelty model responses did not decay to zero in expert mice, which is when the value change experiments occurred, then there would be no difference between the APE and the Novelty model in the Extended Data Fig. 8a,b simulations. Even if that were true then, as the reviewer acknowledges, still only APE would be able to explain all the results showing the signal is movement-related without proposing

that the TS dopamine signal combines movement and novelty signals in some way. In addition, salience/novelty is also inconsistent with our new trial-by-trial analysis where we show that the size of the TS DA response is modulated by differently depending on which action was taken in a particular state, this is despite the recent history of sensory experience being the same in each condition (Figure 3i-m).

The fact that we always see a movement-related DA response in the TS no matter whether the movement occurs in the task or not, that the signal is different for contralateral and ipsilateral movements and that the signal is affected by kinematic parameters, shows that the signal is movement related. The fact that we never see a response to the cues when they are presented when a movement does not occur, even when they are novel, shows that there is no response to the cues that we present per se, equally the size of the movement response is not significantly different depending on whether the cues are present or not. This suggests that if there is a novelty component to the TS DA response then it would have to function to adjust the size of the movement-related DA response. If this is the case, then it would essentially be the same as APE. This is because the movement-related signal would have to be actively suppressed when the action is taken in response to a non-novel context, i.e. the mice would have to develop a prediction.

The reviewer is correct that the parameters used in the models are a choice made by us, this is always the case for models and in this situation, we have tried our best to match our predictions to the known findings in the literature. Pertinent to this discussion is that in the Menegas et al., 2017 paper they saw that TS DA responses to a novel odor were completely suppressed after only 26-30 trials. In their paper the odor delivery was even less predictable than our tones because the mice had no control over their delivery. Due to this we think it is reasonable to set our novelty model to zero after the mice have experienced thousands of tone presentations. In this paper it should be noted that they had not proved the response they saw had anything to do with the stimulus per se as they do not report any data related to the behavioural reaction of the mice. The second point is that in the Menegas et al., 2017 paper when they paired the odor with a value then the response to the odor was sustained. It is for this reason and a lot of literature related to motivational salience that we added value into our salience model to explain why the signal could be sustained following reinforcement. Again though, please note this paper did not attempt to see if this sustained response was due to the initiation of licking behaviour which the mice do in the task. Taken together we feel like our choice of parameters in our models is reasonable and the models cover a reasonable range of possibilities.

Similarly, the significant regression coefficient on dopamine response size for TS (Fig. 5h,j) could also reflect that cues that are more salient (for whatever reason) cause greater effects on subsequent behavior without being related to the chosen action at all. Also, this analysis is discussed in the main text as if it's an ANOVA, whereas in fact (as best as I can tell) dopamine response size, perceptual uncertainty, and their interaction are all fit independently. ("Stats: two-tailed p values for the t-stats of the fitted coefficients are reported.") This seems

wrong, as perceptual uncertainty can (and probably does!) directly influence DA response size, and this will “leak into” the DA response size coefficients if they are regressed separately.

In Figure 5h,j,l & m we are looking at correlations between the size of the TS DA signal and the subsequent behaviour. Because of this the reviewer is correct that these results in isolation do not say anything about what the TS DA release may encode. These experiments are there to show that the behaviour on subsequent trials, both in the degree of repeating a stimulus-action association and in the similarity of the trajectory, is correlated with the size of the TS DA signal.

In terms of the analysis, we apologize for any confusion. Our analysis employs a multiple regression analysis, where perceptual uncertainty, dopamine response size, and their interaction are modeled simultaneously. This approach allows us to estimate the unique contribution of each term while accounting for shared variance. We now better clarify this in the text.

6. The claim that TS dopamine reinforces state-action and not state-outcome associations rests on a grave statistical error. In Fig. 5d, the authors compare real to shuffled data using a Wilcoxon rank-sum test, which is invalid. For a permutation test, the real summary statistic should be compared to a percentile of the null distribution of that summary statistic (e.g. 97.5 for a two-tailed $p < 0.05$). Using a rank-sum test for this purpose is treating each permutation as an independent sample of data, which it is not. (You can imagine increasing the number of shuffles, say, to 100,000, to the point that any effect, no matter the size or sign, would look “significant” on this metric without collecting more data, which tells you automatically that it is invalid.) The alternative to a permutation test here is to just do a Wilcoxon signed-rank test relative to zero (2-sided), which does not appear to be significant by eye. This is the most important panel of this figure; without it, the entire figure essentially falls apart. Indeed, I would argue that the causal effect of TS dopamine on trial-by-trial behavior is the linchpin of the entire paper, and without it, the case for believing in APE becomes significantly weaker.

We thank the reviewer for this important methodological critique regarding our statistical approach. Following their helpful suggestion, we calculated the significance of the effects using a Wilcoxon signed-rank test relative to zero (2-sided). We now report only these statistical results in the manuscript and have modified our figures accordingly. Please note that our conclusions still hold true given that the results are still significant when performing this test. Particularly, for the data and statistical test highlighted and suggested by the reviewer (state-outcome effect of DA-TS but not DA-VS stimulating at 4mW), the p value is 0.018. Please note that we have also slightly changed our analysis as requested by another reviewer. In this case, in the previous manuscript version we excluded the first 75 trials of the stimulation block to allow the animal to accommodate the stimulation block. We now calculate the bias effect using all the trials. This has been reflected in our updated methods.

To further validate these effects, we reasoned that if robust, the effect size should scale with stimulation intensity. To test this, we have repeated all our TS and VS optogenetic experiments using 8mW instead of 4mW stimulation. The results of these experiments confirm all our previous results, including showing that stimulation of TS but not VS DA release can reinforce state-action but not state-outcome associations. The magnitude of this effect is larger for 8mW than for 4mW but at this level of stimulation we still do not see any effect on biasing movements directly. For example, stimulating on a subset of trials (15%) does not cause a choice bias on the stimulated trials and stimulating TS DA release in a closed-loop manner in an open field does not make the mice more likely to initiate an action or change how vigorous the movement is when it is initiated. We now report these results in Figure 5c-f and Extended Data Figure 8c-t.

7. Also in Figure 5, the authors should show all the psychometric curves, for all the sessions, for both VS and TS (Fig. 5c). (They should also show VS bias in Fig. 5f.) This is because strictly speaking, bias is not the only thing to (appear to) change, as performance maxes out well below chance (Fig. 5c). The authors choose to ignore this by fitting a separate max_performance parameter, but it is highly relevant to the conclusion. In particular, different max_performance is contrary to the APE model prediction (Extended Data Fig. 9d), a fact that the authors never remark on, and which suggests a more direct role in movement facilitation for TS dopamine than the authors' APE account.

We thank the reviewer for this suggestion about presenting our data more comprehensively. Below we provide plots showing the psychometric curves for all TS and VS sessions when we stimulate at the center port to assess the ability to reinforce state-action associations. In the paper we focus on showing the psychometric curves for the new 8mW data (Extended Data Figure 8c), as not all 4mW stimulation sessions used the psychometric version of the task and the baseline psychometric curves were more stereotyped in the newer 8mW data. The 4mW psychometric curve data will still be available to the public, and already is, as we have uploaded the code and data for reproducing our figures at https://github.com/HernandoMV/APE_paper.

We have also added more detail in the methods about how we fit our psychometric curves. We use the LogisticRegressionCV function from scikit-learn package in python. We also include the data for 5f where we stimulate dopamine release in the VS.

For 8mW TS:

Evidence for contralateral movement

For 4mW TS:

For 8mW NAc:

For 4mW NAc:

8. The authors point out that APE is (conveniently) a scalar (although see below), allowing it to extend quite nicely to a large space of possible actions. However, the only actions examined in this paper are left and right. This is common in studies of the striatum, which is highly lateralized, and much of that lateralization is also reflected in dopamine activity (Parker et al., 2016; Tsutsui-Kimura et al., 2020). However, (1) it's not immediately clear why APE would only be broadcast to one hemisphere, as presumably it could just be ignored in the ipsilateral hemisphere which is less active and (2) it should also be possible to detect APE in a less lateralized task, just by looking e.g. at sequence entropy (Markowitz et al., 2023). The failure to do so here is unfortunate, and leaves open the possibility that the signals have more to do with a permissive signal for gross movement, as opposed to APE per se.

The reviewer raises important points about lateralization and the broader applicability of APE. Regarding the first point about bilateral signaling, we believe the biological basis for lateralized signals stems from the anatomical organization of the dopaminergic system. The inputs that the DA neurons receive come almost exclusively from one hemisphere (Menegas et al., 2015, Watabe-Uchida et al., 2012). Due to this the ipsi/contra action initiation commands will not be uniformly passed onto DA neurons in each hemisphere.

In terms of the second point, the idea of testing APE by repeating the Markowitz experiments is a good one and we agree that looking at the relationship between the signal and the entropy of the syllable transitions will be a good test of APE. However, we believe this is beyond the current scope of the paper mainly because the results could be more complex than simply replicating the Markowitz paper with TS recordings. The reason for this is that the APEs should be state dependent and there could either be a homogeneous APE broadcast to all striatal regions or there could be closed loop str-DA-str channels, as has been shown anatomically (e.g. Ambrosi & Lerner., 2022, see also our data in response to reviewer 1 point 1). If the latter organization is the case, then the APEs in the TS may not be modulated by sequence transition probability as the TS will not receive the appropriate “state” input. This will be an interesting avenue for further investigation as others have appreciated and recently discussed (Lee et al., 2024).

Related to this point, the authors say, “This raises the possibility that movement-related dopaminergic responses throughout the striatum^{8-11,13,36}, may reflect APE.” They then cite a handful of studies, including Markowitz et al., 2023, which record dopamine in the DS as evidence for this claim. But in the authors’ own model, the DS is receiving RPE as the actor, not APE! They cannot cite these studies in support of their conclusion; if anything, these studies run counter to it.

We think that these studies are consistent with what we believe, which is that all movement-related dopamine activity throughout the striatum may encode APEs.

We have been careful not to speculate about where parts of the value-based controller are located. We also dedicated a section of our discussion highlighting that all movement-related dopamine signals could conceivably reflect APEs, including those that project to the dorsal striatum. This is matched by our final schematic in (Extended Data Figure 9). To make it explicit what we think we have now added a dotted arrow in this schematic to indicate that APE’s may be broadcast to the dorsal striatum, predominantly to the DLS, that receives the most movement-related DA signals.

In terms of the Markowitz paper potentially supporting APE we are only repeating what the authors themselves wrote in their manuscript citing our preprint (ref 48 and the Lindsey et al., theory paper ref 47):

“Second, DLS dopamine may represent the output of a circuit that evaluates the content of spontaneous behaviour. Although dopamine has classically been thought to report reward-prediction errors—which by definition require the provision of reward—it has recently been argued that dopamine may also encode action-prediction errors 47,48 (APEs). APEs are proposed to occur as animals either execute or plan to execute a behaviour that is unexpected in a given context; in the setting of spontaneous behaviour, an APE-like model would predict that DLS dopamine represents the comparison between the expressed (or soon-to-be-expressed) behavioural syllable and that which would have been expressed at a particular moment given an idealized transition matrix. Our finding (similar to that in ref. 9) that syllable-associated dopamine transients reflect the probability of the next expressed behavioural syllable—but convey no

information about syllable identity—is also consistent with the proposed role of APEs in conveying information about action errors that is independent of the specific identity of the expressed behaviour.”

There are other possibilities for what explains the Markowitz results but we, like the authors, are just pointing out that the signals are conceivably consistent with APE.

9. From the Methods, it is not at all clear how the model works under the hood. In Miller et al., 2019 (frequently cited as inspiration by the authors), a_k is a “vector over actions in which all elements are zero except for the one corresponding to” a_k , and A is a 2D matrix indexed by (s, a) . Consistent with this, the update equation that the authors give for W_{tail} implies that the entire matrix is updated. However, elsewhere, the authors go out of their way to say that “the APE term need not be action-specific.” Which one is it? If their model requires a vector-valued dopamine signal, then their fiber photometry-based recordings can’t support it. Alternatively, if for biological reasons, the authors want to make it scalar-valued by only considering the cell corresponding to (s, a) , how does this impact the learning dynamics? In either case, please resolve this ambiguity by making it explicit in the equations.

We are grateful for the reviewer’s careful examination of our modeling approach. In our models the APE term is scalar. In the model of dopamine the APE is essentially a scalar due to rectification of negative components of the APE vector. Thus, the only component of the APE vector that is non-zero is that corresponding to the action actually taken. In the dual-controller both the action taken and the predicted action are scalar terms. For the action taken we take the maximum value of the output of the dual-controller as the actual action, this is a scalar value. For the predicted action we only consider the most predicted action, this is achieved because we initially binarize the instantaneous predicted action vector when calculating the predicted action such that it is 1 for the index corresponding to the maximum predicted action and zero everywhere else. This means that the final predicted action will also be a scalar term. We have now clarified this in the methods section.

The models are able to use the scalar APE signal to update the appropriate cell of the state-action weight matrix because we implement a three-factor learning rule. This means that only the cell corresponding to the states and actions that were actually observed/taken on a given trial is updated by the APE signal and adjusted according to the learning rate. In our model the information regarding the actual action taken would conceivably be provided by an efference copy mechanism as has been proposed for three-factor striatal learning rules (e.g. Fee M., 2015). We illustrate the needed efference copy signal for the actor and TS in Figure 6a and now make this clearer in the methods.

There are several other issues with this section. (1) The critic is defined by a 2×2 matrix that maps sensory states to actions. But traditionally in RL, a critic ignores actions and is directly parameterized as $V(s)$. It’s unclear how important this departure from standard RL is. (2) How can δ_A equal $\arg\max(\delta_A)$? It seems like this ought to be a =

$\text{argmax}(\delta_A)$. (3) Is the role of the differential equations just to low-pass-filter the RPE and APE? If so, make that clearer, and explain why this should occur in the model itself (as opposed to just the mapping from RPE/APE onto dopamine). (4) In one part of the Methods, the authors say that APE is proportional to “the difference between the action taken a_k and the stimulus action strengths given by the stimulus $A(s)$.” Later, they seem to switch variables and say its $a - \hat{a}$. Please be consistent between $A(s)$ and \hat{a} .

We thank the reviewer for pointing out these issues.

1) We have now changed the model such that it has a more traditional critic. We now model the critic as learning the $V(s)$ as the reviewer is correct to mention is the classic way. We have redone all our simulations, and this change does not affect any of the main conclusions. The only main difference is now the Critic weights are initialized at 0 because there is initially no expected value. 2) We thank the reviewer for pointing out this error. We have now corrected this equation. 3) The role of the differential equations is just to low pass filter the APE signal and model how the predicted action changes over time. This is similar to what was done in Miller et al., 2019. To avoid confusion with previous models we now calculate RPE with the standard temporal difference equations instead of using a differential equation. 4) We are now consistent in using the $A(s)$ notation.

10. The authors suggest four putative advantages of a value-free system, but three of them don't seem to work. (1) Stably storing associations in a manner more robust to short-term fluctuations. But there's no reason why you couldn't just adjust the learning rate in your actor, rather than having a completely independently system; indeed, plenty of work has explored these adjustments empirically (e.g. Behrens et al., 2007; Rushworth & Behrens, 2008). (2) Policy complexity. But the mathematical definition of policy complexity explored by Gershman, 2020 is equivalent the mutual information between states and actions. In other words, a highly deterministic mapping from (Low Hz \rightarrow LEFT) and (High Hz \rightarrow RIGHT) would have high, not low, policy complexity, contrary to the authors' assertion. (3) Value-free learning as a Bayesian prior. But the authors make no suggestion as to how the strength of the value-free learner might be modulated by uncertainty.

We think it is important to highlight what some of the reasons may be for the brain employing a value-free learning system, these are discussion points that others have already suggested. Indeed, the ideas that we share all originate from theory or experimental papers that we cite, as such these ideas about the value-free learning system have already been shown to work in theory or practice. We now make more effort to clarify these “advantages” may not specifically relate to our task but be more general consequences of having the value-free system.

Adjustments to the learning rate of the actor is indeed one way that has been proposed to adjust learning in the face of different levels of environmental volatility. This makes sense if

the policy is stored in the actor. What we have shown and what matches very nicely with work from the Hikosaka lab is that stable sensory motor associations are stored in the tail of the striatum (caudate tail in primates), which is also updated by a value-free dopaminergic teaching signal. In their work, which we cite heavily, they have identified the value-based actor and shown that it does not retain control over behaviour or retain the associated value of actions. It is because of this work that we propose that the value-free controller is for whatever reason more robust at storing long-term associations.

Regarding policy complexity, we agree with the reviewers reading the Gershman 2020 paper. However, in our task where there are only two states and two actions there is no other way for the mouse to really solve the task apart from linking the sound to the action. In this case it's not a good test of whether having perseverance helps agents to have an optimal policy. The larger point the paper is making is that having a system that increases perseveration is what allows agents to reduce the policy complexity. APE is one mechanism that has been shown in theory to drive perseveration, as proposed by Miller et al., 2019. Indeed, two recent theory papers from Peter Dayan's and Matthew Botvinick's labs used APE terms to augment their reward function to drive policy compression (Moskovitz et al., 2024, Saanum et al., 2024). Both papers cite our preprint as biological evidence for their proposed mechanism. We have now changed our discussion regarding this point to make it clearer what APE would be contributing in general rather than in our task.

In terms of Bayesian priors, it is true that we don't know how the value-free controller is modulated by uncertainty. Despite this it has been shown in theory that it can function to bias action selection under different types of uncertainty (Miller et al., 2019, Bogacz., 2020). We think that it's intuitive that the value-free system would store a prior for the most frequently used policy, i.e. a default policy that can be used when there's uncertainty about the expected value. This is exactly what's being explored in a branch of AI research that uses new dual-control models e.g. Moskovitz et al., 2024. We have now highlighted in the discussion that future work will be needed to determine how the dual control systems are modulated by different types of uncertainty.

11. Why would the actor but not the critic weights decay, as they are both updated by RPE, which is decaying to zero? Furthermore, is the actor weight decay the only reason that the combined controller can outperform the value-based controller? If so, it seems like cheating. Does this explanation imply that if the rewards were delivered probabilistically, such that RPEs continue to occur, you wouldn't get the same transfer over to the TS? If so, the authors should probably test this prediction experimentally.

We appreciate the reviewer highlighting the need to further clarify why we think it's reasonable to predict that the weights in the value-based actor but not the critic decay without continual reinforcement. From a biological perspective we don't know what would explain the decay in the actor weights vs the critic weights. If we speculate that the actor and the critic are located in the DMS and the VS respectively then there are a lot of molecular differences between the

SPNs in these regions as there presumably also are in the molecular characteristics of their synaptic inputs. There is also a lot of evidence that the representations in the DMS change over the course of learning such that this region is initially needed for task performance and the activity in this region reflects the task requirements. As animals become overtrained the DMS is no longer needed for task performance and the DMS activity no longer reflects task demands. A beautiful example of this can be seen in (Kim & Hikosaka, 2013) but there is a large body of literature on this. This is at least consistent with the weights that drive the DMS decaying as animals become experts at the task. In contrast, there is no evidence that animals lose information related to the expected value in the same tasks (e.g. Schultz et al., 1997, Amo et al., 2022). This highlights that animals can stably retain information about the expected value while information about the policy decays in the DMS, a putative candidate for the value-based actor. We have expanded our discussion to highlight these points.

In our model the reviewer is correct that the combined controller outperforms the value-based model in terms of max performance because the actor weights decay. The combined model will help increase the learning rate irrespective of this feature. It should be noted that we would also be able to replicate our behavioural results if we had used an arbiter to switch control between the two control systems (e.g. Miller et al., 2019, Bogacz et al., 2020) so in this sense having the actor weights decay isn't the only architecture that could reproduce our data. We think that the decay is a more parsimonious explanation of our results, and the basal ganglia literature as opposed to an arbitration model so this is why we use it.

In terms of probabilistic rewards, the results are hard to predict and would be hard to interpret. In one sense probabilistic rewards would mean that at the time of outcomes there would be larger + RPEs for positive outcomes but there would also be - RPEs for negative outcomes so it is unclear at which level the reward outcome RPE response would have to be suppressed (predicted) in order to have an equilibrium point where the weights in the actor start to decay. There is also the point that the state/action weights in the TS would be updated positively on every experienced trial but the weights in the Actor would not. Because of this it is possible that the value-free system could receive more positive updates than the value-based actor. Taken together there may still be transfer of control. Before we do these types of experiments, we want to take the time to determine if there is a region that corresponds to the value-based actor. Once we have tested this, we'll be in a better place to address these kinds of questions but believe this is beyond the scope of this paper.

Minor points:

12. Extended Data Fig. 6k-n uses quartiles, which can mask the true effect size. It would be nice to show the raw data, even if only to supplement the quartiles.

We appreciate the reviewer highlighting that quartiles can mask the effect size. We share this opinion, so we primarily use the quartiles for ease of visualization in Extended Data Fig 5k-m. To ensure this effect holds at a single trial level we also included a regression of single trial data showing the correlation between turn angle and TS dopamine response in early learning. The raw data and regression coefficients for this can be seen in (Extended Data Figure 6c-e). This corresponds to the same data used in Extended Data Fig 6k-m, analyzed at a single trial level. Extended Data Figure 6c shows an example mouse and panel Extended Data Figure 6e shows correlation coefficients for all TS mice.

13. Turn angle does appear to change over time, it's just not significant at the population level (Ext. Data Fig. 7h). However, if "filled circles represent significant correlations for individual mice," then it appears that 4/6 animals increase speed and 5/6 animals decrease turn angle significantly over training. As such, it seems misleading to claim that "Over the course of learning there was no significant trend in how either speed or turn angle changed (Extended Data Fig. 7f-h)."

We thank the reviewer for noticing this. The sentence "Over the course of learning there was no significant trend in how either speed or turn angle changed (Extended Data Fig. 7f-h)" is indeed a little misleading because in some individual mice there is a significant relationship. We have now changed this sentence in the manuscript to make this clearer. We now say "Over the course of learning there was a trend for mice to perform the task faster and with smaller turn angles (Extended Data Fig. 7f-h), this trend was significant in some individual animals but was not significant at the population level."

Why do they plot the dopamine response in Extended Data Fig. 7i but the dopamine residuals in Extended Data Fig. 7j? This doesn't seem like an apples-to-apples comparison. I'm assuming that for the multiple regression model, they predicted only dopamine response. Is this true?

We thank the reviewer for noting that our rationale for the regression approach was not clearly stated. We have now updated the manuscript to make it clear why we have taken this approach and we expand upon this here.

Extended Data Fig. 6i shows the DA response and the predicted DA response from a movement-alone linear regression model. These predicted signals are then subtracted from the DA signal to remove the proportion of the DA signal that can be accounted for by movement (turn angle + speed) alone. Having removed this proportion of the signal, we then perform a regression on the residuals with log trial number as the predictor to examine whether log trial number can account for the DA signal *over and above* movement (the coefficients are reported in Extended Data Fig. 6l). This is why the residuals are shown - to show the data that this model is trying to predict. We now clarify this further in the figure legend.

For completeness we also now plot here the fits of the angle + speed and the trial number regression models to both the data and the residuals. As can be seen in these plots the angle + speed model does not predict any decay in the DA response over time, so it does not match the DA response (panel a below), in contrast the trial number model does match the decay in the DA response over time (panel b below). Equally when we removed the variance explained by the trial number model, we see that this removes the majority of the decay over time response in the DA residuals (panel c below) and provides a better match the angle + speed model, as would be expected from the regression model. In contrast when we remove the variance explained by the angle + turn model there is no real difference between how the DA signal and the residuals decay over time (panel d below), this shows that changes in angle and speed cannot account for the decay in the signal over time.

a, TS dopamine response, binned per 40 trials of an example mouse over the course of training (blue). A linear regression model was built using average speed and turn angle to predict the TS dopamine signal. The model prediction from just the movement parameters over the course of training is shown in gold. **b**, TS dopamine response (blue, as shown in a). A linear regression model was built using the log trial number to predict the TS dopamine signal (the predicted signal is shown in gold). **c**, TS dopamine residuals after the response predicted by trial number (purple model prediction in panel b) is subtracted. A linear regression model was built to predict the residuals based on turn angle and speed (the prediction is shown in gold). **d**, TS dopamine residuals (blue) after the signal predicted by angle and speed (shown in gold in panel a) was subtracted. A linear regression model was built to predict the residual dopamine signal based on log trial number (shown in purple).

As the reviewer assumes when we, in Extended Data Fig. 6m, perform the multiple linear regression model with turn angle, speed and log trial number we are predicting the actual DA response. This analysis is designed to show a slightly different point - that the percentage variance explained by log trial number is higher even in a full model with all these variables. The other regression (Extended Data Fig. 6l) was performed on the residuals to show the effect of log trial number over and above that of movement.

14. There are some typos/omissions, e.g., “The identification of dopamine transients prior to movement initiation led to the idea that movement-related dopamine release may trigger the initiation of 8,10,12,13.” Also the Methods paragraph beginning with “For chronic D-AP5 infusion experiments” is in a different font. The letter σ is used to refer to both a ReLU function and a learning rate in the same section. There are inconsistent heading formats, etc.

Thank you for noticing these errors, these have now all been corrected in the manuscript.

Overall, we thank the reviewer for their thorough and constructive feedback. Their comments have led to significant improvements in our analyses and the clarity of our presentation, ultimately strengthening the conclusions of our paper.

Referee #3 (Remarks to the Author):

A. Summary of the key results

This manuscript shows that dopamine response in the tail of the striatum corresponds with an action prediction error, which provides a mechanism for the value-free learning that has been proposed on theoretical grounds. The paper provides a series of extensive experiments using a simple binary choice task that shows a clear dissociation between reward-related responses (in the ventral striatum) and action-related responses (in the TS).

B. Originality and significance: if not novel, please include reference

The results hold transformative potential for re-defining how we think about dopamine and reinforcement learning, providing anatomically distinct functional roles to dopamine in different parts of the striatum. The long-standing approach in the field (for almost 30 years) has been that dopamine encodes reward prediction errors. More recent work has challenged this view, but mostly along the lines that dopamine may encode more general prediction errors about outcomes (e.g., about identity) or even a broader distribution of outcomes. This work provides the first evidence for a different role of dopamine in learning—through action prediction errors in the tail of the striatum. Such action prediction errors have been proposed theoretically (in cited work) as being a key mechanism for the established and maintenance of learning. This paper provides a potential neural mechanism to this theoretical work. There is a nice parallel to the original RL-dopamine findings, which

discovered a correspondence between the prediction supposed by TD learning and the behaviour of (some) dopamine neurons.

We thank the reviewer for their detailed reading of the literature and for placing our work in this historical context.

C. Data & methodology: validity of approach, quality of data, quality of presentation

1. The experiments together present a fairly strong set of evidence in favour of TS dopamine encoding action prediction errors. There is very strong evidence that the responses are not reward-related and strong evidence that the response are related to action selection. I am a little less convinced that these represent action prediction errors. One of the main pieces of evidence used in favor of this claim is the experiment that shows how these response decrease over the course of learning. While it may be true that APEs should decline over time, other things change over time as well, including likelihood of different actions, and perhaps more abstract features like action initiation costs.

To show an action prediction error would require one step further—showing that the dopamine response on any given trial is a function of the action chosen on previous trials. That's what defines a prediction error—current action minus predicted actions. This is precisely what is shown with reward prediction errors and what made that story so compelling. Here, that is not shown. The closest is with the white noise experiment, where removing any state-action connection increases responding in the TS. This issue might be resolvable with the existing data (and no further experiments), but would require looking at the recent history of choices and showing that the current action response is a function of those. It might require further research that explicitly manipulates the frequency of different choices and assesses the size of the action prediction errors.

We would like to thank the reviewer for this excellent suggestion. We agree that it is crucial to show that the dopamine response on any given trial is a function of the action chosen on previous trials to validate our conclusions that the TS dopamine signal reflects APEs.

To address this question, we performed a linear regression predicting the size of the dopamine response (TS dopamine at time of choice, VS dopamine at time of cue) on correct contralateral trials from previous choices for the same stimulus (left panel below). In this analysis, a positive regression coefficient means there is a larger DA signal on the current trial when the side chosen on the current trial was chosen in response to the same stimulus in the past (how far in the past is given by the x axis). A negative regression coefficient means that the DA response is smaller when the chosen side had been chosen in response to the same stimulus in the past. Separate regressions were performed for each 'lag' (number of trials back value).

a) Example mouse showing the average DA response traces when a contralateral choice was made in response to the appropriate stimuli. The traces were grouped depending on whether the mouse had made an ipsilateral or contralateral choice in response to the stimulus associated with the contralateral choice on the previous trial. **b)** same as **a** but for an example recording of the DA signal in the VS. **c)** Regression coefficients for when the size of the DA peak on correct contralateral choices is predicted from the appropriate or inappropriate contralateral state-action pairing on the previous trials, 1-5 trials back. Regressions were performed separately for each lag. Stars represent 1 sample t test against zero for per mouse regression coefficients, corrected using the Bonferroni method for multiple comparisons. VS: $n=7$ mice, $p = (0.005, 1.0, 1.0, 1.0, 1.0)$, effect sizes = (Cohen d : 2.23, 0.37, 0.23, 0.17, 0.13). TS: $n=6$, $p = (0.04, 0.20, 0.20, 0.47, 0.63)$, effect sizes = (Cohen d : -1.72, -1.13, -1.13, -0.84, -0.75). Lines represent the mean across mice for each recording site, shaded regions depict SEM. **d)** Model predictions for how the APE response should change depending on the conditions described in panel A. **e)** Same as **d** but for the predicted changes in the RPE response.

From this analysis it is clear that the TS DA response on the current trial to a contralateral choice is smaller if the same choice had been chosen in response to the same stimulus on the previous trials (panel a,c), whereas the VS DA response to the cue is larger if the same choice is repeated in response to the same stimulus (panel b, c). This is consistent with TS dopamine encoding APE as a repeated choice would be more predicted, making TS dopamine more suppressed (panel d). VS dopamine would be expected to be larger at the cue in the case of the repeated (correct) choices as the cue would be more predictive of reward (panel e).

These results are now included in our main figures (Fig 3i-m). We thank the reviewer again; this suggestion was very valuable.

2. One major issue is the sheer number of researcher degrees of freedom available in the data analysis pipeline. In other research areas with which I am familiar, this challenge would be handled through pre-registration, but that seems not to be attempted (or maybe not possible here). As a result, every analytic decision (even when justified) seems post-hoc, and there are no robustness tests to ensure that these choices are not causing the results. Were these choices all made data blind? As a few examples just from a few pages (but really it's pervasive in the paper): there is the picking the sessions with the best performance (p. 26), trimming of the photometry trace to maximize variance accounted for (p. 27), arbitrary cut of 1 s on return events (p. 27), and splitting data into larger and small at an arbitrary cuts (p. 29). These analysis decisions have limited justification, and, though perhaps each one could be justified, a convincing analysis would need to show that the results are not driven by post-hoc pipeline decisions to extract the best possible story out of the data.

We appreciate these suggestions. We also believe that it's important to make sure our findings are robust and not just a product of our choices. Pre-registration is very uncommon and difficult in our field. Honestly, we did not know what we would find when we started this project, and we could not have predicted that we would discover APE. Because of this the experiments we ran changed over time as we started to develop our hypothesis and specific tests of the theory.

In all the cases the reviewer highlights we have now tested whether the choices we made had any effect on the results.

On picking the sessions with the best performance for the opto-inhibition experiment:

The rationale of selecting only one session per animal was taken so that the assessment of the inhibition would not be affected more by some animals. As our analysis illustrates the significance of individual sessions, we cannot merge different sessions from the same animal. By only taking one session per mouse we ended up excluding 3 sessions (from 2 animals) from a total of 30 sessions. The results from these excluded sessions can be seen in the plot below (red highlights).

Figure 1f from the paper with the bias on the three excluded sessions (red circles) included.

We picked the sessions that we included based on the best baseline psychometric performance and not whether there was any “effect”. We believed that baseline performance was a good indicator of how well an animal was doing on the task and would offer the most stable control to compare the effect of the opto stimulation to. Below we plot the psychometric curves for the selected sessions (red box) as well as the ones that were excluded. As you can see in mouse D1opto-06 our choice led us to pick a session without a significant effect of the stimulation, and in the other mouse (D1opto-05) both sessions had a comparable effect. We also compared the significance of our results if we selected other sessions, if we include session D1opto-06_Nov29 then our results become slightly more significant with a slightly larger effect size, but any other combination of sessions had no effect on the significance or

effect size of our results. We now describe our rationale for selecting our sessions better in the methods.

To increase the robustness and openness of experimental results it is becoming common in our field to share all the code and data so that other people can both reproduce our figures as well as change statistical tests or include other sessions when rerunning our analysis. This is something that we believe is important and so the code and data for reproducing this and other figures is already available at https://github.com/HernandoMV/APE_paper.

Trimming of photometry trace to maximize variance accounted for: As our task is self paced, animals often take large breaks (of up to 10 minutes per break) during a ~1 hour training session. In early training (as is the case for the regression in Figure 2f, g) these periods of task non-engagement can be even longer. During this time, they often have a rest and chew on some food we provide at the back of the box. Understandably, we do not have any task- associated behavioral events during these periods, but we do have photometry. Therefore, there is a proportion of the photometry traces that cannot possibly be explained by the behavioral events in the kernel regression. We wanted to demonstrate to what extent this limited the percentage variance explained in Figure 2g (which was calculated on the full photometry trace, even during periods where the animals are not engaged). Because we are

unusually performing this type of regression early in training, when animals are less engaged, we suspected this might be important. Because we wanted to present the data with as few assumptions as possible, we presented variance explained statistics without trimming in the main Figure 2g and compared it to the model with trimming only in Extended Data Figure 4m.

The low variance explained is common especially for movement-related dopamine signals (e.g. Parker et al., 2016 and Tsutsui-Kimura et al., 2020).

Arbitrary cut of 1 s on return events: The turns were cut off at 1s length from leaving the side port, to try to better align the return movements in time with each other. Understandably early in training there is a lot of variability in this return movement as there is no task requirement to initiate a new trial quickly. However, for the regression we did not use a cut off time for the movement to be completed in. We did restrict how direct the return movements needed to be, requiring the return trajectory to have a cosine similarity of ≥ 0.9 to the optimal vector within a 10s-time frame. This prevents highly convoluted return trajectories with multiple turns (which are hard to interpret). It should be noted these criteria only excluded $<5\%$ of return movements across the whole dataset.

To demonstrate that the 1s cutoff did not substantially change the effects we have replotted below Extended Data Figure 4 panels j and k with and without the 1s cutoff. From these plots it is clear that the cutoff does not make a substantial difference.

Left no cutoff, right 1s cutoff (Extended data figure 4 panel j):

Left: no cutoff, right: 1s cutoff (Extended data figure 4 panel k):

Below is the distribution of ipsi and contra return movement durations, also showing the 10s window within which the cosine similarity to the optimal trajectory was calculated (dashed line). As you can see, this window for calculating the cosine similarity is reasonable as the majority of movements were already completed by then.

We have updated our methods section substantially to explain these choices and highlight that removing these choices did not significantly affect the results.

Splitting data into larger and small at arbitrary cuts: We followed the methods provided in Lak et al. 2020 as this was the most comparable previous work to the analysis we wanted to perform. However, we do acknowledge that this does not make use of the full variance in the dopamine response size data, as pointed out by a reviewer. We have therefore additionally performed a logistic regression analysis as the reviewer very helpfully suggested (see below in answer to specific question about this analysis). We still include the split data plots for visualization purposes, but all the regression analysis is now done with the logistic regression as the reviewer suggests.

Other cases: We also evaluated other cases where our analysis might have suffered from these arbitrary choices. One case was in the DA stimulation experiments, where we chose to exclude the first initial 75 trials of the stimulation block. This was done to allow the animals to “sample” the new conditions. We now refrain from using this arbitrary sampling period, and instead now analyze all the trials in the block.

3. In Figure 1H and elsewhere, I am not sure how this is a learning rate and not wholly co-extensive with max level? How well does the sigmoid formula used represent what animals actually did? Learning is often negatively accelerated (not sigmoid), and that also seems to

be true in the current data. In addition, this approach assumes a smooth, continuous learning, which does seem apparent in group curves, but may not represent individual animals (not shown here). There is a lot of work on fitting learning curves, showing that they are often not smooth for individuals and better thought of as transitions and modelled with Weibull functions (e.g., Gallistel et al, 2004, in PNAS).

We thank the reviewer for these comments, especially the suggestion about using Weibull function which we were unaware of and does indeed fit our data better (see below for details). From looking at our individual animals' data we can see that learning rate and max level are not wholly co-extensive. Some animals learn rapidly but reach a lower level of performance and vice versa, because of this we report the learning rate and the max level of performance to capture both differences across animals. This is illustrated below with the mice used in (Figure 1k-n, the 6-OHDA control and lesioned mice).

Individual learning curves (performance vs trial number) for the 15 mice used in Figure 1k-n. The data from each session is plotted in a different colour. The black and red lines represent the fit of the Weibull and Sigmoid fits respectively.

As the reviewer points out, learning does not always progress smoothly. Indeed, as can be seen in the data above there are often sessions with rapid improvements in performance. Despite this the sigmoid and Weibull functions still do a good job of capturing the trends in the learning curves for individual mice. As can be seen in the plots above, fitting with the Weibull function (black line) instead of a sigmoid (red line) is generally a slightly better fit for our data. To quantify how much better the fit was we calculated the difference in the sum of errors squared fit for the Weibull and sigmoid functions (negative values when the Weibull fits best), plotted below for the control and 6-OHDA lesioned mice.

The difference in the sum of errors squared fit for the Weibull and sigmoid functions for each of the 15 mice used in Figure 1k-n. Data is split for the control (blue) and the 6-OHDA lesioned mice (brown). Negative values indicate when the Weibull function fits best, i.e. it has a smaller sum of errors squared fit.

This analysis revealed that the Weibull function performs better in general for both the 6-OHDA and Caspase lesion datasets. Because of this we have substituted the fittings of all learning curves in the manuscript, and we now calculate the maximum learning rate as the maximum derivative of the Weibull fit. This change in the fitting did not affect the significance of the effect size of any of our results. We show an example of this below where we used the Weibull fitted curves (new Figure 1n) or the original sigmoid fit results (old Figure 1n) to determine the maximum learning rate.

Weibull fitting

Sigmoid fitting

4. Fig 5. If dopamine in the VS is encoding an RPE, why would it be the case that responding on the next trial was not related to VS responding on the previous trial? An RPE should increase the value of that option and make it more likely to be selected on the next trial. This, of course, is one of the challenges of distinguishing the impacts of reward (value-based) against the impacts of repetition (value-free) and may need to a targeted experiment to disentangle.

This is an important and interesting point that the reviewer raises. To confirm that we do not see a bias induced by optogenetically stimulating dopamine release in the VS we repeated our stimulation experiments using a higher power (8mW vs 4mW). As with our previous experiments this stimulation did not bias mice towards choosing the stimulated port (Figure 5f). This matches what we see with the endogenous DA signal in the VS, where the size of the VS dopamine signal at the time of outcome is not correlated with the mice repeating these choices. It should be noted that the analysis and optogenetic stimulation experiments were conducted in expert mice where performance on that task was >90% accuracy.

One possibility is that we would have seen a different result if we had optogenetically stimulated VS DA release in early learning when behavioural performance was less dependent on the TS. This is because we have shown that in expert mice the behavioural performance is almost exclusively dependent on the TS so any changes caused by our VS optogenetic stimulation may be insufficient to counteract the output of the TS on the short timescale of our optogenetic experiments. In line with this idea, we optogenetically stimulated DA release in the VS and were able to induce a choice bias in a free choice paradigm where there is no established sensory-motor mapping, as had previously been shown (Menegas et al., 2018), this is now reported as Extended Data Figure 8i,j. We did not perform the early stimulation experiments as it is difficult to know even in early learning how much of the behaviour is controlled by the TS or the value-based controller so it would have been difficult

to interpret the results. Because of this we chose to show that we could induce a choice bias in the free-choice paradigm where behaviour is not influenced by the TS.

5. The RL modeling was perhaps the least convincing part of the paper. First, the base model used a semi-Markov formalism that only appears in a (very) small handful of papers and certainly does not represent the standard way that RPEs are formalized. This is even acknowledged in the paper where the schematics (e.g., Figure 7) do not correspond to the model at hand. What justifies this choice of formalism, and would the results generalize to a more standard RL treatment of the RPEs? Beyond that, there are many details missing in the APE (value-free) component of the model, which would seem to be the most important (and novel) component. For example, there is no update equation for the action component and no details on the number of actions in the model, nor what counts as a transition for the semi-Markov model. Without these details, it is very difficult to assess the functioning of the model, its expected behaviour and how well it explains the neural results. In addition, although the paper cites two pieces of theoretical work on APEs as the inspiration for the current work, those models are not directly used, and a novel formalism is adduced yet not directly compared to that prior work.

We thank the reviewer for astutely noting the missing action component update equation, this slip has been remedied. We have also substantially updated the methods section detailing the RL model and our reasons for choosing the semi-Markov formalism. We have also elaborated more on these below.

The actions and state transitions in the model are detailed in full in Extended data figure 6 which shows the complete formalism of the task. We have also drawn more attention to this in the methods.

Our main reason for using the semi-Markov formalism was that it allows direct comparison to the dopamine signals across time within a trial, in our task which has variable timing. It also has the advantage that states best represent the within-trial behavioral stages of the task, whilst allowing for representation of time. In a task with variable timing, this approach appeared to be the most suitable state representation for accurately capturing the structure of the task. The obvious other options, such as a serial compound stimulus (CSC) representation, assigns a separate state to each time point between the cue and the reward. As shown by Daw et al. 2006, the temporal difference errors in that model do not capture the temporal evolution of the dopamine signal, whereas the semi-Markov representation does. Furthermore, while the reviewer rightly points out that the semi-Markov formalism is not widely used in the field, several key studies have used the semi-Markov formalism to model dopamine signals in time in a task with variable timing (Starkweather et al. 2018, Gershman & Uchida 2019). Accordingly, in our view, the semi-Markov representation is a simple way to capture dopamine signals in time in variable timing tasks, in line with important prior literature.

That said, as the reviewer points out, it is important to verify that our core model predictions are not dependent on this choice of time representation. To see that our model is very similar to more standard reinforcement learning applications, consider the definition of value in the semi-Markov model:

$$\hat{V}(s_k) = E[r_{k+1} - \rho d_k + \hat{V}(s_{k+1}) | s_k]$$

As explained in the paper, this definition of value includes dwell time, d_k , and average reward, ρ . In the special case that the dwell time for each state is set to zero (the animal immediately moves away from each state), this definition reduces to the standard definition of value (undiscounted). The ensuing learning algorithm then amounts to standard temporal difference TD(0) learning, widely used in modelling of DA signals.

We have re-run our simulations setting the dwell time to zero for each state. Note that, with this setting, it is not possible to realistically model the within-trial neural traces that we show for the semi-Markov formalism. However, the general predictions of the model (shown in Figure 3) (APE should decrease as RPE cue response increases) are the same. X-axes show trials (arbitrary units as learning rates are not fit to match the data).

The behavior modeling in Figure 6 was defined on a trial-by-trial basis, without accounting for within-trial time. For that reason, we did not use a semi-Markov model there. Instead, the model presented there was meant to illustrate the strength of having two separate systems and their interaction.

We have also adapted the dual controller model so that it has a more standard Critic component that only learns the value of the states and has no role in action. The schematic in Figure 6a is now an accurate reflection of the connections in the model. We have adjusted the methods to make it clear how information is represented in each part of the model and describe in more detail the important update equations.

D. Appropriate use of statistics and treatment of uncertainties

- In Figure 5D, why are the statistical tests compared against a shuffled dataset, rather than simply against one another?

We did not compare groups because this would not reflect the question we are trying to ask. We wanted to know if there was an effect of optogenetically stimulating DA release in the TS or the VS, not whether there was a difference between stimulating DA release in the TS or VS. As another reviewer pointed out there are potential issues with comparing the experimental group to the shuffled dataset so we now use a one sample two-sided (Wilcoxon signed-rank) test comparing the effect of each group against zero.

- Throughout, there are no effect sizes provided to go along with the inferential statistics

Thank you for suggesting this, it is not often done in our field, but it does represent an important piece of information to add for each test. This information has now been added for all our statistical tests.

- The analysis on p. 29 that arbitrarily splits data into small/large at the 65th percentile throws away a lot of the data. A more appropriate analysis would seem to be a logistic regression with size as a continuous predictor.

We agree with the reviewer that a logistic regression is desirable as it does not throw away any variation in the data. We had previously chosen to split the data to repeat the analysis done by Lak et al. to allow for direct comparison to previous work. As suggested by the reviewer, we have now also performed a logistic regression predicting repetition of a previous choice from the previous trial dopamine response size (now a continuous predictor) and log uncertainty. The regression coefficients for the two predictors are shown in the barplot below. There was no trend in relationship with an interaction term, so this has been removed from the regression. This shows that a larger DA response in the TS but not VS response correlates with animals repeating that choice. This analysis is now included in Figure 5i, k.

Left: Regression coefficients from a logistic regression predicting repetition of a previous choice from the previous trial TS dopamine response size (at time of contralateral choice) and log perceptual uncertainty (n=4 mice) (log uncertainty: $p=0.006$, Cohen's $d=3.31$, dopamine (at time of choice): $p=0.03$, Cohen's $d=2.01$). Filled circles represent significant correlations for individual mice. Right: As in the left panel but for VS dopamine at time of choice. (n=5 mice) (log uncertainty: p value= 0.03, Cohen's $d= 1.44$, dopamine (at time of cue): p value=0.29, Cohen's $d=-0.54$).

E. Conclusions: robustness, validity, reliability

See comments elsewhere.

F. Suggested improvements: experiments, data for possible revision

See first comment in section C for suggested improvement with regards to proving that these responses represent an APE.

I'd also like to see more consideration of the goal-directed vs habitual distinction (and connection with DMS/DLS work). Presumably, these action prediction errors are particularly important for (in)forming value-free habitual behaviour, yet they seem to get smaller as learning goes on. How do the proposed mechanisms fit with existing ideas about the emergence and maintenance of habitual behaviour?

We think that the relationship between APE and habit formation is an important one and we thank the reviewer for asking us to explore this connection more. We have altered our discussion to include reference to the important literature about the connection between the DMS/DLS and goal-directed/habitual behaviour.

It should be noted that there is a beautiful theory paper (Miller et al., 2019) that initially proposed that an APE like signal could be used to drive habit formation. In this paper they show a dual value-based/value-free control system that is updated by RPE and APE can explain all the main behavioural findings regarding habit formation. We cite this paper heavily and it does a much more complete job than we can, due to space, of explaining why APE would theoretically provide a better explanation for habit formation than the dominant computational view where model-based and model-free RPEs are used to update the weights for goal-directed and habitual behaviour respectively. We further emphasize this paper in the discussion.

In response to the specific question, it may seem puzzling that a teaching signal that is proposed to drive habit formation gets smaller later in learning at the time point when habits are meant to be "forming" or at least being consolidated. While behaviour may not be under habitual control until later in learning this is not to say that the weights in the habitual controller are not being strengthened throughout the course of learning. Indeed, in the dominant computational model, where model-free/model-based RL algorithms are proposed to drive habitual and goal-directed learning respectively, both learning systems are updated from the very beginning of learning (e.g. Dolan & Dayan, 2013; Doll, Simon, & Daw, 2012). In line with RPE theory the model-free RPE updates for the habitual controller will also get smaller over

time as this system learns to predict the value of the action/states. The proposed reason for the switch from goal-directed to habitual control is not therefore that the teaching signal that updates the habitual controller gets larger over time rather it is proposed that there is an arbitration mechanism in the brain that selects which controller to use based on the degree of certainty that each controller can predict a reward (Daw, Niv, & Dayan, 2005).

More recently dual value-based/value-free control systems have been proposed as a more parsimonious explanation of habitual behaviour (Miller et al., 2019). In this seminal theory paper they proposed the value-free teaching signal would take the form of what we now call an action prediction error (Miller et al., 2019). In their model they were able to show that a dual value-based/value-free system updated by RPE and APE could explain all of the classic results of habitual behaviour. This new model shares some features with popular model-free/model-based models. 1) Both control systems are updated from the start of learning, 2) The size of habitual teaching signals, APEs or model free-RPEs, reduce over learning, 3) They use a Bayesian arbitration mechanism to switch control between the systems.

Taken together in both the model-based/model-free and the RPE/APE theory for habit formation, the teaching signals that drive habit formation decrease over the course of learning. The decrease in the size of the APE over learning is therefore still consistent with a teaching signal that drives habit formation.

G. References: appropriate credit to previous work?

Yes.

H. Clarity and context: lucidity of abstract/summary, appropriateness of abstract, introduction and conclusions

The text (and figures) were exceptionally dense, and it took multiple reads to parse the paper. One easy and major improvement would be to provide a little more methodological detail about the behavioural task in the main text. Simply referring to the “cloud of tones” task did not suffice, which made both deciphering the text difficult, but even all the figures which referred to “% of tones”, rather than frequency. After reading the methods in detail, this makes more sense, but made the paper proper hard to understand.

We apologize for this oversight. We have now added a more detailed description of the task in the main text *“Mice initiate trials by poking in the center port. On each trial the stimulus consisted of a short train of overlapping pure tones distributed over a three-octave range (5–40 kHz). Mice were required to choose between a right and a left reward port depending on whether the stimulus contained mostly low-frequency (5–10 kHz) or high-frequency (20–40 kHz) tones”*.

We have also given a visual example of the stimuli that we present in (Extended Data Figure 1a).

The section on the dual controller was particularly difficult to understand. The section seems to describe a network model (with weights etc), but the specification is such that I could not really understand the structure of the model, the different components, nor how it fits together.

We apologize for this. We have now substantially changed our results and methods section to better explain how the dual controller model is constructed.

- Small point: middle p.15 “negatively reinforce” is misused (it actually means to increase behaviour through the absence of a punishment).

Thank you for clarifying this. We have changed the text to say “our TS dopamine stimulation was not able to induce avoidance behavior”.

Overall, we would like to thank the reviewer for their careful reading of the manuscript and their excellent suggestions. We believe that addressing their points has made the paper stronger.

Referee #1 (Remarks to the Author):

The authors responded thoroughly to all my comments and appeared to do a commendable job with the excellent issues raised by the other Referees. I only offer a few minor points that the authors can address as they see fit.

The readability of some figure text is difficult due to the color and thickness, e.g. 3a 'late' (the text could just be in black, or if color coded, changed to be slightly darker... we'll get the point), Ext Data 9, etc.

The light tones have now been made darker.

I appreciate the shading used to determine significant difference. The authors may want to place these bars at the bottom of the plot, as the shading done in the current way could be confused by an unaware reader as implying the use of orange light stim. This is not required, just a suggestion.

We now more clearly highlight what the shading refers to in the legend

I encourage the authors to expand beyond model-free and make further use of the growing appreciation of policy-based learning rather than value-based. Bennett, Niv, and Lagdon have an excellent review of this and other studies like Hodge et al (bioRxiv) provides corroborative evidence for the idea and the work here.

We agree with the reviewer that the growing appreciation of policy-based learning is important but given space constraints we do not think we can adequately discuss it here.

I applaud the use of the Frechet distance and its inclusion in the main document. As it is not a common metric in much of neuroscience, I might suggest a more thorough description of it in the methods or even results section.

We thank the reviewer for this comment. We have added a little more detail about this method to the methods section

The authors seem to indicate Yttir and Dudman studied DLS, but their work was in DMS if I recall.

We apologies for this mistake this has now been corrected.

Referee #2 (Remarks to the Author):

The authors have performed additional experiments and analyses to address my previous concerns. The optogenetic stimulation experiment was repeated using a stronger stimulation (8 mW instead of 4 mW). Although it remains to be determined whether the 8 mW stimulation condition is physiological, it produced a stronger effect and thus partially replicated the previously observed effect. Ideally, the authors should have also conducted a replication using the previous condition (i.e., 4 mW), but I do not consider it necessary to request that here. The experiment with 8 mW increases confidence in the result. The revised text also provides important information. This work presents an interesting set of results. If

further verified and refined, it could provide a novel framework for understanding diverse dopamine signals.

We thank the reviewer for these comments.

Referee #3 (Remarks to the Author):

The authors have done an excellent job of addressing my concerns about the previous version. I only have some minor suggestions to improve a further version. In particular, from my perspective, the key addition in this revision is the trial-to-trial analysis (now Fig 3), which was really the missing piece to establish this signal as an APE. The modelling work is also much better justified and explained now, though I still think the meatier contribution is the novel neural evidence for the previously theorized APE signal. Also, though adding further explanation about the analytic choices was helpful, I already thought each one was reasonable, as post-hoc analyses often seem to be. Showing the results are robust to other analytic choices was a much better counterpoint.

Here are the minor suggestions I have for improving this version of the manuscript:

- The little paragraph above Fig 3 seems to recap only one of the points in the previous section (and the less convincing one at that). The point about trial-to-trial changes in TS responding is omitted. Same with the caption to Fig 3 and the lead sentence of the next section.

We thank the reviewer for pointing this out. We have now included the trial-to-trial results in the recap and lead sentences.

- The new text describing the logistic regression needs to be stitched into the text better. Right now, it reads more like a response to the previous version (i.e., “without splitting the data” and “now a continuous predictor”), as though the reader had just gone through the old split analysis.

We have better integrated the explanation of these results.

- p. 18. “Classically for instrumental learning” is a confusing use of the word “classically”, which of course evokes classical conditioning (not operant). Easy switch for a different word (typically, commonly, etc...).

This was a good point. We now use the word “typically”.

- Not great that the photometry datasets are being withheld for future use (results are less verifiable and extensible), but the journal policies will dictate whether that’s ok (which it shouldn’t be).

This data has now been made publicly available.

- The Eiselt et al, 2021 paper that is used as source for the modified Weibull does not seem to appear in the references. The Gallistel paper I noted last time should probably get some credit here (it was their innovation).

These two references have now been included.

- Minor quibble with the semi-markov justification: There are other ways to handle variable timing with standard TD/RL (e.g., the microstimulus model as discussed in the newer Sutton & Barto text which also appears as the base model in the Nambodiri papers). The complete serial compound idea was always recognized as a computationally convenient fiction.

This is a very good point. We have a collaboration with Rafal Bogacz's lab where they have used microstimuli to deal with the variable timing and their model can still replicate our results. This data is not included as it is part of a different paper.